# VERITAS: GENERALIZABLE DEEPFAKE DETECTION VIA PATTERN-AWARE REASONING

**Hao Tan**[1,2,3*] **Jun Lan**[3†] **Zichang Tan**[4] **Senyuan Shi**[2] **Ajian Liu**[2]
**Chuanbiao Song**[3] **Huijia Zhu**[3] **Weiqiang Wang**[3] **Jun Wan**[1,2,5§] **Zhen Lei**[1,2,5]

[1]School of Advanced Interdisciplinary Sciences (SAIS), University of Chinese Academy of Sciences
[2]MAIS, Institute of Automation, Chinese Academy of Sciences     [3]Ant Group
[4]Shenzhen Institute of Advanced Technology (SIAT), Chinese Academy of Sciences
[5]School of Artificial Intelligence, University of Chinese Academy of Sciences
{tanhao2023, jun.wan, zhen.lei}@ia.ac.cn    yelan.lj@antgroup.com
[†]Project Lead.   [§]Corresponding author.   ⬤ Project Page: https://github.com/EricTan7/Veritas

## ABSTRACT

Deepfake detection remains a formidable challenge due to the evolving nature of fake content in real-world scenarios. However, existing benchmarks suffer from severe discrepancies from industrial practice, typically featuring homogeneous training sources and low-quality testing images, which hinder the practical usage of current detectors. To mitigate this gap, we introduce **HydraFake**, a dataset that contains diversified deepfake techniques and in-the-wild forgeries, along with rigorous training and evaluation protocol, covering unseen model architectures, emerging forgery techniques and novel data domains. Building on this resource, we propose **VERITAS**, a multi-modal large language model (MLLM) based deepfake detector. Different from vanilla chain-of-thought (CoT), we introduce *pattern-aware reasoning* that involves critical patterns such as "planning" and "self-reflection" to emulate human forensic process. We further propose a two-stage training pipeline to seamlessly internalize such deepfake reasoning capacities into current MLLMs. Experiments on HydraFake dataset reveal that although previous detectors show great generalization on cross-model scenarios, they fall short on unseen forgeries and data domains. Our **VERITAS** achieves significant gains across different out-of-domain (OOD) scenarios, and is capable of delivering transparent and faithful detection outputs.

## 1 INTRODUCTION

Recent advances in Generative AI (Esser et al., 2024; Tian et al., 2024) have revolutionized our digital life, unprecedentedly enriching the diversity of content on social media and short-video platforms. Though bringing immense creativity, such techniques also enable highly convincing deepfakes with minimal cost, posing significant security risks to society. Consequently, Deepfake Detection (DFD), which aims at discerning between real and generated facial images, has become a heated research frontier, galvanizing extensive efforts.

However, current detectors mostly follow a standard evaluation, which involves training on one dataset (Rossler et al., 2019) and testing on others (Dolhansky et al., 2019; Li et al., 2020b; Dolhansky et al., 2020; Zi et al., 2020; Zhou et al., 2021). Despite its popularity, this protocol fails to align with practical industrial scenarios, where abundant training samples are available yet significant out-of-distribution (OOD) generalization challenges (e.g., brand-new forgery types and meticulously synthesized facial images) emerge during testing. Such ***discrepancy*** severely hinders the practical deployment of current detectors. To mitigate the gap, we construct **HydraFake** dataset. As shown in Figure 2, we systematically collect and reproduce advanced deepfake methods, covering diversified deepfake techniques and in-the-wild forgeries from social media. To simulate potential challenges in real-world scenarios, we establish a rigorous and holistic evaluation protocol, where the training set consists of abundant samples but is restricted to three basic forgery types, and the

---

[*]This work was done during the first author's internship at Ant Group.

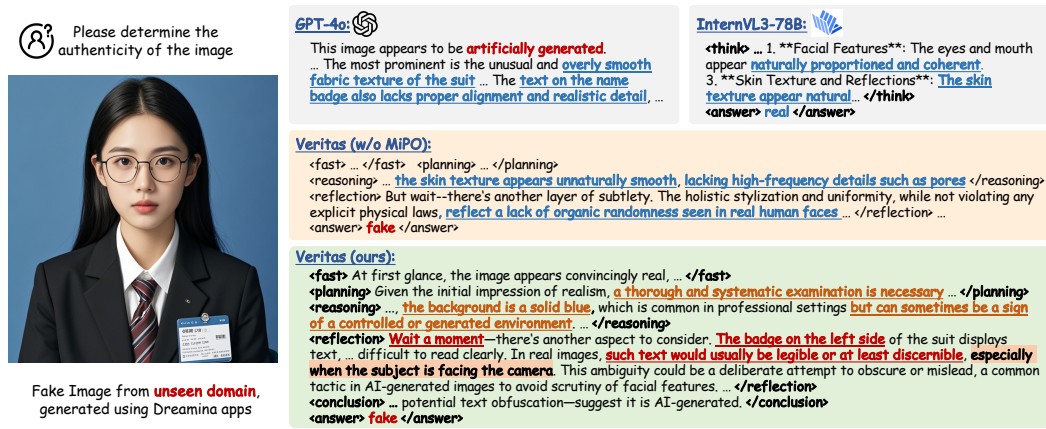

Figure 1: Comparison of the detection outputs. InternVL3-78B (Zhu et al., 2025) gets incorrect answer. GPT-4o (Hurst et al., 2024) and our model trained without the proposed MiPO both fail to provide precise explanation. In contrast, our model gives transparent and faithful decision process.

evaluation involves hierarchical OOD testing, spanning in-domain, cross-model, cross-forgery and cross-domain scenarios, enabling fine-grained understanding of the model's capacities. As presented in Figure 2 (d), under such rigorous evaluation, current SOTA detectors show great generalization on cross-model deepfakes, but limited abilities in cross-forgery and cross-domain scenarios.

To improve the robustness on unseen forgeries and data domains, we seek to ground the generalization abilities of multi-modal large language models (MLLMs) into deepfake detection. Recent efforts (Huang et al., 2024; Guo et al., 2025b; Peng et al., 2025) have made initial attempts, while they focus on the explainability and the classification is still based on expert vision models. In contrast, we explore to seamlessly internalize MLLMs into deepfake detection through their intrinsic reasoning abilities. However, directly applying deep reasoning faces a critical challenge: current MLLMs are extremely short for deepfake detection (Ren et al., 2025; Tariq et al., 2025). Effective reasoning data is necessary to ground the abilities of base model. To achieve this goal, we must answer two key questions: **(1)** what kind of reasoning process is helpful to DFD task? and **(2)** with sufficient data, how can we ensure the model is learning to reason for DFD rather than memorizing?

**For the first question**, we introduce a pattern-aware reasoning framework. Drawing inspiration from recent studies (Zhao et al., 2025; Muennighoff et al., 2025) that demonstrate critical *reasoning patterns* greatly elevate the OOD performance of LLMs, we consider the human mindset for deepfake detection: when determining the authenticity of an image, we tend to make a quick judgment based on our first impression (*fast judgement*), then identify one or two prominent features (*reasoning*) to draw a conclusion (*conclusion*). For more challenging samples, we may conduct a layered analysis (*planning*), and may also engage in more in-depth thinking to support or overturn our initial judgement (*self-reflection*). Based on this analogy, we extract these five thinking patterns to facilitate logical and holistic reasoning. Table 2 empirically shows the benefits of such pattern-aware reasoning over vanilla Chain-of-Thought (CoT). **For the second question**, we introduce a two-stage training pipeline consisting of pattern-guided cold-start and pattern-aware exploration, yielding our VERITAS [1] model. During cold-start, we employ SFT to internalize thinking patterns. Besides, we introduce a Mixed Preference Optimization (MiPO) strategy that leverages mixed non-preference data and human-annotated preference data to steer the model toward faithful and fine-grained reasoning. As shown in Figure 1, MiPO greatly improves the reasoning quality, mitigating the memorizing behavior. To further facilitate adaptive planning and self-reflection, we propose Pattern-aware Group Relative Policy Optimization (P-GRPO), which shapes reasoning behavior through online sampling and pattern-aware reward mechanism. As a result, VERITAS shows great generalization on unseen forgeries and data domains, providing transparent and precise decision process (Figure 1).

To sum up, our main contributions are:

---

[1] VERITAS means "Truth" in Latin.

- **Dataset**: We introduce **HydraFake**, a dataset that simulates real-world challenges with hierarchical generalization testing, advancing the evaluation protocol in deepfake detection and helping developers better locate the deficiencies of their detectors.

- **Method**: We propose a two-stage training pipeline that grounds the capabilities of MLLMs into deepfake detection through pattern-aware reasoning. Our model supports adaptive planning and self-reflection, delivering transparent and human-aligned decision end-to-end.

- **Performance**: Our VERITAS model achieves significant improvements over state-of-the-art detectors on cross-forgery and cross-domain scenarios, and our cold-start model serves as a strong reasoning foundation for further customization.

## 2 RELATED WORK

### 2.1 DEEPFAKE DETECTION AND DATASETS

Deepfake detection aims to distinguish generated facial images from authentic human faces. Previous efforts have explored spatial-level (Ojha et al., 2023; Yan et al., 2024b; Tan et al., 2024b; Nguyen et al., 2024; Fu et al., 2025; Yan et al., 2024a;c; Yang et al., 2025c), frequency-level (Qian et al., 2020; Tan et al., 2024a; Zhou et al., 2024; Kashiani et al., 2025) and sequence-level (Gu et al., 2021; 2022b;a; Yan et al., 2025) approaches, achieving remarkable progress on traditional benchmarks. To train a generalizable detector, some methods attempt to find "bias-free" fake images either through spatial-domain blending (Li et al., 2020a; Shiohara & Yamasaki, 2022; Zhao et al., 2021), frequency-domain blending (Zhou et al., 2024; Kashiani et al., 2025) or feature-level augmentation (Yan et al., 2024b). However, the commonly adopted protocol, i.e., training on FF++ (Rossler et al., 2019) and testing on others (Dolhansky et al., 2019; Li et al., 2020b; Dolhansky et al., 2020; Zi et al., 2020; Zhou et al., 2021), suffers from two problems: (1) the training sources are overly narrow, and (2) the testing data exhibit limited forgery types and low-resolution. Although many timely datasets (Yan et al., 2024a; Zhang et al., 2024b; Li et al., 2025; Huang et al., 2025b; Wang et al., 2025a; Wen et al., 2025a; Xia et al., 2025) have been proposed for AIGC detection, the pace of deepfake detection has lagged behind. As a result, previous methods are biased towards such settings, exhibiting degraded generalization when learning from varying sources or mixed artifacts. To mitigate this problem, we introduce a hierarchical protocol in our HydraFake dataset, aiming to comprehensively reflect the generalization capability of the detectors.

### 2.2 MLLMS FOR DEEPFAKE DETECTION

With the proliferation of MLLMs (Liu et al., 2023; Bai et al., 2025; Zhu et al., 2025), recent focus has shifted to explainable deepfake and AIGC detection. However, most methods still rely on small vision models for the final decision. For instance, M2F2-Det (Guo et al., 2025b) determines the authenticity purely based on CLIP models, where LLM is leveraged as a plug-in interpreter. Similarly, DD-VQA (Zhang et al., 2024a), FFAA (Huang et al., 2024) and VLF-FFD (Peng et al., 2025) develop post-processing system to aggregate embeddings from small vision models. Some methods (He et al., 2025; Sun et al., 2025; Chen et al., 2025b) attempt to directly adopt the outputs from LLMs, e.g., Sun et al. (Sun et al., 2025) construct precise forgery explanations to release the power of MLLMs. Recent methods (Huang et al., 2025a; Xu et al., 2024b; Zhou et al., 2025) also adopt MLLMs and curated datasets for AIGC detection. However, these methods generate post-hoc explanations by first determining the answer. The potential of reasoning abilities for deepfake detection is still underexplored. The most recent methods (Gao et al., 2025; Xia et al., 2025) explore the reasoning for AIGC detection, while neglecting adaptive reasoning patterns and is not tailored for facial forgery. Different from previous methods, we introduce human-like reasoning into deepfake detection, achieving promising improvements and delivering transparent decisions end-to-end.

## 3 HYDRAFAKE DATASET

In this part, we introduce our HydraFake dataset, including the construction process and evaluation protocol. Detailed statistics and information are provided in Appendix A.1.

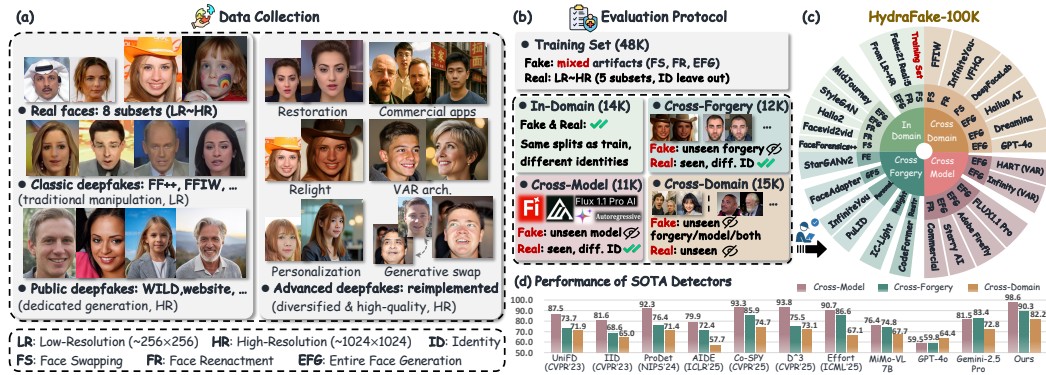

Figure 2: **Overview of HydraFake dataset**. **(a)** We carefully collect and reimplement advanced deepfake techniques to construct our HydraFake dataset. Real images are collected from 8 datasets. Fake images are from classic datasets, high-quality public datasets and our self-constructed deepfake data. **(b)** We introduce a rigorous evaluation protocol. Training data contains abundant samples but limited forgery types. Evaluations are split into four levels. **(c)** Illustration of the subsets in different evaluation splits. **(d)** Performance of existing detectors. Most detectors generalize well on Cross-Model setting but perform poorly on Cross-Forgery and Cross-Domain scenarios.

## 3.1 DATA COLLECTION

**Real Images.** As shown in Figure 2 (a), the real images are collected from 8 public datasets, containing both low-resolution (i.e., LFW (Huang et al., 2008), CelebA (Liu et al., 2015), FaceForensics++ (FF++) (Rossler et al., 2019), FFIW (Dolhansky et al., 2019)) and high-resolution images (i.e., FFHQ (Karras et al., 2019), VFHQ (Xie et al., 2022), UADFV (Yang et al., 2019) and CelebAHQ (Karras et al., 2017)). The collected images are rigorously partitioned for training and testing.

**Fake Images.** The fake images come from three sources:

- **Classic deepfake** data sampled from FF++ (Rossler et al., 2019) DF40 (Yan et al., 2024d) and FFIW (Dolhansky et al., 2019), which mainly contain face swapping (FS) and face reenactment (FR) forgeries from 10 generative models. The artifacts are mostly localized.

- **Public deepfake** data sampled from WILD (Bongini et al., 2025), seeprettyface website and TalkingHeadBench (Xiong et al., 2025). This contains carefully synthesized faces from 16 popular generators. However, there still exist corner cases such as fresh forgery types.

- **Advanced deepfake** data where we further reimplemented and crawled 10K deepfake data from 10 advanced generators. Besides traditional deepfake techniques, HydraFake dataset contains Face Restoration (Zhou et al., 2022), Face Relighting (Zhang et al., 2025b), Face Personalization (Jiang et al., 2025; Guo et al., 2024), Generative Face Swapping (Han et al., 2024) and deepfakes from Visual AutoRegressive models (VAR) (Han et al., 2025; Tang et al., 2024). To simulate real-world challenges, we also crawled 1K deepfake images from social media, which include practical deepfakes generated from commercial apps, including GPT-4o (Hurst et al., 2024), Dreamina (team, 2025a) and Hailuo AI (team, 2025b).

**Quality Control.** For classic deepfake datasets, we only select FF++ (Rossler et al., 2019) and FFIW (Zhou et al., 2021), while not involving DFDC (Dolhansky et al., 2020), DFDCP (Dolhansky et al., 2019) and WDF (Zi et al., 2020) due to their low quality (e.g., unexpected blurring in real images). For our self-constructed deepfake data, we conduct strict quality control, e.g., for face personalization, we use Qwen2.5-VL-72B to tailor sample-specific prompts rather than using template-like prompts as in (Bongini et al., 2025). For face relighting, we generate multiple lighting sources for each identity and manually select high-quality samples. After filtering and balancing, our HydraFake dataset contains 50K real images and 50K fake images.

## 3.2 EVALUATION PROTOCOL

**Training.** As shown in Figure 2 (b), the training set contains 48K images. Real images are from 5 subsets, with other 3 subsets left out for testing. Fake images involves 21 subsets while only contains

3 forgery types (i.e., FS, FR and EFG). This is to simulate practical setting, where abundant training images are available but various forgery types and generative models remain unseen.

**Evaluation.** The evaluation is divided into four distinct levels:

- **In-Domain** (14K): testing images share the training data source but with different identities.
- **Cross-Model** (11K): fake images are generated by unseen models under controlled conditions like the template-based textual prompts. This includes SOTA models from recent years (e.g., FLUX1.1-Pro (Black Forest Labs, 2024), Adobe FireFly (Adobe, 2023), Starry AI (AI, 2023)), distinct model architectures (e.g., VAR (Han et al., 2025; Tang et al., 2024) and Video AR model (Sand-AI, 2025)). The real images are from in-domain set but with different identities.
- **Cross-Forgery** (12K): fake images are generated by unseen manipulation techniques, involving attribute editing, generative face swapping, IP-preserved personalization, face relighting and face restoration. The real images are from in-domain set but with different identities. This split is to evaluate the model's capacity to detect fake images generated by unseen manipulation.
- **Cross-Domain** (15K): fake images are either generated under controlled conditions or collected from the web, including both unseen forgeries and unseen models. The real images are from unseen datasets (i.e., VFHQ (Xie et al., 2022), UADFV (Yang et al., 2019) and FFIW (Dolhansky et al., 2019)). The images are of different qualities, posing strong challenges.

## 4 METHOD

In this section, we detail the two-stage training pipeline of **VERITAS**, including pattern-guided cold-start and pattern-aware reinforcement learning, as shown in Figure 3.

### 4.1 PATTERN-GUIDED COLD-START

To internalize thinking patterns for deepfake detection, we first employ a pattern-guided cold-start. Different from common practice, we involve two steps: Supervised Fine-Tuning (SFT) for format injection, and a Mixed Preference Optimization (MiPO) strategy to align the reasoning process.

**SFT Pattern Injection.** Suppose the SFT dataset is denoted as $\mathcal{D}_1 = \{(\boldsymbol{q}, \boldsymbol{s})_i\}_{i=1}^{N_1}$, where $\boldsymbol{s}$ is the target output sequence including pattern-aware reasoning and final answer. $\boldsymbol{q}$ denotes input image and user query. The training objective maximizes the likelihood of generating $\boldsymbol{s}$ given input $\boldsymbol{q}$:

$$\mathcal{L}_1 = -\mathbb{E}_{(\boldsymbol{q},\boldsymbol{s})\sim\mathcal{D}_1} \sum_{t=1}^{T} \log \pi_\theta(\boldsymbol{s}_t \mid \boldsymbol{q}, \boldsymbol{s}_{<t}), \tag{1}$$

where $\pi_\theta$ denotes the token distribution from the current model. In the following we introduce the construction process of our training data $\mathcal{D}_1$.

To minimize human costs, we use MLLMs for automated annotation, similar to recent practices (Huang et al., 2024; Xu et al., 2024b). However, this encounters two challenges in our case: (1) The MLLMs tend to overlook some subtle artifacts like abnormal optical focusing. (2) The model prioritizes producing logical paths than to accurately locating artifacts. To mitigate the two issues, we construct a multi-step annotation pipeline. We first manually inspect a subset and summarize a comprehensive artifacts taxonomy (Figure 9 (a)): (1) **Perceptible** structural anomalies, which are immediately visible and easy to detect. (2) **Subtle** low-level artifacts, which require careful inspection. (3) **Cognitive** violations of physical laws, which are implicit and require connecting to common sense or real-world knowledge. Then, we decouple the annotation into three *specialized yet coherent* steps (Figure 9 (b)), resulting in 36K samples for $\mathcal{D}_1$. Detailed process and all prompt templates are provided in Appendix A.4. Annotated examples are presented in Figure 9 (c).

**MiPO Reasoning Alignment.** To further facilitate human-aligned reasoning, we meticulously curate a mixed preference dataset $\mathcal{D}_2 = \{(\boldsymbol{q}, \boldsymbol{s}_w, \boldsymbol{s}_l^{\phi})_i\}_{i=1}^{N_2} \cup \{(\boldsymbol{q}, \boldsymbol{s}_w, \boldsymbol{s}_l^{\psi})_i\}_{i=1}^{N_2'}$. Specifically, we collect two types of non-preference data for the fake images: (1) the trajectories where the answer is correct but the reasoning content is not precise or detailed enough (i.e., $\boldsymbol{s}_l^{\phi}$). (2) the trajectories where the answer is incorrect (i.e., $\boldsymbol{s}_l^{\psi}$). $\boldsymbol{s}_w$ denotes preferred reasoning traces, which are precisely annotated

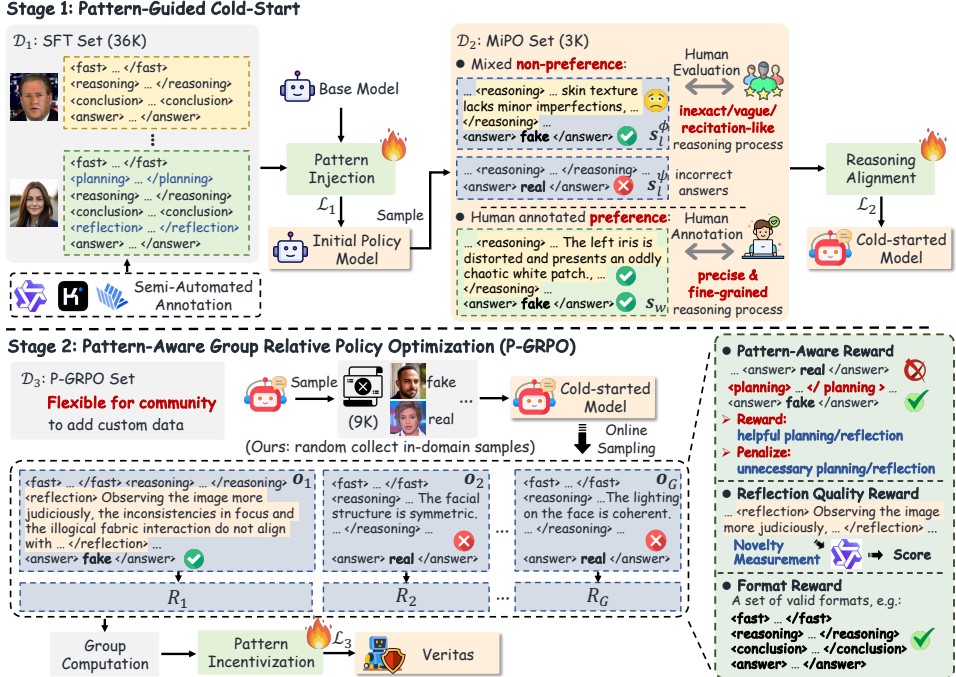

Figure 3: **Overview of two-stage training pipeline**. **(a)** For Pattern-Guided Cold-Start, we first employ SFT to internalize thinking patterns. Then we introduce MiPO to facilitate human-aligned reasoning. The MiPO dataset consists of mixed non-preference data, encouraging model to perform precise and fine-grained reasoning. **(b)** For Pattern-Aware GRPO, we introduce pattern-aware reward to incentivize adaptive reasoning ability on pattern granularity, yielding our VERITAS model.

by our human experts. Both $s_l^\phi$ and $s_l^\psi$ are sampled from the outputs of the SFT model, yielding 3K high-quality paired samples for dataset $\mathcal{D}_2$. Note that the images in $\mathcal{D}_2$ strictly come from the in-domain training set, without introducing any OOD samples. Suppose the SFT model is denoted as $\pi_{\theta_{\text{SFT}}}$, the training objective for MiPO is formulated as:

$$\mathcal{L}_2 = -\mathbb{E}_{(\boldsymbol{q},\boldsymbol{s}_w,\boldsymbol{s}_l)\sim\mathcal{D}_2}\left[\log\sigma\left(\beta\log\frac{\pi_\theta(\boldsymbol{s}_w|\boldsymbol{q})}{\pi_{\theta_{\text{SFT}}}(\boldsymbol{s}_w|\boldsymbol{q})} - \beta\log\frac{\pi_\theta(\boldsymbol{s}_l|\boldsymbol{q})}{\pi_{\theta_{\text{SFT}}}(\boldsymbol{s}_l|\boldsymbol{q})}\right)\right], \quad (2)$$

where $\sigma(\cdot)$ denotes the sigmoid function and $\beta$ controls the strength that the model deviates from the reference model. As shown in Figure 1, by learning from such mixed rejected traces, our model can perform more precise and fine-grained reasoning compared to pure SFT cold-start.

## 4.2 PATTERN-AWARE EXPLORATION

After cold-start, the trained model possesses the fundamental reasoning capacities for deepfake detection. However, it still fails on more challenging samples. To mitigate this, we introduce Pattern-Aware GRPO (P-GRPO) to encourage the model to perform comprehensive reasoning and potential self-reflection. Unlike recent approaches (Tu et al., 2025; Xiao et al., 2025) that encourage adaptive reasoning through length reward, we suppose that absolute reasoning length is not critical. Instead, we incentivize appropriate thinking patterns through the pattern-aware reward mechanism.

Suppose the training data for P-GRPO is $\mathcal{D}_3 = \{(\boldsymbol{q},\boldsymbol{a})_i\}_{i=1}^{N_3}$, where $\boldsymbol{a}$ denotes the binary answer. We randomly sampled 9K images from in-domain training set. For a given query $\boldsymbol{q}$, P-GRPO samples $G$ responses $\{o_1, o_2, ..., o_G\}$ using the current policy model $\pi_{\theta_{\text{old}}}$. The quality of each response $\{R_1, R_2, ..., R_G\}$ is evaluated through reward functions. Suppose the cold-started model $\pi_{\theta_{\text{cold}}}$ is adopted as reference policy, the training objective is formulated as:

$$\mathcal{L}_3 = -\mathbb{E}_{(\boldsymbol{q},\boldsymbol{a})\sim\mathcal{D}_3, \{\boldsymbol{o}_i\}_{i=1}^G\sim\pi_{\theta_{\text{old}}}(\cdot|\boldsymbol{q})}$$

$$\frac{1}{\sum_{i=1}^G|\boldsymbol{o}_i|}\sum_{i=1}^G\sum_{t=1}^{|\boldsymbol{o}_i|}\left[\min\left(r_{i,t}(\theta)A_{i,t}, \text{clip}(r_{i,t}(\theta), 1-\epsilon, 1+\epsilon)A_{i,t}\right) - \beta' D_{\text{KL}}[\pi_\theta\|\pi_{\theta_{\text{cold}}}]\right], \quad (3)$$

where

$$r_{i,t}(\theta) = \frac{\pi_\theta(o_{i,t} \mid \boldsymbol{I}, \boldsymbol{o}_{i,<t})}{\pi_{\theta_{\text{old}}}(o_{i,t} \mid \boldsymbol{I}, \boldsymbol{o}_{i,<t})}, \quad A_{i,t} = \frac{R_i - \text{mean}(\{R_1, \ldots, R_G\})}{\text{std}(\{R_1, \ldots, R_G\})}. \quad (4)$$

The reward $R_i$ for each response is evaluated from three perspectives:

**Pattern-aware Reward.** Suppose $\mathcal{C} \in \{0, 1\}$ represents the correctness of the final answer, with $\mathcal{C} = 1$ denoting the answer is right. $\mathcal{P} \in \{0, 1\}$ and $\mathcal{R} \in \{0, 1\}$ represents whether the reasoning involves "planning" and "self-reflection", respectively. The pattern-aware reward is defined as:

$$R_{\text{pattern}} = \begin{cases} 2.0, & \text{if } \mathcal{C}=1 \ \wedge (\mathcal{P}=1 \vee \mathcal{R}=1), \\ 1.0, & \text{if } \mathcal{C}=1 \ \wedge \mathcal{P}=0 \wedge \mathcal{R}=0, \\ 0.0, & \text{if } \mathcal{C}=0 \ \wedge \mathcal{P}=0 \wedge \mathcal{R}=0, \\ -0.5, & \text{if } \mathcal{C}=0 \ \wedge \mathcal{P}=1 \wedge \mathcal{R}=0, \\ -1.0, & \text{if } \mathcal{C}=0 \ \wedge \mathcal{R}=1. \end{cases} \quad (5)$$

Specifically, we encourage the model to reach correct answers through planning and self-reflection by assigning a larger reward (i.e., 2.0) if they are involved in the reasoning process. However, if these patterns lead to incorrect answers, we impose a penalty for its overthinking. Since self-reflection is a more decisive pattern, we assign a larger penalty (i.e., $-1.0$) for errors resulting from it.

**Reflection Quality and Format Reward.** To facilitate meaningful self-reflection, we assess the quality of reflection by an external model $\mathcal{M}$: $R_{\text{ref}} = \mathcal{M}(\boldsymbol{S})$. The criterion is the originality of the reflection, i.e., whether it introduces new perspectives rather than restating prior discoveries. The model only obtains $R_{\text{ref}}$ when the answer is correct. For format reward $R_{\text{fmt}}$, we predefine some combinations of reasoning patterns and set $R_{\text{fmt}} = 1$ when the response conforms to valid formats.

Suppose $\mathbb{I}(\cdot)$ is the indicator function. The final reward $R$ for each response is defined as:

$$R = R_{\text{pattern}} + \lambda_1 R_{\text{ref}} \cdot \mathbb{I}(\mathcal{C}=1) + \lambda_2 R_{\text{fmt}}. \quad (6)$$

In practice, given that only verifiable answers are required, the training data $\mathcal{D}_3$ can be freely expanded. Our cold-start model serves as a solid reasoning foundation, upon which the community can utilize custom data with P-GRPO to achieve more powerful reasoning model for deepfake detection.

## 5 EXPERIMENTS

### 5.1 EXPERIMENTAL SETUP

**State-of-the-Art Methods.** We trained 10 state-of-the-art (SOTA) detectors on our dataset, including F3Net (Qian et al., 2020), UniFD (Ojha et al., 2023), IID (Huang et al., 2023), FreqNet (Tan et al., 2024a), ProDet (Cheng et al., 2024), NPR (Tan et al., 2024b), AIDE (Yan et al., 2024a), Co-SPY (Cheng et al., 2025), $D^3$ (Yang et al., 2025b), Effort (Yan et al., 2024c). We also assess 4 open-source MLLMs of similar size to our model, including Qwen2.5-VL-7B (Bai et al., 2025), InternVL3-8B (Zhu et al., 2025), MiMo-VL-7B (Team, 2025) and GLM-4.1V-9B-Thinking (Hong et al., 2025), along with 2 powerful closed-source models GPT-4o (Hurst et al., 2024) and Gemini-2.5-Pro (Comanici et al., 2025). Besides, we evaluate recent MLLM-based forgery detectors, including FakeShield (Xu et al., 2024b), M2F2-Det (Guo et al., 2025b), SIDA (Huang et al., 2025a), FakeVLM (Wen et al., 2025c), FFAA (Huang et al., 2024). More details are in Appendix A.2.

**Metrics.** Following previous works (Zhang et al., 2024a; Guo et al., 2025b), we take Accuracy (Acc) to measure the model performance. Precision and Recall are reported in Appendix A.5.

**Implementation Details.** We implement **VERITAS** with InternVL3-8B (Zhu et al., 2025). For the cold-start SFT, we train the model for 3 epochs using LoRA (Hu et al., 2022) (rank=128, $\alpha$=256). The learning rate is set to $5 \times 10^{-5}$, with a batch size of 64. For cold-start MiPO, the model is trained for 2 epochs with the same setting of SFT. For P-GRPO, we further train the model for 2 epochs with the same LoRA setting. The learning rate is set to $1 \times 10^{-6}$ with a batch size of 16. $G$ is set to 4, with a temperature of 1.0. $\beta$ and $\beta'$ are set to 0. We take UnifiedReward-Qwen-3B (Wang et al., 2025d) as the reward model $\mathcal{M}$. For each stage, we directly adopt model from the last step.

Table 1: Performance comparison (Acc.) on HydraFake dataset. In-domain (ID) results are averaged. To ensure fair comparisons with MLLM-based detectors, 1) we exclude ID set in their average results and 2) further restrict the training scope of our method to FF++, StyleGAN, StableDiffusion XL and FFHQ (similar to FFAA), yielding "VERITAS-MINI". The best results are **bolded** and second best are underlined. More metrics in Appendix A.5.

| Method | ID | Cross-Model | | | | | | Cross-Forgery | | | | | | Cross-Domain | | | | | | Avg. |
|---|---|---|---|---|---|---|---|---|---|---|---|---|---|---|---|---|---|---|---|---|
| | | ADF | FLUX | StarryAI | MAGI-1 | HART | Infinity | St.GAN2 | ICLight | CodeF. | InfinityY. | PuLID | FaceAda. | Deepface. | InfinityY. | Dreamina | HailuoAI | GPT-4o | FFIW | |
| *Small Vision Models* | | | | | | | | | | | | | | | | | | | | |
| F3Net (*ECCV'20*) | 85.3 | 86.7 | 87.8 | 78.6 | 85.0 | 86.0 | 82.9 | 41.3 | 48.9 | 71.9 | 84.9 | 85.5 | 72.6 | 57.7 | 78.5 | 55.6 | 68.6 | 66.2 | 66.4 | 73.2 |
| UniFD (*CVPR'23*) | 82.7 | 90.7 | 93.8 | 82.5 | 73.0 | 94.4 | 90.7 | 61.8 | 81.9 | 75.4 | 73.7 | 68.1 | 81.3 | 67.4 | 67.3 | 80.5 | 75.2 | 73.3 | 67.5 | 78.0 |
| IID (*CVPR'23*) | 83.4 | 83.3 | 82.8 | 80.0 | 80.2 | 81.1 | 82.2 | 41.4 | 53.3 | 79.7 | 81.8 | 81.8 | 73.7 | 65.2 | 69.9 | 63.8 | 63.3 | 63.8 | 64.2 | 72.4 |
| FreqNet (*AAAI'24*) | 66.8 | 60.3 | 76.7 | 59.0 | 69.2 | 77.1 | 75.1 | 33.1 | 73.1 | 70.3 | 72.8 | 77.4 | 67.7 | 50.6 | 67.0 | 62.1 | 59.3 | 58.3 | 51.2 | 64.6 |
| ProDet (*NIPS'24*) | 90.5 | 92.6 | 94.2 | 88.2 | 91.9 | 93.8 | 93.1 | 56.3 | 58.6 | 80.8 | 88.1 | 91.0 | 83.3 | 58.1 | 82.9 | 71.3 | 75.6 | 66.3 | 74.1 | 80.6 |
| NPR (*CVPR'24*) | 75.6 | 68.8 | 91.2 | 59.5 | 82.6 | 91.3 | 84.0 | 47.7 | 67.8 | 60.6 | 79.8 | 89.0 | 67.7 | 52.6 | 73.0 | 76.6 | 62.3 | 50.2 | 46.0 | 69.8 |
| AIDE (*ICLR'25*) | 80.4 | 68.8 | 86.3 | 64.0 | 88.9 | 95.4 | 76.0 | 56.7 | 79.2 | 86.1 | 74.2 | 62.4 | 75.7 | 59.7 | 67.9 | 49.7 | 58.0 | 51.9 | 59.2 | 70.6 |
| Co-SPY (*CVPR'25*) | 86.3 | 93.5 | 95.5 | 85.3 | 93.3 | 96.6 | 95.3 | 77.0 | 92.5 | 88.6 | 90.6 | 79.1 | 87.3 | 67.6 | 80.0 | 82.5 | 74.0 | 79.5 | 64.3 | 84.7 |
| D³ (*CVPR'25*) | 87.3 | 93.6 | 95.6 | 91.3 | 90.7 | 95.8 | 95.5 | 62.4 | 71.6 | 82.9 | 80.0 | 82.4 | 73.7 | 69.7 | 74.6 | 78.1 | 70.9 | 80.8 | 64.3 | 81.1 |
| Effort (*ICML'25*) | 94.7 | 82.8 | 96.5 | 78.0 | 90.5 | 97.8 | 98.3 | 64.7 | **94.8** | 89.7 | 89.5 | 92.9 | 88.0 | 64.8 | 82.2 | 61.5 | 66.4 | 53.8 | 74.0 | 82.2 |
| *Generic MLLMs* | | | | | | | | | | | | | | | | | | | | |
| Qwen2.5-VL-7B | 51.2 | 50.0 | 50.0 | 49.7 | 50.0 | 52.0 | 52.9 | 50.5 | 56.7 | 50.7 | 53.6 | 54.5 | 51.6 | 50.7 | 53.6 | 80.2 | 67.5 | 52.5 | 50.5 | 54.1 |
| InternVL3-8B | 54.0 | 54.0 | 49.8 | 49.0 | 56.6 | 55.8 | 57.2 | 62.9 | 54.2 | 62.9 | 63.6 | 54.8 | 67.7 | 54.4 | 67.1 | 77.1 | 66.5 | 47.4 | 51.8 | 58.3 |
| MiMo-VL-7B | 63.8 | 74.5 | 77.1 | 82.5 | 60.3 | 82.4 | 81.4 | 48.7 | 82.6 | 76.4 | 79.7 | 78.4 | 82.8 | 57.7 | 75.6 | 79.4 | 70.7 | 67.7 | 54.9 | 72.5 |
| GLM-4.1V-9BThink | 56.4 | 55.2 | 52.3 | 50.5 | 51.6 | 68.4 | 60.7 | 54.3 | 68.4 | 63.3 | 65.7 | 55.1 | 81.0 | 58.7 | 72.7 | 83.7 | 69.2 | 52.0 | 53.9 | 61.7 |
| GPT-4o | 53.5 | 57.7 | 52.0 | 51.4 | 59.9 | 81.2 | 54.8 | 66.4 | 58.9 | 52.5 | 64.4 | 60.9 | 55.5 | 49.4 | 62.0 | 90.7 | 73.7 | 58.0 | 52.8 | 60.8 |
| Gemini-2.5-Pro | 72.2 | 64.9 | 92.4 | 82.8 | 62.5 | 93.4 | 93.2 | 73.7 | 83.3 | 87.4 | 85.5 | 84.7 | 85.6 | 67.2 | 75.6 | 87.5 | 82.4 | 70.9 | 53.0 | 78.9 |
| *MLLM-based Forgery Detectors* | | | | | | | | | | | | | | | | | | | | |
| M2F2-Det (*CVPR'25*) | - | 56.0 | 57.7 | 59.8 | 61.8 | 61.3 | 55.4 | 78.9 | 65.5 | 80.0 | 57.4 | 57.5 | 76.3 | 73.0 | 56.3 | 67.2 | 50.6 | 53.0 | 70.6 | 63.2 |
| FakeShield (*ICLR'25*) | - | 64.3 | 64.0 | 61.5 | 63.1 | 61.8 | 63.3 | 64.0 | 57.3 | 60.9 | 58.1 | 63.6 | 63.7 | 50.2 | 83.8 | 53.8 | 51.3 | 53.9 | 55.6 | 60.8 |
| SIDA-7B (*CVPR'25*) | - | 97.3 | 97.7 | 79.5 | 59.3 | 98.5 | 95.0 | 59.8 | 60.6 | 62.3 | 89.7 | 94.4 | 63.3 | 50.4 | 81.9 | 80.0 | 78.0 | 68.9 | 57.3 | 76.3 |
| SIDA-13B (*CVPR'25*) | - | 80.7 | 78.5 | 54.8 | 52.5 | 91.3 | 82.4 | 63.7 | 61.2 | 68.2 | 56.7 | 67.1 | 84.3 | 60.8 | 58.2 | 88.3 | 74.0 | 74.1 | 59.9 | 69.8 |
| FFAA (*Arxiv'24*) | - | 55.1 | 50.9 | 72.9 | 63.5 | 60.8 | 57.6 | 82.7 | 70.9 | 71.8 | 58.4 | 62.4 | 86.0 | 67.7 | 58.4 | 55.3 | 59.2 | 49.6 | 68.3 | 64.0 |
| FakeVLM (*NIPS'25*) | - | 78.2 | 78.5 | 77.0 | 74.5 | 76.5 | 76.8 | 70.8 | 76.2 | 76.2 | 76.9 | 76.5 | 77.7 | **75.7** | 83.6 | 81.5 | 80.8 | 78.7 | 74.5 | 77.3 |
| VERITAS-MINI | - | 95.5 | 99.1 | 97.3 | 72.8 | 97.0 | 96.1 | 82.5 | 76.3 | 90.0 | 83.7 | 82.9 | 79.3 | 72.5 | 78.7 | 92.0 | **93.0** | 85.5 | 70.6 | 85.8 |
| VERITAS (cold-start) | 96.8 | 79.5 | 99.6 | 96.0 | 99.9 | 99.7 | 99.9 | 84.0 | 65.3 | 94.8 | 86.2 | 93.4 | 86.7 | 55.9 | 73.5 | 93.7 | 89.3 | 88.1 | 76.4 | 87.3 |
| **VERITAS (ours)** | 97.3 | 94.8 | 99.8 | 97.0 | 99.9 | 99.9 | 99.9 | 90.3 | 75.7 | 97.0 | 91.8 | 95.1 | 91.7 | 58.6 | 84.1 | 92.3 | 90.2 | 89.2 | 78.5 | 90.7 |

Table 2: Effect of the proposed pattern-aware reasoning.

| Model | ID | CM | CF | CD |
|---|---|---|---|---|
| w/o Reasoning | **97.8** | 93.3 | 73.0 | 69.5 |
| Post-hoc Explanation | 96.3 | 95.0 | 79.0 | 76.8 |
| Flexible Reasoning | 96.2 | 94.3 | 81.2 | 76.8 |
| Pattern-aware Reason. | 96.9 | **98.4** | **87.4** | **80.1** |
| Δ Flexible Reason. | +0.7 | +4.1 | +6.2 | +3.3 |

Table 3: Ablations on the reward functions in P-GRPO.

| $R_{pattern}$ | $R_{ref}$ | $R_{fmt}$ | $R_{acc}$ | ID | CM | CF | CD |
|---|---|---|---|---|---|---|---|
| | | ✓ | ✓ | 97.2 | 96.4 | 86.3 | 79.9 |
| | ✓ | ✓ | ✓ | 97.0 | 97.3 | 87.0 | 80.7 |
| ✓ | | ✓ | | 97.3 | 97.7 | 87.9 | 81.4 |
| ✓ | ✓ | ✓ | | 97.3 | 98.6 | 90.3 | 82.2 |
| Δ GRPO | | | | +0.1 | +2.2 | +4.0 | +2.3 |

Table 4: Ablations on different base models and model sizes.

| Base Model | ID | CM | CF | CD |
|---|---|---|---|---|
| Qwen2.5-VL-7B | 96.8 | 97.7 | 89.0 | 81.4 |
| MiMo-VL-7B | 93.0 | 98.6 | 82.6 | **83.0** |
| InternVL3-2B | 97.4 | 97.4 | 87.3 | 80.4 |
| InternVL3-8B | 97.3 | 98.6 | 89.3 | 82.2 |
| InternVL3-14B | 98.5 | 99.3 | 92.2 | 82.6 |

## 5.2 MAIN RESULTS

**Comparison to SOTA detectors.** As shown in Table 1, our VERITAS model achieves SOTA performance on four evaluation scenarios, achieving 6.0% averaged gains over the previous best. Existing detectors show great performance on cross-model split (over 90% for D³) but fall short on cross-forgery and cross-domain scenarios (mostly less than 85%). VERITAS mitigates the gap, achieving over 90.0% accuracy on unseen forgery such as face restoration and personalization, and over 90.0% on in-the-wild data from Dreamina and 89.2% on GPT-4o. The cold-start model also achieves promising results, but without incentivizing planning and self-reflection, the cross-forgery results are degraded. More results and analyses can be found in Appendix A.5.

**Comparison to SOTA MLLMs.** Compared to our base model, VERITAS achieves 32.4% averaged gain, suggesting the effectiveness of our training strategy. The models with similar sizes show limited abilities for deepfake detection, with less than 60% accuracy. Gemini-2.5-Pro shows the best capacities among these MLLMs, even outperforming some of the fine-tuned detectors. VERITAS surpasses Gemini-2.5-Pro by 11.8%, demonstrating great generalization.

**Comparisons to MLLM-based detectors.** For fair comparisons, we restrict our training scope (Table 1). Even with limited data scope, VERITAS-MINI *still outperforms existing MLLM-based detectors*, indicating the effectiveness of the proposed framework. M2F2-Det and FFAA, though targeted at deepfake detection, suffer from poor generalization on HydraFake. SIDA-7B and FakeVLM achieve promising results by contrast. Moreover, VERITAS exhibits certain advantages in both detection accuracy and reasoning depth (Figure 6). More cases can be found in Appendix A.8.

Table 5: Effect of reasoning patterns. SFT and P-GRPO are performed for comparisons.

| Model | ID | CM | CF | CD | Avg. |
|---|---|---|---|---|---|
| Flexible Reasoning | 96.2 | 94.3 | 81.2 | 76.8 | 87.1 |
| Pattern-aware Reasoning | 96.9 | 98.4 | **87.4** | **80.1** | **90.7** |
| w/o <fast> | **97.3** | **98.8** | 86.9 | 79.1 | 90.5 |
| w/o <planning> | 96.7 | 96.9 | 85.0 | **80.1** | 89.7 |
| w/o <reflection> | 97.0 | 97.2 | 82.5 | 77.3 | 88.5 |
| w/o <conclusion> | 97.2 | 98.2 | 86.2 | 79.0 | 90.1 |

Table 6: Ablations on non-preference in MiPO.

| Model | ID | CM | CF | CD | Avg. |
|---|---|---|---|---|---|
| **VERITAS** | **97.3** | **98.6** | **90.3** | **82.2** | **92.1** |
| w/o MiPO | 96.9 | 98.4 | 87.4 | 80.1 | 90.7 |
| MiPO (w/o $s_l^\phi$) | 96.9 | 98.6 | 89.2 | 81.4 | 91.5 |
| MiPO (w/o $s_l^\psi$) | 65.3 | 64.8 | 58.6 | 54.3 | 60.8 |

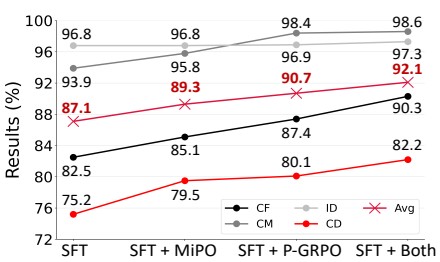

Figure 4: Ablations on the training stages. "Avg" is directly averaged across four splits.

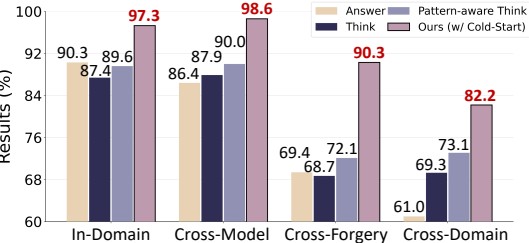

Figure 5: Effect of Cold-Start. We compare different settings of pure RL.

## 5.3 ABLATION STUDIES

We provide primary ablations in main text. More analyses on the training protocol (A.5.2), results on recent benchmark (Li et al., 2025)(A.5.2), selection of P-GRPO training data (A.5.4) and reward model (A.5.6), hyperparameters (A.5.5) and efficiency analysis (A.5.3) can be found in Appendix.

**Effect of pattern-aware reasoning.** As shown in Table 2, we compare different reasoning paradigms using SFT and P-GRPO training. Although the improvements on in-domain datasets are marginal, our pattern-aware reasoning demonstrates clear advantages to flexible reasoning on OOD scenarios, achieving 6.2% and 3.3% gains on CF and CD testing respectively. The post-hoc explanation adopted in recent methods exhibits degraded performance in OOD testing, further verifying the superiority of pattern-aware reasoning.

**Ablations on different training stages.** As shown in Figure 4, we investigate the effect of each training stage. Applying MiPO or P-GRPO upon SFT model both achieve significant gains, with P-GRPO performing better, which is due to the online sampling and pattern-aware incentivization. Applying MiPO before P-GRPO yields the best performance, achieving 2.9% and 2.1% gains on CF and CD testing respectively. This is because *MiPO ensures high-quality rollouts in subsequent stage*, facilitating *more accurate policy updates for online RL*.

**Effect of Pattern-guided Cold-Start.** As shown in Figure 5, we investigate different RL settings without cold-start. The training data keeps consistent with our two-stage pipeline. Answer-only model achieves better ID results while incorporating thinking improves CM and CD performance. However, all settings underperform the model with cold-start. The low-quality explorations lead to unstable training. Results in Figure 4 further verify the effectiveness of MiPO during cold-start.

**Effect of Pattern-aware GRPO.** As shown in Table 3, our P-GRPO achieves noticeable improvements compared to original GRPO. Specifically, pattern-aware reward outperforms the vanilla accuracy reward especially on CF and CD scenarios. The reflection quality reward benefits both original GRPO and our P-GRPO, which demonstrates the importance of high-quality self reflection. In Appendix A.5.4, we observe that by adding several "unseen" data in P-GRPO, the ODD performance can be further improved, demonstrating promising *scalability* with only binary labels required.

**Effect of specific reasoning patterns.** As shown in Table 5, "fast judgement" is helpful for CF and CD, but is not critical overall. "planning" is more effective on CM, since the fully synthesized images require a more holistic and structured analysis. "self-reflection" is critical especially on CF and CD, as it incentivizes the model to discover those unseen artifacts. "conclusion" provides certain gains, suggesting that synthesizing separate evidence into a coherent verdict is also important.

**Ablations on the non-preference in MiPO.** As shown in Table 6, $s_l^\phi$ helps improve the performance on CF (+1.3%) and CD (+0.8%) scenarios. To understand the effects, we provide a qualitative case in Figure 20. Without $s_\phi$, the model still gets correct answers, but the analysis is superficial and less detailed, which causes certain failures on unseen forgeries that might require in-depth reasoning.

Table 7: Evaluation of reasoning quality. We utilize score and pairwise ELO rating.

Table 8: Robustness on Compression, Blur and Resize. The results are averaged across all sets.

| Model | Score Evaluation | | ELO Rating |
|---|---|---|---|
| | GPT-4o | Gemini-2.5-Pro | |
| MiMo-VL-7B | 3.0731 | 2.5785 | 695.0 |
| GPT-4o | 2.4718 | 2.1619 | 785.1 |
| Gemini-2.5-Pro | 4.1681 | 4.0070 | 966.9 |
| VERITAS (w/o MiPO) | 4.2538 | 4.1502 | 984.0 |
| VERITAS (w/ DPO) | 4.5077 | 4.2863 | 1210.0 |
| **VERITAS (w/ MiPO)** | **4.6479** | **4.4214** | **1359.0** |

| Method | Original | JPEG Compression | | | Gaussian Blur | |
|---|---|---|---|---|---|---|
| | | QF=90 | QF=70 | QF=50 | $\sigma = 1.0$ | $\sigma = 2.0$ |
| UniFD | 78.0 | 77.1 | 75.4 | 72.2 | 76.5 | 75.4 |
| IID | 72.4 | 71.4 | 70.1 | 69.7 | 72.1 | 71.9 |
| ProDet | 80.6 | 80.1 | 79.4 | 78.7 | 79.7 | 77.8 |
| Co-SPY | 84.7 | 84.5 | 83.4 | 81.0 | 82.7 | 77.0 |
| Effort | 82.2 | 73.1 | 79.6 | 66.3 | 79.1 | 70.4 |
| VERITAS | 90.7 | 90.1 | 88.7 | 87.4 | 88.8 | 84.3 |

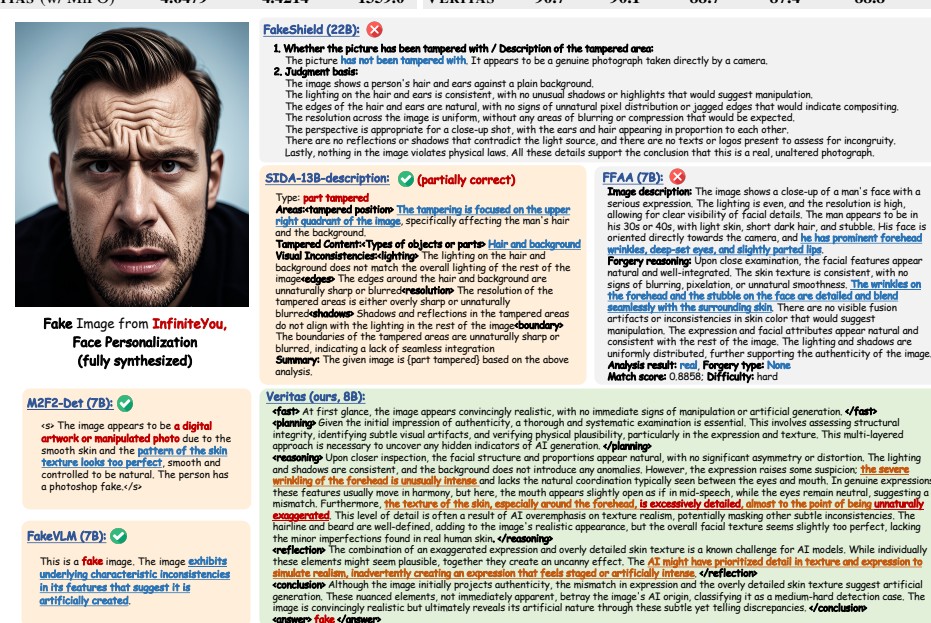

Figure 6: Reasoning comparisons between VERITAS and existing MLLM-based detectors.

## 5.4 FURTHER ANALYSES

**Evaluation of reasoning quality.** To evaluate the reasoning quality, we take two types of assessments: (1) score evaluation which is based on predefined criteria (Figure 33). (2) Pairwise comparison which directly compares outputs from two models. We adopt MLLM-as-a-Judge (Chen et al., 2024a), using GPT-4o and Gemini-2.5-Pro for evaluation. Similar to (Zhou et al., 2025), we randomly select 1K samples for evaluation. As shown in Table 7, our model achieves the best score and ELO rating, where MiPO greatly improves the reasoning quality. Moreover, our MiPO outperforms DPO in raising reasoning quality, which verifies the effectiveness of mixed non-preference strategy.

**Different fine-tuned base models and model sizes.** As shown in Table 4, we adopt different MLLMs as our base model. InternVL3-8B outperforms Qwen2.5-VL-7B and MiMo-VL-7B, due to the dynamic high resolution strategy. InternVL3-2B achieves promising performance with fewer parameters, while scaling up to 14B yields considerable gains on CM and CF scenarios.

**Robustness evaluation.** We investigate the performance under JPEG compression and Gaussian blur. Results in Table 8 highlight the robustness of our model. Our model achieves consistently high performance under JPEG compression and maintains state-of-the-art results across different perturbations. Notably, this robustness is achieved without training on corresponding data augmentations such as random Gaussian blur, which instead are commonly adopted in previous methods.

## 6 CONCLUSION

In this paper, we introduce HydraFake dataset and VERITAS model. HydraFake introduces a holistic evaluation protocol to comprehensively measure the generalization capacities. We then train a multi-modal large language model (MLLM) based deepfake detector trained with our two-stage pipeline. Results on HydraFake show that current detectors struggle on cross-forgery and cross-domain scenarios, while our model greatly mitigates the gap and is capable of delivering transparent decision process. We hope this work can inspire more generalizable and reliable deepfake detection.

ACKNOWLEDGMENTS

This work was supported by the Beijing Natural Science Foundation JQ23016, the Chinese National Natural Science Foundation Projects 62476273, 62406320, 62276254 and U23B2054, the Science and Technology Development Fund of Macau Project 0123/2022/A3, 0140/2024/AGJ, 0044/2024/AGJ and 0084/2024/RIB2, and Ant Group.

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

# A  APPENDIX

The appendix is organized as follows:

- §A.1 Details of HydraFake Dataset.

    - §A.1.1 Training Set.
    - §A.1.2 In-Domain Evaluation.
    - §A.1.3 Cross-Model Evaluation.
    - §A.1.4 Cross-Forgery Evaluation.
    - §A.1.5 Cross-Model Evaluation.

- §A.2 More Implementation Details.

- §A.3 More Discussions.

- §A.4 Multi-Step Annotation Pipeline.

- §A.5 More Experimental Results.

    - §A.5.1 More Results and Analyses on HydraFake.
    - §A.5.2 Cross Benchmark Comparison.
    - §A.5.3 Efficiency Comparison.
    - §A.5.4 Effect of Training Data in P-GRPO Stage.
    - §A.5.5 Analysis of Hyperparameters.
    - §A.5.6 Effect of Different Reward Model.

- §A.6 Full Prompt Templates.

- §A.7 More Qualitative Results.

- §A.8 More Qualitative Comparisons with Existing MLLM-based Detectors.

- §A.9 Failure Analysis of VERITAS.

- §A.11 Ethics Statement.

- §A.12 Limitations and Future Work.

## A.1  DETAILS OF HYDRAFAKE DATASET

In this section, we provide more details about our HydraFake Dataset. We introduce the dataset from the perspective of training and evaluation protocols.

### A.1.1  TRAINING

**Real Images.** HydraFake dataset contains real images from 8 public datasets. We extract 5 subsets as the training set, containing 3 low-quality subsets and 2 high-quality subsets. The low-quality images include FF++ (Rossler et al., 2019), CelebA (Liu et al., 2015) and LFW (Huang et al., 2008). The high-quality images include FFHQ (Karras et al., 2019) and CelebAHQ (Karras et al., 2017). This results in 24K real images for training.

**Fake Images.** In practical scenario, there are abundant fake images for training, while these images have two attributes: (1) the quality of the images varies greatly, and (2) the forgery types are often limited. To mimic such setting, we extract 21 subsets as the training set and strictly control the seen forgeries. We only include face swapping (FS), face reenactment (FR) and entire face generation (EFG) in our training set, leaving various forgery types unseen. Moreover, the deepfake methods in our training set are not the latest, leaving fresh methods in the evaluation.

- **FS**: FF++, BlendFace, FSGAN, SimSwap, FaceDancer, MobileSwap.

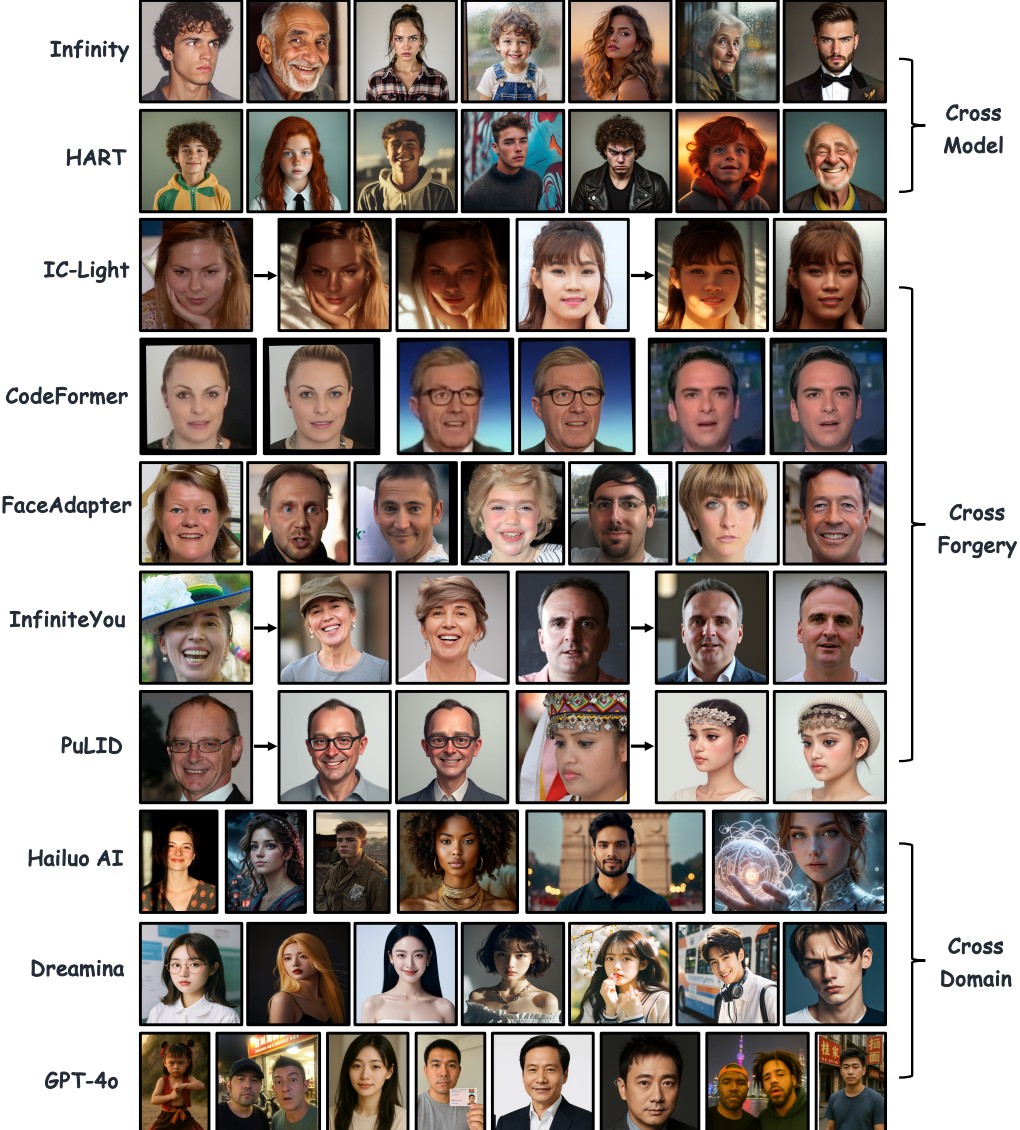

Figure 7: Examples of our self-constructed subsets in HydraFake datasets.

- **FR**: FF++, Facevid2vid, Hallo, Hallo2, LivePortrait, AniPortrait, EmoPortrait

- **EFG**: Dall-e 1, StyleGAN, StyleGAN2, VQGAN, Midjourney, Seeprettyface, Stable Cascade, Stable Diffusion XL, Attend-and-Excite.

### A.1.2 IN-DOMAIN EVALUATION

For in-domain testing, we select 5 subsets from the training set, and use the unseen identities as the testing samples. Specifically, we make a balance on the image quality and forgery types, choosing a FS dataset FF++ (low-quality), two FR datasets Facevid2vid (low-quality) and Hallo2 (high-quality), and two EFG datasets StyleGAN (Karras et al., 2019) (high-quality) and Midjourney (high-quality). For low-quality subsets, the real images are sampled from FF++. For high-quality subsets, the real images are sampled from FFHQ.

### A.1.3  CROSS-MODEL EVALUATION

The cross-model testing images come from deepfakes generated using unseen models. While the difficulty varies according to the model architectures. This includes 6 subsets, i.e., Adobe Firefly (Adobe, 2023), MAGI-1 (Sand-AI, 2025), Flux1.1 Pro (Black Forest Labs, 2024), StarryAI (AI, 2023), Infinity (Han et al., 2025) and HART (Tang et al., 2024).

**Infinity.** Infinity is a Bitwise Visual AutoRegressive Model (VAR) capable of generating high-resolution and photorealistic images from textual prompts. Infinity redefines visual autoregressive model under a bitwise token prediction framework with an infinite-vocabulary tokenizer and bitwise self-correction. We reproduce the Infinity-8B model, which is capable of generating $1024 \times 1024$ images. The control prompts are generated using Qwen3-32B model, which leverages a template-like sentence, balances between gender and age and avoids any semantic conflicts (e.g., wrinkles on the little girl's face) with the help of LLM.

**HART.** Hybrid Autoregressive Transformer (HART) is an autoregressive (AR) visual generation model capable of directly generating $1024 \times 1024$ images, rivaling diffusion models in image generation quality. It contains a hybrid image tokenizer to improve the fidelity of the generated images. We reproduce HART model, which is based on Qwen2-VL-1.5B. The textual prompts are consistent with Infinity.

**Adobe Firefly.** Adobe Firefly is a proprietary suite of multimodal generative AI models developed by Adobe. It is built upon a deeply customized and optimized diffusion model. It exclusively utilizes the Adobe Stock library, open-licensed content, and public domain works, thereby designed to ensure the commercial viability and mitigate copyright risks of its generated outputs. We collect this subset from WILD (Bongini et al., 2025), which is generated using template-like textual prompts.

**StarryAI.** Starry AI is an advanced generative artificial intelligence model engineered for high-fidelity text-to-image synthesis. Its core architecture integrates a transformer-based encoder for the semantic interpretation of textual prompts with a latent diffusion model for the iterative synthesis of visual content. The model excels at translating complex, abstract descriptions into visually coherent and stylistically nuanced imagery. We collect this subset from WILD (Bongini et al., 2025), which is generated using template-like textual prompts.

**MAGI-1.** MAGI-1 is an autoregressive denoising video generation model (Video AR) generating videos chunk-by-chunk instead of as a whole. It excels in generating high-quality, temporally consistent videos from text or image prompts. With support for large-scale model sizes and long context lengths, it is well-suited for a wide range of creative and generative video applications. We collect this subset from TalkingHeadBench (Bongini et al., 2025).

**Flux1.1 Pro.** Flux1.1 Pro is currently the most advanced model of Flux series, which is introduced by Black Forest Labs. It is designed for fast, high-resolution and realistic text-to-image generation. We collect this subset from WILD (Bongini et al., 2025), which is generated using template-like textual prompts.

### A.1.4  CROSS-FORGERY EVALUATION

The cross-forgery testing images come from deepfakes generated by unseen forgeries. With the rapid development of generative techniques, novel types of forgery are constantly emerging, such as portrait relighting (Zhang et al., 2025b) and IP-preserved personalization (Jiang et al., 2025; Guo et al., 2024). To assess the model's generalization capacities when encountering these emerging deepfake methods, we collect 5 representative forgery methods in our dataset, including face relighting (Zhang et al., 2025b), face restoration (Zhou et al., 2022), generative face swapping (Han et al., 2024), facial attribute editing (Choi et al., 2020) and face personalization (Jiang et al., 2025; Guo et al., 2024).

**Face Relighting.** The method is based on IC-Light (Zhang et al., 2025b), which is an emerging ability in the generative models and is becoming prevailing. "IC-Light" means "Imposing Consistent Light", which is capable of adjusting the lighting sources and intensity in the image while keeping the subject highly unchanged. The condition is based on textual prompts. We sampled real images from FFHQ (Karras et al., 2019) and reproduced IC-Light to change the lighting condition of these real images. We implemented 10 lighting types (e.g., "sunshine from window", "soft studio lighting"

Table 9: Data list of the hierarchical evaluation protocol in HydraFake dataset.

| Evaluation Split | Method | Sub-Type | Venue | Data Scale | Resolution |
|---|---|---|---|---|---|
| **In-Domain** | FaceForensics++ | FS | ICCV'19 | 8,960 | $256 \times 256$ |
| | Facevid2vid | FR | Arxiv'19 | 2,000 | $256 \times 256$ |
| | Hallo2 | FR | ICLR'25 | 1,660 | $256 \times 256$ |
| | StyleGAN | EFG | CVPR'19 | 600 | $1024 \times 1024$ |
| | Midjourney | EFG | None | 600 | $1024 \times 1024$ |
| **Cross-Model** | Adobe Firefly | Proprietary | None | 600 | $1024 \times 1024$ |
| | StarryAI | Proprietary | None | 600 | $1024 \times 1024$ |
| | Flux1.1 Pro | Customized | None | 600 | $1024 \times 1024$ |
| | MAGI-1 | Video AR | None | 1,048 | $256 \times 256$ |
| | HART | Image AR | Arxiv'24 | 4,200 | $1024 \times 1024$ |
| | Infinity | Image AR | CVPR'25 | 4,200 | $1024 \times 1024$ |
| **Cross-Forgery** | StarGANv2 | Editing | CVPR'20 | 2,000 | $256 \times 256$ |
| | CodeFormer | Restoration | NIPS'22 | 1,750 | $512 \times 512$ |
| | IC-Light | Relighting | ICLR'25 | 2,082 | $1536 \times 1536$ |
| | FaceAdapter | Generative FS | ECCV'24 | 300 | $1024 \times 1024$ |
| | PuLID | Personalization | NIPS'24 | 3,360 | $1024 \times 1024$ |
| | InfiniteYou | Personalization | ICCV'25 | 3,244 | $1024 \times 1024$ |
| **Cross-Domain** | Hailuo AI | Commercial | None | 1,000 | $256 \sim 1536$ |
| | Dreamina | Social media | None | 952 | $1024 \times 1024$ |
| | GPT-4o | Social media | None | 630 | $159 \sim 1536$ |
| | DeepFaceLab | Classic dataset | PR 2023 | 3,094 | $256 \times 256$ |
| | FFIW | Classic dataset | CVPR'21 | 6,832 | $256 \times 256$ |
| | InfiniteYou-CD | Personalization | ICCV'25 | 2,960 | $1024 \times 1024$ |

and "neon light in city") and 4 lighting sources (i.e., "left", "right", "top" and "bottom"). We use multiple seeds for each condition, and then manually filter out those of low quality.

**Face Restoration.** The method is based on CodeFormer (Zhou et al., 2022), which can recover low-quality (e.g., blurred) natural faces to high-quality counterparts, even when the inputs are severely degraded. It can generate high-quality faces while maintaining the fidelity. In fact, this is a helpful technique that has positive usage in many domains. But considering this can be also used for low-quality deepfake images, we take this as an unseen forgery in our dataset. Specifically, we sampled some low-quality fake images from DF40 (Yan et al., 2024d) and TalkingHeadBench (Xiong et al., 2025), and then employ CodeFormer to restore them into $512 \times 512$ images.

**Facial Attribute Editing.** Facial attribute editing is a common manipulation, involving altering facial attributes such as hairstyle and makeup. In our dataset we leave this type out for testing. We collect images generated by StarGANv2 (Choi et al., 2020) from DF40 (Yan et al., 2024d).

**IP-preserved Face Personalization.** IP-preserved face personalization technology enables the generation of synthetic faces that closely retain the distinctive visual attributes of original intellectual property (IP). By producing highly realistic and IP-consistent deepfakes, it can facilitate unauthorized exploitation or impersonation of protected characters and personalities. With the advancement of generative models, face personalization techniques are now capable of maintaining high-fidelity while following complex contextual and subject-specific instructions (e.g., transforming an ID photo into an image of the singer in the bar). We reproduce two timely methods PuLID (Guo et al., 2024) and InfiniteYou (Jiang et al., 2025). We sample real images from FFHQ as the source images. To enhance the realism and *semantic coherence* of face personalization, we employ Qwen2.5-VL-72B to generate customized prompts for each image.

**Generative Face Swapping.** The face swapping data in existing datasets are often produced by conventional approaches such as graphics-based methods or GAN models. While nowadays the generative-based methods are capable of generating high-fidelity swapped faces, which are based on Diffusion models. Considering the latest methods such as DreamID (Ye et al., 2025a) and DynamicFace (Wang et al., 2025c) are not open-sourced yet, we implemented FaceAdapter (Han et al., 2024), which produces high-quality swapping data. We will keep tracking the advancements in these methods and update our dataset. The source faces are sampled from FFHQ. We manually filter out those low-quality generated images, only maintaining high-fidelity samples.

### A.1.5 CROSS-DOMAIN EVALUATION

The "domain" in our dataset mainly refers to *data source*. For instance, the cross-forgery data are generated using in-domain real images from FFHQ, which alters the manipulation methods but keeps the data source unchanged. But for cross-domain testing, the fake images are either generated from unseen real data srouce or entirely generated by commercial models. And we also crawled fake images from social media, which serves as a challenging cross-domain evaluation. Specifically, our cross-domain testing can be clustered into three types: (1) classic datasets, including DeepFaceLab from DF40 (Yan et al., 2024d) and FFIW (Dolhansky et al., 2019) which is widely adopted in existing benchmarks. (2) Reproduced deepfakes, including face personalization generated using real images from VFHQ (Xie et al., 2022). (3) In-the-wild deepfakes, where we collect data from social media such as Xiaohongshu and TikTok. We retrieved images through the tags of the posts and collected the images generated by GPT-4o (Hurst et al., 2024) and Dreamina (team, 2025a), and cropped out the digital watermarks. We further generate deepfake videos using Hailuo AI (team, 2025b), and extract 8 frames for each video.

---

**Valid Output Formats in P-GRPO**

Format 1 (Basic):
```
<fast> ... </fast>
<reasoning> ... <reasoning>
<conclusion> ... <conclusion>
<answer> ... <answer>
```

Format 2 (With **Planning**):
```
<fast> ... </fast>
<planning> ... <planning>
<reasoning> ... <reasoning>
<conclusion> ... <conclusion>
<answer> ... <answer>
```

Format 3 (With **Self-Reflection**):
```
<fast> ... </fast>
<reasoning> ... <reasoning>
<reflection> ... <reflection>
<conclusion> ... <conclusion>
<answer> ... <answer>
```

Format 4 (With **Planning** and **Self-Reflection**):
```
<fast> ... </fast>
<planning> ... <planning>
<reasoning> ... <reasoning>
<reflection> ... <reflection>
<conclusion> ... <conclusion>
<answer> ... <answer>
```

---

Figure 8: Valid output formats in $R_{fmt}$ of P-GRPO.

### A.2 MORE IMPLEMENTATION DETAILS

**Training resources.** Our model is trained with 8 PPUE GPUs based on ms-swift (Zhao et al., 2024). The theoretical peak computational capacity (TFLOPS) of one PPUE GPU is roughly half of an NVIDIA A100 GPU, and each PPUE GPU has 96GB VRAM. With such infrastructure, the SFT and MiPO stage take 5.5 hours and 2 hours, respectively. The P-GRPO stage takes 11 hours on 9K training samples. All the inferences are conducted on a single PPUE GPU.

**Training details of previous methods.** For previous SOTA methods, we reproduce them based on DeepfakeBench (Yan et al., 2023). For F3Net (Qian et al., 2020), UniFD (Ojha et al., 2023),

IID (Huang et al., 2023), FreqNet (Tan et al., 2024a), ProDet (Cheng et al., 2024), NPR (Tan et al., 2024b) and Effort (Yan et al., 2024c), we reproduce them based on DeepfakeBench (Yan et al., 2023). For AIDE (Yan et al., 2024a), Co-SPY (Cheng et al., 2025) and $D^3$ (Yang et al., 2025b), we train the model with official codes and perform inference using DeepfakeBench. The images are first randomly cropped into $256 \times 256$ and then resized to $224 \times 224$. For Co-SPY (Cheng et al., 2025), the images are resized to $384 \times 384$ following the official implementation. We apply a series of data augmentations during training, including random flipping, rotation, gaussian blur, brightness and contrast alternation, color jitter and JPEG compression. Following official guides, AIDE and $D^3$ are trained for 100 epochs. FreqNet and NPR are trained for 50 epochs. The first stage of Co-SPY (i.e., artifacts and semantic encoders) are trained for 20 epochs, and the second stage (i.e., combination) is trained for another 10 epochs. Effort is trained for 10 epochs and other methods are trained for 20 epochs. During testing, the images are resized to $224 \times 224$. For all these methods, we curate a validation set containing 4K in-domain images for model selection.

**Other details.** The training data from all stages are strictly sampled from HydraFake training set. The 36K SFT data are randomly sampled and balanced across forgery types. The 3K MiPO pairs are selected based on SFT models' outputs. 800 images that the SFT model fails to reach all correct answers under 8 rollouts are selected. Each image is paired with 4 manually selected non-preference chains (from SFT model's outputs) and 1 manually annotated preference chain, resulting in 3K samples for MiPO. The 9K P-GRPO data are randomly sampled and balanced across forgery types. For open-sourced MLLMs, we provide prior knowledge and instruct the model to perform thinking in the prompts. The full prompts are provided in Figure 38, Figure 39 and Figure 40. For Gemini-2.5-Pro, we enable thinking and searching. $\lambda_1$ and $\lambda_2$ are set to 1.0 and 0.25, respectively. The valid output formats for $R_{fmt}$ in P-GRPO are listed in Figure 8.

## A.3 MORE DISCUSSIONS

**Difference between explainable and reasoning deepfake detection.** In this part, we formulate different task settings of MLLM-based deepfake detection. Given the input image $\boldsymbol{I}$, deepfake detection aims to determine its authenticity $\mathcal{Y} \in \{0, 1\}$, where 1 means the image is fake and vice versa. Suppose the input image and query are collectively denoted as $\boldsymbol{q}$. The sequential outputs of MLLM are denoted as $\boldsymbol{s} = \{\boldsymbol{s}_1, \boldsymbol{s}_2, ..., \boldsymbol{s}_T\}$, where $T$ is sequence length. The conditional probability of sequence $\boldsymbol{s}$ is written as $P(\boldsymbol{s}|\boldsymbol{q}) = \prod_{t=1}^{T} P(\boldsymbol{s}_t|\boldsymbol{q}, \boldsymbol{s}_{<t})$.

Recent works (Chen et al., 2024b; He et al., 2025) utilize MLLM for explainable deepfake detection, where LLM first generates the answer $\boldsymbol{s}_{\mathcal{A}}$ (e.g., "fake" or "this image is fake"). Then a detailed explanation sequence $\boldsymbol{s}_{\mathcal{E}}$ is generated based on $\{\boldsymbol{q}, \boldsymbol{s}_{\mathcal{A}}\}$ (where $\mathcal{A} \ll \mathcal{E}$). For simplicity, we suppose the answer $\boldsymbol{s}_{\mathcal{A}}$ is in a single word. The process can be decomposed into:

$$P(\boldsymbol{s}|\boldsymbol{q}) = \prod_{t=1}^{\mathcal{A}} P(\boldsymbol{s}_t|\boldsymbol{q}) \cdot \prod_{t=\mathcal{A}+1}^{\mathcal{A}+\mathcal{E}} P(\boldsymbol{s}_t|\boldsymbol{q}, \boldsymbol{s}_{<\mathcal{A}+\mathcal{E}}), \tag{7}$$

where the final decision (i.e., $\prod_{t=1}^{\mathcal{A}} P(\boldsymbol{s}_t|\boldsymbol{q})$) is solely conditioned on the input image. This is fundamentally similar to the small vision models, where the distributional mapping $f : \boldsymbol{I} \to \mathcal{Y}$ is estimated directly within a single token, prone to overfitting.

In contrast, we formulate the deepfake detection as a reasoning task. The MLLM first conduct holistic reasoning, denoted as $\boldsymbol{s}_{\mathcal{R}}$, and the final answer is then determined in $\boldsymbol{s}_{\mathcal{A}}$:

$$P(\boldsymbol{s}|\boldsymbol{q}) = \prod_{t=1}^{\mathcal{R}} P(\boldsymbol{s}_t|\boldsymbol{q}, \boldsymbol{s}_{<\mathcal{R}}) \cdot \prod_{t=\mathcal{R}+1}^{\mathcal{R}+\mathcal{A}} P(\boldsymbol{s}_t|\boldsymbol{q}, \boldsymbol{s}_{<\mathcal{R}+\mathcal{A}}), \tag{8}$$

where the final answer (i.e., $\prod_{t=\mathcal{R}+1}^{\mathcal{R}+\mathcal{A}} P(\boldsymbol{s}_t|\boldsymbol{q}, \boldsymbol{s}_{<\mathcal{R}+\mathcal{A}})$) is building on inputs and reasoning process. The mapping is altered into $f' : \boldsymbol{I} \to \mathcal{R} \to \mathcal{Y}$, enabling more comprehensive and adaptive modeling.

**Large Reasoning Models.** Large Language Models (LLMs) are inherently good reasoners for general tasks. A simple prompt engineering could activate the reasoning behaviors of LLMs (Wei et al., 2022; Kojima et al., 2022), which is termed as Chain-of-Thought (CoT). Building on the impressive capabilities of CoT, the community began to explore large-scale and structured reasoning, leading to powerful reasoning models (Jaech et al., 2024; Guo et al., 2025a; Xu et al., 2024a) through tailored

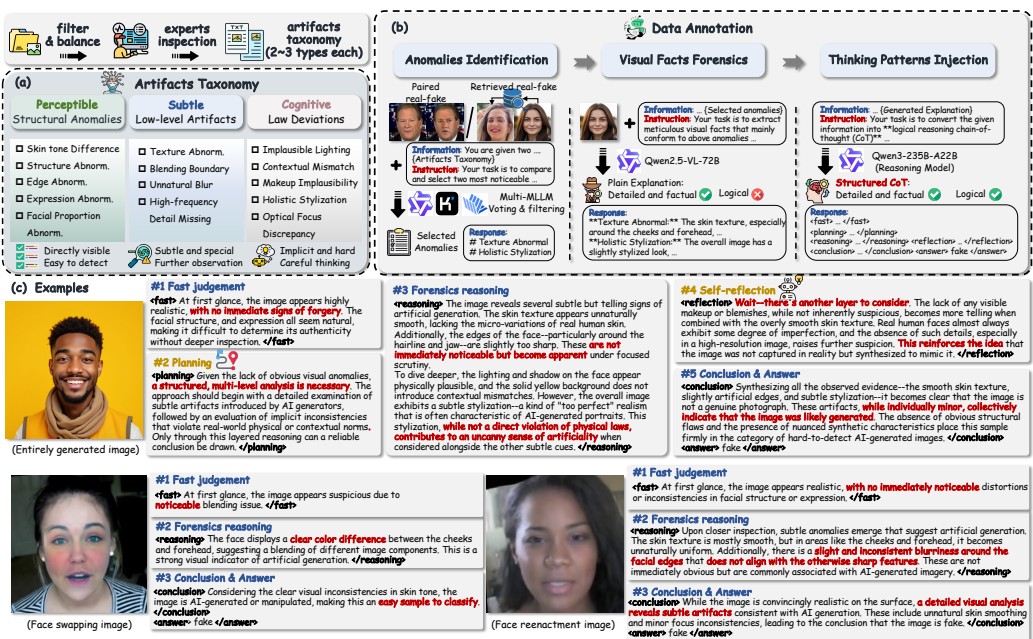

Figure 9: **Construction pipeline of Pattern-Aware SFT data**. **(a)** We first inspect a subset and summarize the artifacts into three clusters. **(b)** Then we introduce a multi-step strategy to generate pattern-aware reasoning data. **(c)** Annotated examples. The reasoning process evolves in complexity and depth (as highlighted in **red**), culminating in a final answer through synthesis of all evidence.

post-training. For instance, DeepSeek-R1 (Guo et al., 2025a) adopts reasoning cold-start followed by Reinforcement Learning with Verifiable Rewards (RLVR) to incentivize the general reasoning capabilities. Inspired by the success of RLVR, recent works (Liu et al., 2025; Zhang et al., 2025a) attempt to introduce pure rule-based RL into multimodal domain. However, a recent study (Yue et al., 2025) points out that pure RLVR can not introduce novel abilities to base model. We also empirically reveal the suboptimal performance achieved by pure RL. Therefore, we first introduce a high-quality cold-start (Liao et al., 2025; Chen et al., 2025a; Team et al., 2025) to internalize the thinking patterns to the base model. Unlike general tasks that require diverse and flexible thinking patterns (Zhan et al., 2025), deepfake detection is a well-defined task. Therefore, we establish a unified reasoning framework to facilitate efficient thinking.

**Rationale for reasoning in deepfake detection.** A possible concern is that even humans can fail on some highly realistic deepfakes. Applying reasoning for deepfake detection also faces the same problem. However, we point out that there is a physiological limit on human perception. The **human** excels at high-level semantic reasoning, such as judging contextual plausibility, but is less equipped to detect subtle, low-level digital artifacts like subtle blurriness or unnatural texture patterns. In contrast, the **machine** can be trained to perceive these subtle artifacts with superhuman accuracy. The primary challenge, which traditional detectors face, is not perception but generalization, i.e., they tend to overfit to specific artifact patterns. This is where the pattern-aware reasoning becomes crucial. Our goal is not to mimic a human's intuitive guess, but to emulate a forensic expert's systematic investigation. Our approach uniquely **combines** *the machine's superhuman perception* with a *structured and human-like reasoning framework* (e.g., planning, reasoning, self-reflection, conclusion). As illustrated in Figure 21, the model can perceive subtle artifacts (e.g., barely noticeable blurriness and faint texture anomalies) that are nearly imperceptible to the human. Therefore, reasoning is crucial not to replicate the fallible human eye, but to provide a logical structure that effectively leverages the model's perceptual abilities for robust generalization.

## A.4 Multi-Step Annotation Pipeline of SFT data

As shown in Figure 9, for **fake images** we divide the annotation process into three steps:

**Step-I**: Based on human inspection, we found that most fake images exhibit 2 to 3 types of artifacts, hence we aim to find the two most prominent artifacts. To reduce model bias, we employ an ensemble voting strategy, leveraging Qwen2.5-VL-72B (Bai et al., 2025), Kimi-VL-A3B-Thinking (Team et al., 2025) and InternVL3-78B (Zhu et al., 2025) to sample 5 times individually. Only the answers that receive more than 10 votes are selected, ensuring the reliability of the final results.

**Step-II**: In this stage, we aim to extract visual details that conform to the identified artifacts. We found that Qwen2.5-VL-72B performs better than GPT-4o (Hurst et al., 2024) here, capable of generating detailed and factual responses. This yields concrete explanatory texts, like recent practice (Wen et al., 2025c; Gao et al., 2025; Zhang et al., 2025c). However, such *plain* explanations lack human-like reasoning logic, which hinders the generalization to OOD samples.

**Step-III**: To emulate human mindset, we further transform the above explanations into logical chains. We define five "thinking patterns" and instruct the model to rewrite the explanations into different tags strictly based on the original meaning. We observed that large reasoning models are inherently adept at generating highly logical content. Therefore, we use Qwen3-235B-A22B (Yang et al., 2025a) for this step, yielding high-quality reasoning data. Finally, the data undergo a filtering process, which involves rule-based filtering and balancing among different forgery types.

For **real images**, we only use the last two steps (without the need for anomalies detection). For Step-I we provide the full artifacts list to Qwen2.5-VL-72B for comprehensive visual facts forensics. For Step-II we adopt Qwen3-235B-A22B to convert the explanation texts into pattern-aware reasoning chain. Specifically, for some low-quality real images, which may contain some misleading artifacts like unexpected blurriness or missing visual details, we instruct the model to point this out and put them in the "self-reflection" content. However, the model is not capable of directly perceive such minor artifacts especially when told the image is authentic. Hence, we provide a rough difficulty information based on dataset level and encourage the model to perform self-reflection on those difficult images. This can mitigate the problem while it can not be fully addressed due to the significant loss of details in those low-resolution images.

## A.5 More Experimental Results

### A.5.1 More Results and Analyses on HydraFake

We provide full experimental results for all subsets. To help better understanding the model performance, we report Precision and Recall for each subset, with fake being the positive label.

The in-domain results are shown in Table 10. Effort achieves the best results among previous detectors. It is worth noting that, under the mixed training sources, previous methods even struggle on in-domain datasets, i.e., most methods achieve less than 90% average performance. This is mainly due to the degraded performance on low-resolution datasets such as FF++ and Facevid2vid. We suppose this is a major deficiency in current deepfake detectors, which tend to bias towards image resolutions. Our model achieves better performance, achieving over 99.5% on those high resolution images, while the results on low resolution subsets still have room for improvement.

The cross-model results are shown in Table 11. $D^3$ achieves the best results among previous detectors. Lots of previous methods achieve good performance on cross-model scenarios, with averaged performance greater than 90%, such as $D^3$, ProDet, Co-SPY and Effort. These models show great performance on cross-model data, especially on VAR architectures, achieving over 95% accuracy. Our model demonstrates extraordinary generalization performance on cross-model data, achieving almost 99% accuracy, while the recall capacity on proprietary models (e.g., Adobe Firefly and StarryAI) still has room for improvement.

The cross-forgery results are shown in Table 12. Effort achieves the best performance among previous detectors. The performance of most detectors are limited. For instance, those detectors tailored for deepfake facial images (i.e., IID and ProDet), showing extremely limited recall capacities when encountering facial attribute editing and face relighting. Most methods achieve moderate performance on face restoration and personalization. These results verify that current detectors exhibit limited abilities to generalize unseen forgeries. Besides, it is worth noting that Effort achieve excellent performance on face relighting, greatly surpassing our method. We suppose the reason is that Effort freezes CLIP's semantic encoder, allowing the model to focus solely on detecting whether an

Table 10: Performance comparison on the **In-Domain (ID)** subset of HydraFake dataset. The best results are **bolded** and the second best are underlined. We report Accuracy (Acc.), Precision (P.) and Recall (R.) and the averaged results (Avg.) are reported in Accuracy.

| Method | FaceForensics++ | | | Facevid2vid | | | Hallo2 | | | StyleGAN | | | Midjourney | | | Avg. |
|---|---|---|---|---|---|---|---|---|---|---|---|---|---|---|---|---|
| | Acc. | P. | R. | Acc. | P. | R. | Acc. | P. | R. | Acc. | P. | R. | Acc. | P. | R. | |
| F3Net (*ECCV'20*) | 84.5 | 80.5 | 90.9 | 89.6 | 83.6 | 98.6 | 85.0 | 78.3 | 97.0 | 82.5 | 77.4 | 91.6 | 84.8 | 78.6 | 95.6 | 85.3 |
| UniFD (*CVPR'23*) | 73.8 | 80.4 | 62.8 | 81.7 | 83.2 | 79.6 | 77.3 | 88.8 | 62.5 | 94.3 | 91.5 | 97.6 | 86.3 | 90.6 | 81.0 | 82.7 |
| IID (*CVPR'23*) | 83.8 | 86.6 | 79.9 | 92.8 | 89.5 | 97.0 | 79.1 | 72.2 | 94.6 | 80.7 | 72.1 | 100.0 | 80.5 | 71.9 | 100.0 | 83.4 |
| FreqNet (*AAAI'24*) | 52.2 | 51.2 | 94.4 | 54.5 | 52.3 | 100.0 | 71.2 | 66.7 | 84.5 | 77.5 | 69.8 | 96.6 | 78.8 | 70.8 | 98.0 | 66.8 |
| ProDet (*NIPS'24*) | 84.8 | 82.2 | 88.8 | 90.9 | 84.6 | 100.0 | 91.4 | 89.0 | 94.4 | 92.5 | 88.7 | 97.3 | 93.0 | 88.4 | 99.0 | 90.5 |
| NPR (*CVPR'24*) | 59.6 | 56.5 | 84.1 | 66.6 | 60.5 | 95.7 | 74.3 | 87.2 | 57.0 | 88.8 | 91.7 | 85.3 | 88.7 | 92.9 | 83.6 | 75.6 |
| AIDE (*ICLR'25*) | 58.9 | 58.7 | 59.6 | 78.6 | 70.3 | 94.5 | 84.5 | 92.4 | 75.2 | 86.8 | 93.1 | 99.0 | 93.0 | 92.7 | 93.3 | 80.4 |
| Co-SPY (*CVPR'25*) | 71.3 | 76.0 | 62.3 | 89.3 | 82.9 | 99.1 | 82.1 | 91.6 | 70.7 | 95.0 | 92.2 | 98.3 | 94.0 | 94.0 | 94.0 | 86.3 |
| D³ (*CVPR'25*) | 72.5 | 70.3 | 77.8 | 82.1 | 74.7 | 96.9 | 91.1 | 91.7 | 90.3 | 96.3 | 94.2 | 98.6 | 94.7 | 91.3 | 98.6 | 87.3 |
| Effort (*ICML'25*) | 92.9 | 94.4 | 91.2 | 96.4 | 94.6 | 98.4 | 95.9 | 95.3 | 96.5 | 95.0 | 97.5 | 92.3 | 93.5 | 96.1 | 90.6 | 94.7 |
| Qwen2.5-VL-7B | 51.3 | 93.5 | 2.8 | 51.5 | 94.3 | 3.3 | 50.0 | 50.0 | 1.1 | 51.5 | 80.0 | 4.0 | 51.5 | 76.4 | 4.3 | 51.2 |
| InternVL3-8B | 55.9 | 55.5 | 59.0 | 53.7 | 53.6 | 54.6 | 56.3 | 83.8 | 15.6 | 52.0 | 66.6 | 8.0 | 52.3 | 69.4 | 8.3 | 54.0 |
| MiMo-VL-7B | 56.2 | 55.3 | 64.0 | 62.2 | 59.4 | 77.1 | 60.4 | 63.9 | 47.6 | 66.4 | 69.3 | 58.6 | 73.6 | 72.0 | 77.3 | 63.8 |
| GLM-4.1V-9BThink | 58.0 | 64.0 | 36.3 | 59.6 | 65.6 | 40.0 | 54.2 | 89.6 | 9.4 | 56.0 | 87.5 | 14.0 | 54.3 | 80.9 | 11.3 | 56.4 |
| GPT-4o | 49.2 | 20.0 | 0.5 | 53.0 | 84.2 | 8.0 | 49.8 | 33.3 | 0.5 | 63.2 | 100.0 | 26.5 | 52.5 | 91.6 | 5.5 | 53.5 |
| Gemini-2.5-Pro | 62.0 | 77.9 | 33.5 | 53.2 | 66.6 | 13.0 | 66.0 | 80.9 | 42.5 | 93.9 | 89.3 | 100.0 | 85.8 | 73.7 | 76.6 | 72.2 |
| **VERITAS (ours)** | 90.9 | 90.2 | 91.7 | 96.1 | 92.7 | 100.0 | 99.8 | 99.6 | 100.0 | 99.8 | 99.6 | 100.0 | 100.0 | 100.0 | 100.0 | **97.3** |

Table 11: Performance comparison on the **Cross-Model (CM)** subset of HydraFake dataset. The best results are **bolded** and the second best are underlined. We report Accuracy (Acc.), Precision (P.) and Recall (R.) and the averaged results (Avg.) are reported in Accuracy.

| Method | Adobe Firefly | | | FLUX1.1Pro | | | StarryAI | | | MAGI-1 | | | HART (VAR) | | | Infinity (VAR) | | | Avg. |
|---|---|---|---|---|---|---|---|---|---|---|---|---|---|---|---|---|---|---|---|
| | Acc. | P. | R. | Acc. | P. | R. | Acc. | P. | R. | Acc. | P. | R. | Acc. | P. | R. | Acc. | P. | R. | |
| F3Net (*ECCV'20*) | 86.7 | 80.0 | 97.7 | 87.8 | 80.4 | 100.0 | 78.6 | 76.2 | 83.3 | 85.0 | 78.6 | 96.2 | 86.0 | 78.6 | 99.0 | 82.9 | 78.6 | 90.4 | 84.5 |
| UniFD (*CVPR'23*) | 90.7 | 92.9 | 88.0 | 93.8 | 91.7 | 96.3 | 82.5 | 88.8 | 74.3 | 73.0 | 84.7 | 56.1 | 94.4 | 92.2 | 97.0 | 90.7 | 92.8 | 88.2 | 87.5 |
| IID (*CVPR'23*) | 83.3 | 75.0 | 100.0 | 82.8 | 74.4 | 100.0 | 80.0 | 72.1 | 97.6 | 80.2 | 72.6 | 96.5 | 81.1 | 72.6 | 100.0 | 82.3 | 73.7 | 100.0 | 81.6 |
| FreqNet (*AAAI'24*) | 60.3 | 59.5 | 64.3 | 76.7 | 68.4 | 99.0 | 59.0 | 59.1 | 58.3 | 69.2 | 65.5 | 81.1 | 77.1 | 69.4 | 96.7 | 75.1 | 68.7 | 92.3 | 69.6 |
| ProDet (*NIPS'24*) | 92.6 | 88.3 | 98.3 | 94.2 | 89.5 | 100.0 | 88.2 | 88.0 | 88.3 | 91.9 | 87.5 | 97.7 | 93.8 | 89.0 | 99.8 | 93.1 | 88.7 | 98.6 | 92.3 |
| NPR (*CVPR'24*) | 68.8 | 86.0 | 45.0 | 91.2 | 91.5 | 90.6 | 59.5 | 78.8 | 26.0 | 82.6 | 91.9 | 71.5 | 91.3 | 93.1 | 89.3 | 84.0 | 92.6 | 73.8 | 79.6 |
| AIDE (*ICLR'25*) | 68.8 | 85.5 | 43.3 | 86.3 | 91.0 | 80.6 | 64.0 | 86.6 | 33.0 | 88.9 | 93.6 | 83.6 | 95.4 | 94.3 | 96.6 | 76.0 | 91.0 | 57.7 | 79.9 |
| Co-SPY (*CVPR'25*) | 93.5 | 93.6 | 93.3 | 95.5 | 94.1 | 97.0 | 85.3 | 93.4 | 76.0 | 93.3 | 92.2 | 72.7 | 96.6 | 94.5 | 98.9 | 95.3 | 94.1 | 96.7 | 93.3 |
| D³ (*CVPR'25*) | 93.6 | 90.3 | 97.7 | 95.6 | 92.3 | 99.6 | 91.3 | 92.2 | 90.3 | 90.7 | 90.5 | 91.0 | 95.8 | 92.2 | 99.9 | 95.5 | 93.1 | 98.3 | 93.8 |
| Effort (*ICML'25*) | 82.8 | 97.1 | 67.6 | 96.5 | 96.0 | 97.0 | 78.0 | 94.2 | 59.6 | 90.5 | 96.5 | 84.1 | 97.8 | 97.2 | 98.5 | 98.3 | 97.3 | 99.4 | 90.7 |
| Qwen2.5-VL-7B | 50.0 | 50.0 | 1.3 | 50.0 | 50.0 | 1.3 | 49.7 | 0.0 | 0.0 | 50.0 | 50.0 | 0.9 | 52.0 | 83.8 | 4.9 | 52.9 | 87.3 | 6.8 | 50.8 |
| InternVL3-8B | 54.0 | 74.0 | 12.3 | 49.8 | 47.3 | 3.0 | 49.0 | 28.5 | 1.3 | 56.6 | 83.5 | 16.4 | 55.8 | 83.1 | 14.5 | 57.2 | 84.5 | 17.6 | 53.7 |
| MiMo-VL-7B | 74.5 | 74.7 | 74.0 | 77.1 | 76.7 | 78.0 | 82.5 | 68.0 | 55.3 | 60.3 | 64.0 | 47.1 | 82.4 | 77.9 | 90.6 | 81.4 | 77.2 | 89.2 | 76.4 |
| GLM-4.1V-9BThink | 55.2 | 82.9 | 13.0 | 52.3 | 76.9 | 6.6 | 50.5 | 66.6 | 2.0 | 51.6 | 81.5 | 4.2 | 68.4 | 96.1 | 38.2 | 60.7 | 95.5 | 22.5 | 56.5 |
| GPT-4o | 57.7 | 96.9 | 16.0 | 52.0 | 83.3 | 5.0 | 51.4 | 85.7 | 3.0 | 59.9 | 80.0 | 26.4 | 81.2 | 92.5 | 67.9 | 54.8 | 87.5 | 10.6 | 59.5 |
| Gemini-2.5-Pro | 64.9 | 78.8 | 41.2 | 92.4 | 90.3 | 94.9 | 82.8 | 92.3 | 72.0 | 62.5 | 79.0 | 34.1 | 93.4 | 88.5 | 100.0 | 93.2 | 89.1 | 98.5 | 81.5 |
| **VERITAS (ours)** | 94.8 | 99.6 | 89.9 | 99.8 | 99.6 | 100.0 | 97.0 | 100.0 | 94.0 | 99.9 | 99.8 | 100.0 | 99.9 | 99.8 | 100.0 | 99.9 | 99.8 | 99.9 | **98.6** |

image has been manipulated, which is critical in detecting relighting where the identities and other semantics are largely unchanged.

The cross-domain results are shown in Table 13. Co-SPY achieves the best results among previous detectors. Most methods including ours achieve degraded performance. Specifically, previous methods almost fail on all cross-domain subsets, while our method still achieves robust performance on in-the-wild forgeries (e.g., 92.3% on Dreamina and 89.2% on GPT-4o). The performance on Deep-FaceLab is extremely limited. Different from cross-forgery and cross-model scenarios, the poor performance is due to low Precision. Similar problem also exists in previous detectors. This means those unseen low resolution real images are hard for model to distinguish. Overall, our model strikes great improvements on cross-domain scenarios.

Besides, the zero-shot MLLMs tend to classify facial images into real photographs. Even GPT-4o fails on many cases, exhibiting extremely low recall (i.e., less than 10%). Gemini-2.5-Pro demonstrates strong capacities for deepfake detection, especially on high-resolution images, even beating most fine-tuned specialized detectors. Aggregating the above observations, we can find a intuitive but interesting phenomenon: the MLLM-based detectors (especially reasoning MLLMs) are good at analyzing high-resolution images (typically over $512 \times 512$) but fall short on low-resolution counterparts. Once the MLLMs can "see" the image details, they are able to make accurate judgments

Table 12: Performance comparison on the **Cross-Forgery (CF)** subset of HydraFake dataset. The best results are **bolded** and the second best are underlined. We report Accuracy (Acc.), Precision (P.) and Recall (R.) and the averaged results (Avg.) are reported in Accuracy.

| Method | StarGANv2 | | | IC-Light | | | CodeFormer | | | InfiniteYou | | | PuLID | | | FaceAdapter | | | Avg. |
|---|---|---|---|---|---|---|---|---|---|---|---|---|---|---|---|---|---|---|---|
| | Acc. | P. | R. | Acc. | P. | R. | Acc. | P. | R. | Acc. | P. | R. | Acc. | P. | R. | Acc. | P. | R. | |
| F3Net (*ECCV'20*) | 41.3 | 20.7 | 6.2 | 48.9 | 47.9 | 24.8 | 71.9 | 72.8 | 69.8 | 84.9 | 78.2 | 96.8 | 85.5 | 79.5 | 95.7 | 72.6 | 77.5 | 62.8 | 67.5 |
| UniFD (*CVPR'23*) | 61.8 | 81.7 | 30.4 | 81.9 | 89.1 | 72.6 | 75.4 | 88.0 | 58.7 | 73.7 | 87.0 | 55.6 | 68.1 | 83.6 | 45.1 | 81.3 | 90.3 | 69.6 | 73.7 |
| IID (*CVPR'23*) | 41.4 | 32.8 | 16.6 | 53.3 | 54.3 | 41.0 | 79.7 | 72.9 | 94.4 | 81.8 | 73.4 | 99.6 | 81.8 | 73.3 | 99.9 | 73.7 | 68.2 | 87.1 | 68.6 |
| FreqNet (*AAAI'24*) | 33.1 | 11.4 | 5.0 | 73.1 | 67.6 | 88.9 | 70.3 | 66.6 | 81.5 | 72.8 | 67.4 | 88.3 | 77.4 | 69.9 | 96.2 | 67.7 | 64.9 | 75.0 | 65.7 |
| ProDet (*NIPS'24*) | 56.3 | 67.5 | 24.5 | 58.6 | 69.8 | 34.4 | 80.8 | 84.7 | 75.3 | 88.1 | 89.9 | 85.8 | 91.0 | 88.1 | 94.9 | 83.3 | 86.0 | 79.0 | 76.4 |
| NPR (*CVPR'24*) | 47.7 | 23.6 | 2.1 | 67.8 | 85.2 | 43.2 | 60.6 | 78.3 | 29.3 | 79.8 | 99.1 | 66.0 | 89.0 | 92.8 | 84.5 | 67.7 | 84.9 | 41.9 | 68.8 |
| AIDE (*ICLR'25*) | 56.7 | 74.6 | 20.3 | 79.2 | 91.5 | 64.3 | 86.1 | 90.8 | 80.2 | 74.2 | 89.4 | 55.0 | 62.4 | 83.9 | 30.5 | 75.7 | 90.3 | 56.7 | 72.4 |
| Co-SPY (*CVPR'25*) | 77.0 | 91.1 | 59.9 | 92.5 | 93.8 | 91.0 | 88.6 | 90.3 | 86.5 | 90.6 | 93.2 | 87.5 | 79.1 | 89.9 | 65.5 | 87.3 | 88.7 | 85.4 | 85.9 |
| D³ (*CVPR'25*) | 62.4 | 82.4 | 31.5 | 71.6 | 86.3 | 51.4 | 82.9 | 90.9 | 73.1 | 80.0 | 88.7 | 68.7 | 82.4 | 89.8 | 73.1 | 73.7 | 84.1 | 57.4 | 75.5 |
| Effort (*ICML'25*) | 64.7 | 93.2 | 31.7 | 94.8 | 96.3 | 93.2 | 89.7 | 97.1 | 81.9 | 89.5 | 96.0 | 82.3 | 92.9 | 96.2 | 89.2 | 88.0 | 95.0 | 80.2 | 86.6 |
| Qwen2.5-VL-7B | 50.5 | 64.7 | 2.2 | 56.7 | 90.8 | 15.2 | 50.7 | 70.0 | 2.4 | 53.6 | 90.3 | 8.1 | 54.5 | 90.4 | 10.1 | 51.6 | 63.6 | 4.7 | 52.9 |
| InternVL3-8B | 62.9 | 87.7 | 30.0 | 54.2 | 77.1 | 12.0 | 62.9 | 86.9 | 30.5 | 63.6 | 92.1 | 29.7 | 54.8 | 80.1 | 12.7 | 67.7 | 96.3 | 35.8 | 61.0 |
| MiMo-VL-7B | 48.7 | 47.7 | 26.2 | 82.6 | 76.4 | 94.3 | 76.4 | 75.1 | 78.9 | 79.7 | 76.3 | 85.7 | 78.4 | 75.8 | 83.5 | 82.8 | 79.1 | 88.3 | 74.8 |
| GLM-4.1V-9BThink | 54.3 | 89.8 | 9.7 | 68.4 | 94.7 | 39.0 | 63.3 | 95.7 | 27.8 | 65.7 | 96.5 | 32.6 | 55.1 | 87.4 | 11.9 | 81.0 | 97.9 | 62.8 | 64.6 |
| GPT-4o | 66.4 | 82.6 | 41.5 | 58.9 | 100.0 | 18.0 | 52.5 | 91.6 | 5.5 | 64.4 | 100.0 | 29.0 | 60.9 | 94.0 | 23.5 | 55.5 | 100.0 | 10.1 | 59.8 |
| Gemini-2.5-Pro | 73.7 | 89.7 | 53.5 | 83.3 | 89.7 | 75.1 | 87.4 | 86.0 | 89.3 | 85.5 | 86.2 | 84.5 | 84.7 | 88.4 | 80.0 | 85.6 | 85.3 | 85.8 | 83.4 |
| **VERITAS (ours)** | 90.3 | 99.5 | 81.0 | 75.7 | 99.5 | 51.5 | 97.0 | 98.6 | 95.3 | 91.8 | 98.9 | 84.5 | 95.1 | 99.5 | 90.6 | 91.7 | 98.6 | 84.6 | **90.3** |

Table 13: Performance comparison on the **Cross-Domain (CD)** subset of HydraFake dataset. The best results are **bolded** and the second best are underlined. We report Accuracy (Acc.), Precision (P.) and Recall (R.) and the averaged results (Avg.) are reported in Accuracy.

| Method | DeepFaceLab | | | InfiniteYou-CD | | | Dreamina | | | Hailuo AI | | | GPT-4o | | | FFIW | | | Avg. |
|---|---|---|---|---|---|---|---|---|---|---|---|---|---|---|---|---|---|---|---|
| | Acc. | P. | R. | Acc. | P. | R. | Acc. | P. | R. | Acc. | P. | R. | Acc. | P. | R. | Acc. | P. | R. | |
| F3Net (*ECCV'20*) | 57.7 | 54.7 | 89.1 | 78.5 | 72.6 | 91.6 | 55.6 | 55.0 | 60.9 | 68.6 | 63.3 | 88.6 | 66.2 | 63.6 | 75.5 | 66.4 | 64.8 | 71.4 | 65.5 |
| UniFD (*CVPR'23*) | 67.4 | 64.1 | 79.2 | 67.3 | 75.9 | 50.5 | 80.5 | 77.2 | 86.3 | 75.2 | 75.4 | 74.8 | 73.3 | 73.6 | 72.7 | 67.5 | 65.4 | 74.5 | 71.9 |
| IID (*CVPR'23*) | 65.2 | 65.1 | 65.5 | 69.9 | 63.0 | 95.9 | 63.8 | 58.0 | 99.6 | 63.3 | 57.6 | 100.0 | 63.8 | 58.2 | 97.8 | 64.2 | 66.6 | 56.6 | 65.0 |
| FreqNet (*AAAI'24*) | 50.6 | 50.3 | 98.7 | 67.0 | 62.3 | 85.8 | 62.1 | 58.4 | 83.4 | 59.3 | 56.9 | 76.8 | 58.3 | 56.3 | 73.3 | 51.2 | 50.6 | 95.0 | 58.1 |
| ProDet (*NIPS'24*) | 58.1 | 54.6 | 95.3 | 82.9 | 81.3 | 85.4 | 71.3 | 71.3 | 71.4 | 75.6 | 71.2 | 86.0 | 66.3 | 69.9 | 57.4 | 74.1 | 73.0 | 76.5 | 71.4 |
| NPR (*CVPR'24*) | 52.6 | 52.4 | 56.6 | 73.0 | 82.1 | 58.7 | 76.6 | 80.9 | 69.5 | 62.3 | 70.7 | 42.0 | 50.2 | 50.4 | 20.9 | 46.0 | 46.3 | 50.9 | 60.1 |
| AIDE (*ICLR'25*) | 59.7 | 57.5 | 73.9 | 67.9 | 75.2 | 53.2 | 49.7 | 49.4 | 25.2 | 58.0 | 60.6 | 45.8 | 51.9 | 53.0 | 33.3 | 59.2 | 57.3 | 71.2 | 57.7 |
| Co-SPY (*CVPR'25*) | 67.6 | 63.8 | 81.2 | 80.0 | 85.1 | 72.7 | 82.5 | 80.7 | 85.3 | 74.0 | 75.7 | 70.6 | 79.5 | 78.7 | 80.9 | 64.3 | 64.2 | 64.7 | 74.7 |
| D³ (*CVPR'25*) | 69.7 | 70.0 | 69.1 | 74.6 | 78.8 | 67.2 | 78.1 | 72.2 | 91.4 | 70.9 | 67.7 | 79.8 | 80.8 | 75.1 | 92.0 | 64.3 | 66.0 | 58.7 | 73.1 |
| Effort (*ICML'25*) | 64.8 | 59.1 | 96.3 | 82.2 | 75.4 | 95.5 | 61.5 | 57.6 | 86.5 | 66.4 | 60.1 | 97.4 | 53.8 | 52.9 | 69.3 | 74.0 | 68.3 | 89.7 | 67.1 |
| Qwen2.5-VL-7B | 50.7 | 71.7 | 2.4 | 53.6 | 92.8 | 7.9 | 80.2 | 99.4 | 60.7 | 67.5 | 100.0 | 35.1 | 52.5 | 100.0 | 5.1 | 50.5 | 56.1 | 4.5 | 59.2 |
| InternVL3-8B | 54.4 | 55.5 | 44.1 | 67.1 | 83.6 | 42.7 | 77.1 | 85.4 | 65.3 | 66.5 | 78.9 | 45.0 | 47.4 | 37.5 | 7.6 | 51.8 | 52.0 | 45.1 | 60.7 |
| MiMo-VL-7B | 57.7 | 56.4 | 68.6 | 75.6 | 73.7 | 79.6 | 79.4 | 73.5 | 91.9 | 70.7 | 68.3 | 76.5 | 67.7 | 67.1 | 69.5 | 54.9 | 54.8 | 56.2 | 67.7 |
| GLM-4.1V-9BThink | 58.7 | 82.6 | 22.0 | 72.7 | 95.1 | 47.8 | 83.7 | 96.0 | 70.3 | 69.2 | 92.1 | 42.0 | 52.0 | 68.4 | 8.2 | 53.9 | 60.8 | 21.7 | 65.0 |
| GPT-4o | 49.4 | 41.7 | 2.5 | 62.0 | 98.0 | 24.5 | 90.7 | 98.8 | 82.4 | 73.7 | 100.0 | 47.5 | 58.0 | 94.4 | 17.0 | 52.8 | 59.3 | 17.8 | 64.4 |
| Gemini-2.5-Pro | 67.2 | 72.4 | 55.6 | 75.6 | 88.7 | 59.0 | 87.5 | 89.5 | 84.9 | 82.4 | 95.2 | 68.2 | 70.9 | 96.6 | 43.2 | 53.0 | 67.6 | 11.5 | 72.8 |
| **VERITAS (ours)** | 58.6 | 54.7 | 100.0 | 84.1 | 94.1 | 72.8 | 92.3 | 90.0 | 95.1 | 90.2 | 86.5 | 95.2 | 89.2 | 86.3 | 93.2 | 78.5 | 76.1 | 83.0 | **82.2** |

and provide human-aligned reasoning process. Conversely, small vision models exhibit certain advantages on low-resolution images, as such data is more suited for distribution modeling. Therefore, a collaborative system of MLLMs and small models could be a promising future direction.

## A.5.2 CROSS BENCHMARK COMPARISON

In Table 14, we provide a cross benchmark comparison. We select the facial subsets from AI-GIBench (Li et al., 2025). Note that these subsets also remain unseen in our HydraFake training set, which serves as an OOD testing of our method. Since the real splits contain images of common objects, we substitute these images with facial images from VFHQ. To investigate the impact of training sources, we also train existing methods on FF++ (Rossler et al., 2019) similar to previous one-to-many setting. The quantity of training samples for FF++ and HydraFake are kept consistent (both are 48K). Specifically: **(1)** our HydraFake-CD is more challenging than AIGIBench, with the best result being 6.7% lower (82.2% vs. 88.9%) and the second best result showing 10.0% decrease (74.7% vs. 84.7%). **(2)** On broader datasets, recent AIGC detection methods (e.g., Co-SPY and Effort) still demonstrates clear advantages to the methods tailored for deepfake (e.g., IID and ProDet). This may indicate that specialized modules for facial images may struggle to generalize well to modern fully synthesized and high-fidelity deepfakes. On these data, concurrently modeling the semantics and artifacts (like Co-SPY and Effort) may be more effective. **(3)** Expanding the

Table 14: Cross benchmark comparison. Performance (Acc.) on AIGIBench (Li et al., 2025) and the HydraFake-CD set. For previous methods, we implement two settings: (1) **train on FF++** (Rossler et al., 2019) similar to previous setting, and (2) **train on HydraFake** dataset that contain multiple sources. The quantity of training samples for FF++ and HydraFake are kept consistent. The performance of recent methods increase when trained on more diverse sources (highlighted in gray ), while similar gains are not observed for deepfake detection methods (highlighted in blue ).

| Method | Training | HydraFake-CD | | | | | | | AIGIBench | | | | | | | | |
| | | Deepface. | Infinite Y. | Dreamina | Hailuo AI | GPT-4o | FFIW | Avg. | BLIP | EAS | InfinitelD | InSwap | IPAdapter | R3GAN | StyleSwin | WITR | Avg. |
|---|---|---|---|---|---|---|---|---|---|---|---|---|---|---|---|---|---|
| F3Net (*ECCV'20*) | FF++ | 52.9 | 56.4 | 63.8 | 76.0 | 69.5 | 52.6 | 61.9 | 65.6 | 44.1 | 51.8 | 52.5 | 72.7 | 51.1 | 47.1 | 46.1 | 53.9 |
| UniFD (*CVPR'23*) | FF++ | 72.9 | 54.3 | 63.0 | 59.2 | 57.6 | 66.9 | 62.3 | 51.3 | 80.7 | 77.1 | 63.5 | 67.3 | 67.1 | 78.3 | 71.1 | 69.6 |
| IID (*CVPR'23*) | FF++ | 51.1 | 61.8 | 81.1 | 83.3 | 75.6 | 61.2 | 69.0 | 64.6 | 39.2 | 80.8 | 44.7 | 87.3 | 52.6 | 48.0 | 45.6 | 57.9 |
| ProDet (*NIPS'24*) | FF++ | 63.2 | 64.8 | 85.6 | 83.8 | 70.8 | 63.1 | 71.9 | 65.2 | 46.6 | 84.4 | 55.1 | 81.4 | 51.1 | 49.3 | 48.2 | 60.2 |
| Co-SPY (*CVPR'25*) | FF++ | 67.6 | 82.0 | 82.4 | 74.0 | 64.3 | 64.3 | 72.4 | 78.2 | 86.2 | 80.0 | 82.6 | 55.5 | 76.3 | 86.3 | 92.8 | 79.7 |
| D$^3$ (*CVPR'25*) | FF++ | 71.8 | 65.4 | 59.6 | 44.3 | 48.4 | 62.9 | 58.7 | 48.7 | 80.3 | 60.0 | 68.9 | 55.5 | 60.3 | 66.3 | 59.4 | 62.4 |
| Effort (*ICML'25*) | FF++ | 76.2 | 76.9 | 55.3 | 49.0 | 49.7 | 70.7 | 63.0 | 83.8 | 85.9 | 82.2 | 77.3 | 71.3 | 81.4 | 82.3 | 83.1 | 80.9 |
| F3Net (*ECCV'20*) | HydraFake | 57.7 | 78.5 | 55.6 | 68.6 | 66.2 | 66.4 | 65.5 | 72.1 | 77.7 | 69.8 | 80.4 | 62.9 | 53.5 | 44.3 | 74.5 | 66.9 |
| UniFD (*CVPR'23*) | HydraFake | 67.4 | 67.3 | 80.5 | 75.2 | 73.3 | 67.5 | 71.9 | 60.2 | 90.4 | 64.9 | 76.1 | 80.3 | 82.3 | 78.2 | 91.2 | 78.0 |
| IID (*CVPR'23*) | HydraFake | 65.2 | 69.9 | 63.8 | 63.3 | 63.8 | 64.2 | 65.0 | 78.3 | 65.8 | 76.0 | 62.7 | 77.8 | 58.3 | 48.2 | 81.3 | 68.6 |
| ProDet (*NIPS'24*) | HydraFake | 58.1 | 82.9 | 71.3 | 75.6 | 66.3 | 74.1 | 71.4 | 82.4 | 85.9 | 81.1 | 86.7 | 71.7 | 54.3 | 47.7 | 88.4 | 74.8 |
| Co-SPY (*CVPR'25*) | HydraFake | 67.6 | 80.0 | 82.5 | 74.0 | 79.5 | 64.3 | 74.7 | 77.6 | 88.1 | 89.2 | 81.3 | 57.4 | 78.7 | 86.2 | 94.2 | 81.6 |
| D$^3$ (*CVPR'25*) | HydraFake | 69.7 | 74.6 | 78.1 | 70.9 | 80.8 | 64.3 | 73.1 | 71.4 | 87.8 | 80.9 | 78.9 | 68.6 | 77.9 | 63.7 | 93.4 | 77.8 |
| Effort (*ICML'25*) | HydraFake | 64.8 | 82.2 | 61.5 | 66.4 | 53.8 | 74.0 | 67.1 | 82.1 | 89.1 | 84.0 | 85.8 | 84.8 | 87.1 | 82.3 | 82.7 | 84.7 |
| VERITAS (ours) | HydraFake | 58.6 | 84.1 | 92.3 | 90.2 | 89.2 | 78.5 | 82.2 | 81.9 | 93.5 | 88.4 | 89.3 | 81.8 | 91.3 | 85.2 | 99.8 | 88.9 |

Table 15: Performance comparison on broader benchmarks, including LOKI (Ye et al., 2024), Fake-Clue (Wen et al., 2025c), Forensics-Bench (Wang et al., 2025b), AIGIBench (Li et al., 2025) and Nano-banana-150K (Ye et al., 2025b). Results of facial data in LOKI are also reported, since we target at deepfake detection.

| Method | LOKI | | LOKI (facial) | | FakeClue | | Forensics-Bench | | AIGIBench | | Nano-banana | |
| | Acc. | F1 | Acc. | F1 | Acc. | F1 | Acc. | F1 | Acc. | F1 | Acc. | F1 |
|---|---|---|---|---|---|---|---|---|---|---|---|---|
| UniFD (*CVPR'23*) | 54.5 | 58.7 | 74.8 | 69.7 | 61.6 | 64.2 | 53.6 | 54.8 | 78.0 | 80.2 | 49.1 | 36.0 |
| ProDet (*NIPS'24*) | 53.8 | 56.6 | 63.2 | 66.4 | 62.9 | 69.8 | 65.1 | 72.0 | 74.8 | 73.9 | 64.6 | 63.8 |
| Co-SPY (*CVPR'25*) | 61.7 | 65.8 | 79.1 | 75.6 | 68.1 | 72.4 | 70.8 | 76.0 | 81.6 | 84.3 | 52.0 | 40.9 |
| D$^3$ (*CVPR'25*) | 47.3 | 41.2 | 79.5 | 80.1 | 60.7 | 59.2 | 56.6 | 59.8 | 77.8 | 75.3 | 70.7 | 73.0 |
| Effort (*ICML'25*) | 53.8 | 50.0 | 84.3 | 84.6 | 65.0 | 63.2 | 57.0 | 59.4 | 84.7 | 87.6 | 62.7 | 52.1 |
| InternVL3-8B | 53.0 | 51.6 | 52.6 | 15.3 | 59.1 | 62.2 | 60.5 | 65.4 | 55.6 | 56.5 | 51.3 | 38.9 |
| MiMo-VL-7B | 65.1 | 64.3 | 69.7 | 65.0 | 67.2 | 71.6 | 63.8 | 70.5 | 62.8 | 64.3 | 60.7 | 55.6 |
| VERITAS (ours) | 72.1 | 77.8 | 89.0 | 88.2 | 85.9 | 88.4 | 70.8 | 74.9 | 88.9 | 90.4 | 86.3 | 89.0 |

training data from FF++ to HydraFake brings promising gains for recent AIGC detection methods. For instance, D$^3$ increases from 58.7% to 73.1% on HydraFake. However, similar gains are not observed for deepfake detection methods, e.g., both IID and ProDet suffer from performance drop on HydraFake when trained on more diverse sources. This further reveal *the gap* between current deepfake detection methods and practical usage. When we have abundant training sources, *the generalization performance does not scale up as expected*. A comprehensive benchmark is necessary to measure the detectors' capacities more practically. In Table 15, we conduct additional evaluations on extensive benchmarks, including generic AIGC detection task. Notably, VERITAS shows promising performance on these AIGC benchmarks, e.g., 72.1% on LOKI and 85.9% on FakeClue. Note that VERITAS is only trained with facial forgery data. Moreover, VERITAS generalizes well on the latest editing model (i.e., Nano-banana). We also provide reasoning cases in Figure 23, 24, 25, 26, 27 to show the impressive adapation capacities of VERITAS.

### A.5.3 EFFICIENCY COMPARISON

In Table 16, we provide an efficiency analysis of our model. All the data are obtained on a single PPUE GPU with the original Transformers library implementation. We report the averaged inference time of a batch of images with the batch size set to 8. Specifically, we compare the efficiency of different reasoning paradigms of MLLMs. For post-hoc explanation and flexible reasoning models, we do not perform MiPO since the human annotated data is hard to obtain. We present a experi-

Table 16: Analysis on efficiency. We calculate the inference time (seconds) of a batch of images with the batch size set to 8. The experiments are conducted on a single PPUE GPU with the `Transformers` library. We select samples of different resolutions and difficulty for illustration.

| Model | Low Resolution ($\downarrow$) | | High Resolution ($\downarrow$) | | Acc. ($\uparrow$) |
|---|---|---|---|---|---|
| | Easy | Hard | Easy | Hard | |
| Post-hoc Explanation | 27.08 | 28.19 | **37.64** | **36.06** | 83.4 |
| Flexible Reasoning | 24.60 | 29.37 | 44.43 | 47.44 | 86.8 |
| Ours (cold-start) | **19.26** | **24.94** | 40.72 | 46.40 | 89.3 |
| Ours | 22.35 | 25.60 | 49.76 | 54.22 | **92.1** |
| $\Delta$ **Post-hoc Exp.** | ↓**4.73** | ↓**2.59** | ↑**12.12** | ↑**18.16** | ↑**8.7** |
| $\Delta$ **Flexible Reason.** | ↓**2.25** | ↓**3.77** | ↑**5.33** | ↑**6.78** | ↑**5.3** |

Table 17: Ablation studies of the hyperparameter $\beta'$ (strength of KL penalty in P-GRPO).

| Value of $\beta'$ | ID | CM | CF | CD |
|---|---|---|---|---|
| $\beta'=0.04$ | 96.8 | 98.4 | 89.1 | 80.2 |
| $\beta'=0.01$ | 96.8 | 98.3 | 89.6 | 81.5 |
| $\beta'=0.001$ | **97.3** | **98.6** | 89.3 | 81.9 |
| $\beta'=0.0$ | **97.3** | **98.6** | **90.3** | **82.2** |

Table 18: Ablation studies of the hyperparameter $G$ (number of generations within each group).

| Value of $G$ | ID | CM | CF | CD |
|---|---|---|---|---|
| $G=4$ | 97.3 | **98.6** | **90.3** | **82.2** |
| $G=8$ | **97.4** | 98.4 | 89.8 | 82.0 |
| $G=12$ | 96.7 | 97.6 | 88.8 | 80.5 |
| $G=16$ | 96.9 | 93.2 | 88.9 | 80.4 |

mental prototype here to provide a understanding of the inference efficiency of our model. Since inference efficiency is influenced by input resolutions and task difficulty, we divide samples into four parts. Low resolutions are images with $256 \times 256$ size and high resolutions are $1024 \times 1024$. Specifically, (1) our model achieves faster inference on low-resolution images, while becomes slower on high-resolution inputs. As discussed in Appendix A.5, when the model can perceive finer details, it can perform more thorough reasoning, leading to improved accuracy and more inference time. (2) Compared to flexible reasoning models, VERITAS incurs no significant increase in computational cost, yet achieves a 5.3% performance gain, demonstrating the effectiveness of pattern-aware reasoning. (3) Post-hoc explanation models exhibit little variation in efficiency across easy and hard samples, typically performing rigid, point-to-point analysis without adaptive reasoning. (4) Without the P-GRPO to activate "self-reflection" and "planning" mechanisms, our cold-start model achieves better efficiency while still maintaining competitive performance.

### A.5.4 EFFECT OF TRAINING DATA IN P-GRPO STAGE

In Table 19, we investigate the impact of training data in P-GRPO. For our VERITAS, we adopt balanced sampling among manipulation types, which achieves superior performance compared to randomly sampled data. As adopted in mathematical and coding problems, the hard sampling (the samples that models fail to reach all correct answers in 8 rollouts) achieves inferior performance in our case, but this still yields improvements over cold-start model. Moreover, we add about $1/3$ unseen data from AIGIBench into P-GRPO stage. Note that these data are unseen during previous training stages, but are not overlapped with the testing domain of HydraFake. As shown in Table 19, this yields promising improvements on cross-forgery scenarios (3.8% over our VERITAS). From the observations, we point out that the cold-start model is a *good policy model*. While in this paper we only use in-domain data during P-GRPO for fair comparisons, the users can add OOD data flexibly to elevate the detection ability, which can be achieved in two approaches: (1) (*a cheap and scalable way*) adopt data with binary labels and our P-GRPO for training. (2) (*a fine-grained and controllable way*) use the cold-start model or VERITAS to further construct a high-quality CoT dataset for customized deepfake data. This may require manual preference filtering but can further enhance the reasoning quality on target data.

### A.5.5 ANALYSIS OF HYPERPARAMETERS

**Analysis of hyperparameter $\beta'$.** $\beta'$ controls the strength of penalty when the outputs deviate from the reference model. From Table 17, smaller $\beta'$ yields better performance on cross-forgery and cross-domain sets, suggesting that stronger exploration helps activate advanced reasoning behaviors of the cold-start model, improving generalization on OOD data.

Table 19: Ablation studies on the training data of P-GRPO. The cold-started model serves as the baseline. We try four types of data selection. See analysis for details.

| Training Data | ID | CM | CF | CD |
|---|---|---|---|---|
| Baseline (cold-start) | 96.8 | 95.8 | 85.1 | 79.5 |
| Random (seed=42) | 96.9 | 98.2 | 88.4 | 80.1 |
| Random (seed=100) | 97.0 | 97.9 | 88.0 | 80.6 |
| Balanced sampling | **97.3** | **98.6** | 90.3 | 82.2 |
| Hard sampling | 96.9 | 98.4 | 88.6 | 81.3 |
| Partially unseen | 95.7 | **99.4** | **94.1** | **82.4** |

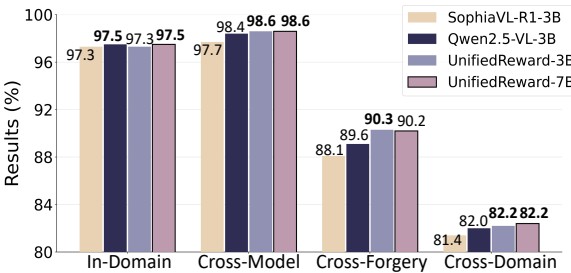

Figure 10: Ablations on the reward model $\mathcal{M}$.

**Analysis of hyperparameter $G$ (number of generated rollouts in P-GRPO).** From Table 18, more generations within one group do not bring improvements while the training costs increase. We attribute this to the task gap between deeepfake detection and common tasks. While mathematical problems often admit multiple valid solution paths, deepfake detection is a fact-based classification task with a more constrained reasoning space. In such cases, excessive group size leads to redundant exploration. This is also the reason we apply cold-start before the online RL stage, which ensures meaningful exploration during RL.

### A.5.6 Effect of Different Reward Model

We investigate different reward models, including SophiaVL-R1-Thinking-Reward-Model-3B (Fan et al., 2025), Qwen2.5-VL-3B, UnifiedReward-Qwen-3B and UnifiedReward-Qwen-7B. From Figure 10, SophiaVL-R1-3B achieves degraded performance. This is due to that Sophia is specifically trained to measure the quality of CoT, while the instruction following ability is limited, which is not capable of measuring the novelty of reflection content. In contrast, Qwen2.5-VL-3B and UnifiedReward-Qwen-3B can better distinguish the reflection quality. Further scaling up to 7B does not bring significant improvements.

### A.6 Full Prompt Templates

In this section, we provide all the prompt templates used in our method. This includes the following parts:

- The prompts for pattern-aware SFT data annotation. For fake images, the process contains three stages as shown in Figure 28, Figure 29 and Figure 30. For real images, the process contains two stages (without the anomalies detection stage) as shown in Figure 31 and Figure 32.
- The prompts for reasoning quality evaluation, which contains score evaluation (Figure 33) and pairwise evaluation (Figure 34).
- The prompt for generating personalization prompts, as shown in Figure 35.
- The prompt for reflection quality reward model $\mathcal{M}$, as shown in Figure 36.
- The system prompt for our **VERITAS** model as shown in Figure 37. The system prompts for all training stages and inference stage are consistent.
- The prompt for zero-shot inference of MLLMs. We tested several prompts for each MLLM. Firstly we found that directly prompting them to perform pattern-aware reasoning like **VERITAS** fails in most cases. Therefore we perform common CoT instead. The prompts for Qwen2.5-VL-7B, InternVL3-8B and GLM-4.1V-9B-Thinking are provided in Figure 38. For MiMo-VL-7B, we found that providing priori information is harmful to the performance, and we adopt simple system prompt instead, as shown in Figure 39. Similarly, we keep the default system prompt and only constraining the output format in user prompt for GPT-4o and Gemini-2.5-Pro, as shown in Figure 40.

### A.7 More Qualitative Results

We provide more examples of our model's reasoning outputs. Specifically, our model can perform adaptive pattern-aware reasoning, generating direct and concise analysis for obviously fake images.

It can also conduct thorough and holistic analysis for high-fidelity fake images. All the examples except FF++ are from OOD scenarios. It is worth noting that while being trained on pure in-domain facial data, VERITAS exhibits promising AIGC analysis abilities, e.g., the infeasible date of birth on ID card (Figure 16) and over-stylized texture of fabric (Figure 15). Such abilities mainly emerges from the combination of MiPO and P-GRPO. As mentioned in our main text, MiPO ensures high-quality rollouts in subsequent stage, which enables more accurate policy updates for online RL. The effective explorations during RL facilitate the deep reasoning capacities. Note that such observation is different from that of BusterX++ (Wen et al., 2025b), which found that cold-start constrains the output distribution and shrinks the OOD generalization abilities. We suppose the discrepancy is due to the *rich semantics in AI-generated content allow the MLLMs to succeed with pure RL*, since they are proficient at capturing semantic-level clues and is capable of generating high-quality rollouts at initial stage. However, the semantics of deepfake images are extremely limited, with most anoamlies lying on low-level artifacts. In such cases, cold-start is necessary and our work incorporates SFT and MiPO to instill the human-aligned reasoning capacities into base models.

## A.8    MORE QUALITATIVE COMPARISONS WITH EXISTING MLLM-BASED DETECTORS

We provide more reasoning comparisons between VERITAS and existing MLLM-based detectors. Among the compared models, M2F2-Det and FFAA are specialized for deepfake detection, while other methods are generic forgery detection models. As shown in Figure 17, 18, 19, **M2F2-Det** excels at performing faithful analyses within facial region. However, it lacks consideration of deeper dimensions, resulting in suboptimal performance on certain fully synthesized data that require considerations about overall context. **FFAA** provides more detailed analyses. However, the logical coherence between "description" and "reasoning" part is weak, and the "reasoning" part lacks in-depth understanding. **FakeShield** falls short on analyzing fully synthesized facial data, but it demonstrates certain advantages in local artifact analysis (Figure 18 lower), since it is specifically trained for IMDL tasks. **SIDA-13B-description** provides generally high-quality explanations. However, it has a tendency to classify real facial images as fake. **FakeVLM** provides low-quality explanations regarding facial forgeries despite its high detection accuracy, e.g., *most cases are explained as "The image exhibits underlying characteristic inconsistencies in its features that suggest it is artificially created"*. Such vague and template-like explanantions are likely due to its large-scale SFT training nature. In contrast, VERITAS generates holistic and faithful reasoning process.

## A.9    FAILURE ANALYSIS OF VERITAS

For **real** images, the failures mainly clustered at low-resolution data. As shown in Figure 22 upper, these data are generally in low quality, where the unexpected artifacts such as localized blurriness would affect the model's judgement. For **fake** images, failures mainly occur on totally unseen forgery types such as face relighting. However, although the final answer is incorrect, VERITAS still figure out suspicious clues, e.g., "overly uniform water droplets raise **red flag**" and "the warm lighting introduces **uncertainty**" in Figure 22. This providing valuable insights that could be used for further scrutiny or future improvements.

## A.10    THE USE OF LARGE LANGUAGE MODELS

We used LLMs for grammatical refinement and language polishing of the paper, aiming to improve the clarity and readability. Some MLLMs are used for the annotation of reasoning data, which is a common practice. Besides, the LLMs are not involved in research design or idea generation.

## A.11    ETHICS STATEMENT

All real facial data used in this work are from publicly available academic datasets. The fake images include those from public benchmarks and those generated by our team using generative models or face-swapping techniques. The latter were created only from public or synthetic data, with no unauthorized use of personal images. Our work focuses on improving deepfake detection to combat misinformation, and all data are used strictly for non-commercial, academic purposes.

## A.12 Limitations and Future Work

While HydraFake involves multi-level evaluations, it is limited to the image modality. With recent advances in video generation models, extracting frames from videos and detecting manipulations solely based on spatial artifacts is challenging. Moreover, as analyzed in Appendix A.5, our Veritas model still exhibits shortages on low-quality subsets such as DeepFaceLab and FFIW as the reasoning requires more visual details. Therefore, we figure out the **future directions**: **(1)** A collaborative system of MLLMs and small vision models, since the MLLM-based detectors (especially reasoning MLLMs) are good at analyzing high-resolution images while small vision models exhibit certain advantages on low-resolution counterparts. This has been explored by a recent work (Chen et al., 2024b), while how to develop a more adaptive or agent-like system is still an interesting problem. **(2)** A unified image-video deepfake benchmark. Recent video generation models are capable of creating high-fidelity talking faces and hand-face interactions (e.g., touching eyebrows or nose), posing new challenges to facial security systems. Due to high frame-level realism, traditional frame-based detectors often fail. Consequently, there is a growing need for unified detection frameworks capable of handling both image and video inputs, as well as rigorous benchmarks to facilitate the development of robust detection methods.

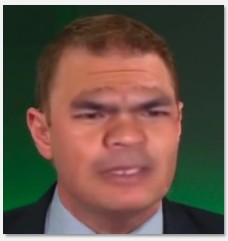

(Fake Image from FF++)

**\<fast\>** At first glance, the image appears to be fake due to noticeable blending boundary. **\</fast\>**
**\<reasoning\>** Upon close inspection. The skin tone around the eyes is darker than the rest of the face, indicating a poor blending job. **\</reasoning\>**
**\<conclusion\>** Considering the noticeable blending boundary and skin tone inconsistency, the image is not a real photograph but rather a manipulated one. **\</conclusion\>**
**\<answer\>** fake **\</answer\>**

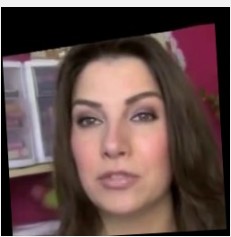

(Fake Image from DF40)

**\<fast\>** At first impression, the image appears relatively realistic, making it difficult to ascertain immediate signs indicative of deepfake generation, which calls for a detailed examination. **\</fast\>**
**\<planning\>** Given the initial semblance of authenticity, I employ a systematic approach to detect potential anomalies. This involves verifying facial proportions, structure, and identifying subtle artifacts that suggest AI generation. Specifically, I focus on three main areas: checking visual symmetry, analyzing texture and edge details, and assessing physical plausibility, such as lighting and shadow alignment, to accurately determine possible discrepancies. **\</planning\>**
**\<reasoning\>** Starting with facial proportion, the initial alignment of features appears balanced and consistent with typical human proportions, initially masking any potential alterations. However, upon closer inspection, the alignment of the eyebrows exhibits subtle deviations. In particular, the left eyebrow shows a noticeable distortion and maintains an unusual angle, which doesn't mirror the natural symmetry expected in human faces. Furthermore, the right eye possesses an oval shape that diverges from the more commonly observed almond-shaped eye form, signalling a potential manipulation or blending issue.
Additionally, examining the hair texture and its interaction with the background, there is an inconsistency—an overly smooth texture near the edge of the hair compared to a stronger outline for the hairline itself. This presents potential digital processing artifacts, suggesting the hairline may have limitations in maintaining its natural state when integrating AI techniques.
Another aspect to consider is the interaction between the subject and the background. The lighting on the subject's face seems generally consistent with the surrounding environment, but the shadowing around the edges of the face, especially on the left side near the distortion in the eyebrow, feels slightly unnatural. This might indicate an attempt to digitally consolidate elements from different sources, further pointing toward manipulation. **\</reasoning\>**
**\<conclusion\>** Despite seemingly realistic initial perceptions, the eyebrow distortion, irregular eye shape, and inconsistencies in hair texturing and shadowing collectively suggest AI-generated wave artifacts. These elements, while subtly present, stand out upon careful scrutiny, identifying the image as medium hard to detect AI generation forgery. These anomalies, discernible through detailed evaluation, strongly indicate the use of face-swapping technology, showcasing inherent flaws characteristic of manipulated imagery. **\</conclusion\>**
**\<answer\>** fake **\</answer\>**

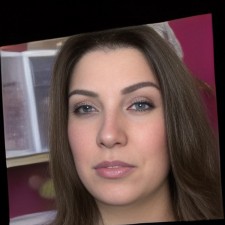

(Fake Image from DF40, **after restoration**)

**\<fast\>** At first glance, the image presents a reasonably realistic appearance without immediately evident anomalies, necessitating deeper inspection to classify its authenticity. **\</fast\>**
**\<planning\>** Given the initial realistic impression, a thorough, systematic analysis is required. The process involves evaluating structural coherence, identifying subtle visual artifacts, and verifying physical feasibility, particularly in the lighting and reflections. This comprehensive approach is necessary to discern the nuances indicative of AI-generated content. **\</planning\>**
**\<reasoning\>** When assessing the structural elements and proportions, they appear harmonious without immediate discrepancies, suggesting an expertly rendered image. Delving into finer details, the texture and coloration raise questions: there is a subtle color difference between the eyes, particularly noticeable in the iris details on the left eye, where the color seems lighter than the other eye. This indicates potential blending errors common in digital transformations. Moreover, the skin on the forehead seems unnaturally smooth, lacking the typical fine lines or pores found in natural skin, highlighting another area where AI might have attempted to enhance realism but introduced inconsistencies instead. These nuances suggest the image may be a result of AI manipulation. Additionally, the background introduces confusion; the sharpness and brightness of the background elements don't align with the focus and lighting on the face, questioning whether they stem from the same original scene or were artificially inserted to simulate depth.
Focusing on the hairline and the surrounding areas, there is a subtle yet noticeable blurring around the edges, an attempt by the AI to blend or separate features unnaturally, often seen in deepfake simulations. These incongruencies, while subtle, point toward artificial modification, as they diverge from the consistent clarity typically found in genuine imagery. Furthermore, the shadows and highlights on the face do not always correspond perfectly with the apparent light source near the top left, implying manipulated lighting effects. **\</reasoning\>**
**\<reflection\>** Upon further reflection, there are subtle discrepancies in the interaction between the facial features and the surrounding elements that enhance the impression of digital alteration. The eyes are half-closed, and the reflection in the eyes leaves something to be desired. Usually, a reflection might show the environment or a slight suggestion of the surrounding lighting, but here the reflection appears either overly subtle or implied rather than naturally integrated. This lack of unpredictability could be a testament to AI generation's focus on feature alignment rather than environmental realism. This subtle reflection misalignment might have been overlooked initially but adds to the growing evidence of forgery. **\</reflection\>**
**\<conclusion\>** Overall, despite its initial realistic perception, a deep exploration uncovers subtle discrepancies in color, texture, subtle errors in background integration, and the incongruent lighting effects that cumulatively indicate the likelihood of AI generation. These features— though subtle—are sufficient to classify the image as a high-quality deepfake, reflecting the sophisticated yet flawed synthesis typical of medium-hard to detect forgeries. Thus, while less apparent at first glance, detailed scrutiny reveals the artificial origins. **\</conclusion\>**
**\<answer\>** fake **\</answer\>**

Figure 11: Reasoning output of our **VERITAS** model.

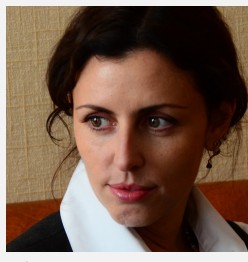

(Real Image from FFHQ)

**<fast>** The image is likely authentic based on initial observation, showing no obvious signs of manipulation. **</fast>**
**<planning>** To confirm the authenticity, a structured analysis is conducted in three progressive stages: first checking for visual anomalies, then identifying any subtle AI-generated artifacts, and finally verifying adherence to physical laws. Since no clear abnormalities are present, deeper inspection is necessary to ensure there are no hidden inconsistencies. **</planning>**
**<reasoning>** The facial structure is natural and symmetrical with consistent proportions and realistic expression, indicating no visible manipulation. Upon closer inspection, the skin texture and edges appear smooth and continuous, with no signs of artificial blending or unnatural blurring. Fine details like skin pores and hair texture are preserved, and the lighting and shadows align with natural illumination. The background matches the subject in style and lighting, and there are no inconsistencies in focus or stylization that would suggest artificial generation. **</reasoning>**
**<conclusion>** After evaluating multiple layers of evidence, the image shows no signs of manipulation and is consistent with a real photograph. The authenticity is confidently supported by the coherence of facial features, texture, and environmental alignment. **</conclusion>**
**<answer>** real **</answer>**

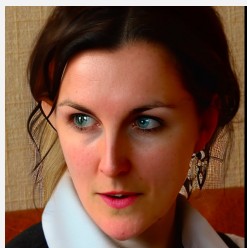

(Fake Image from **Generative Face Swapping,** using FaceAdapter)

**<fast>** At first glance, the image appears realistic with no glaring anomalies, suggesting a high-quality photograph. **</fast>**
**<planning>** Given the initial impression of authenticity, a thorough and systematic examination is necessary. This involves checking structural integrity, identifying subtle visual artifacts, and ensuring physical plausibility in terms of lighting, texture, and focus. This multi-layered approach is essential to uncover any hidden signs of AI manipulation. **</planning>**
**<reasoning>** Upon closer inspection, the facial structure and proportions appear natural, with consistent skin tones and no obvious distortions. The lighting and shadows align well with the background, and the overall composition maintains a realistic appearance. However, subtle inconsistencies begin to emerge: the hair on the left side shows a strange, almost metallic sheen, which is not typical of natural hair and suggests digital alteration. Additionally, the earring on the right side of the face displays an unusual pattern and texture that deviate from real-world expectations, appearing more like a digital artifact than a genuine accessory. The focus on the face is sharp, but the background is slightly blurred, which, while not inherently suspicious, could mask other manipulations. These details, while subtle, raise questions about the image's authenticity. **</reasoning>**
**<reflection>** Wait a moment--there's another point to consider. The lighting on the face seems consistent, but the way it interacts with the hair and accessories is slightly off. The metallic sheen and the earring's texture don't just look unnatural; they feel like they were added to enhance realism but instead create an uncanny effect. This kind of artifact is often a byproduct of AI generation, where the system struggles to replicate realistic textures and materials. Furthermore, the slight mismatch in focus and texture between the face and the accessories suggests that different parts of the image might have been generated or altered separately and then composited. **</reflection>**
**<conclusion>** While the image initially projects a convincing facade, the metallic hair and abnormal earring texture are telltale signs of AI manipulation. These subtle inconsistencies, combined with the unnatural interaction of lighting and focus, indicate that the image is likely a medium-hard to detect forgery. Despite its realistic appearance, these artifacts betray its artificial origins, confirming it as a product of digital generation. **</conclusion>**
**<answer>** fake **</answer>**

Figure 12: Reasoning output of our **VERITAS** model.

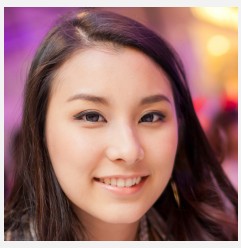 (Real Image from FFHQ)

**<fast>** At first glance, the image appears highly realistic with no obvious signs of manipulation or artificial generation. **</fast>**
**<planning>** To confirm its authenticity, we should systematically analyze the image through three progressive layers: first checking for visual structural abnormalities, then inspecting for subtle AI artifacts, and finally verifying adherence to physical laws. This structured approach ensures a thorough and logical evaluation. **</planning>**
**<reasoning>** The face exhibits consistent skin tone and natural proportions, with symmetric features that conform to typical human biometrics. The expression is coherent, with the eyes matching the subtle cues of a genuine smile. Upon closer inspection, the facial edges are smooth, textures are realistic, and there are no unnatural blurs or missing high-frequency details like fine hair strands or skin pores. The lighting and shadows are consistent, and the blurred background aligns stylistically and optically with the subject's lighting, indicating a natural depth of field. There are no implausible makeup effects or stylization that would suggest digital generation. **</reasoning>**
**<conclusion>** Given the absence of any detectable artifacts across all levels of inspection, the image is best classified as authentic, with all observed traits aligning with those of a real photograph. **</conclusion>**
**<answer>** real **</answer>**

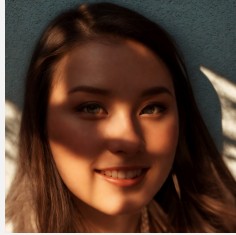 (Fake Image from **Face Relighting**, using IC-Light)

**<fast>** On first inspection, the image appears realistic, but certain aspects seem slightly off, raising a subtle sense of unease about its authenticity. **</fast>**
**<planning>** Given the lack of immediate and obvious flaws, a thorough examination is necessary. My analysis will focus on assessing the structural integrity of the facial features, checking for inconsistencies in texture and sharpness, and analyzing the physical plausibility of lighting and shadows. This multi-layered approach is vital for identifying any subtle characteristics of AI generation that are not apparent at first glance. **</planning>**
**<reasoning>** Initially, the facial structure and proportions appear largely consistent with realistic human features, with symmetrical features and a generally natural appearance. However, upon closer inspection, some anomalies become apparent. The nose appears slightly off-center, which disrupts the natural balance expected in a truly symmetrical face, hinting at possible digital manipulation. The eyes, while seemingly natural, exhibit watery reflections that lack the depth seen in real human eyes, suggesting an artificial rendering. Moving beyond the structural issues, let's examine the brighter parts of the face, particularly around the right cheek and jawline. The skin appears unnaturally smooth and slightly blurred, creating a contrast with the sharper details of the hair around the face. This inconsistency in sharpness, where the hair is clearly defined while the skin lacks clarity, is common in AI-generated images and indicates possible blending errors during generation.
Furthermore, the shadows cast by the hair on the subject's forehead and neck seem too soft and less defined than the hair's clarity relative to the face. These shadows should naturally match the strength and angle of the light source effecting the subject, but their inconsistency suggests poor modeling or rendering by the AI. Observing the overall lighting, there is a noticeable lack of realistic shadow definition and depth when the lighting should naturally create more pronounced contrasts given the intensity of the shadow play. The lighting gives the face a somewhat painted or airbrushed appearance, as if the AI tried to replicate the softness of impressionist painting rather than capture the nuanced, natural shadows and highlights of real photography.
Moreover, the setting and background, while seemingly consistent in tone and ambient lighting, also carry a slightly artificial aura that matches the subtle anomalies in facial features. This uniformity might be too perfect, hinting at a digitally integrated setting rather than a natural capture. The overall image feels a bit too harmonious, reinforcing the possibility of digital synthesis. **</reasoning>**
**<conclusion>** Despite an initially convincing facade, the mismatched symmetry of facial features, the unnatural clarity and blur discrepancies, and inconsistent lighting and shadow modeling reveal the image's AI-generated origins. While these signs are not immediately obvious, the accumulation of these subtle anomalies strongly indicates that the image is a sophisticated but nonetheless artificially generated forgery. These findings confirm the image as medium hard to detect due to the subtlety of its artifacts, requiring careful observation and analysis to uncover its true nature. **</conclusion>**
**<answer>** fake **</answer>**

Figure 13: Reasoning output of our **VERITAS** model.

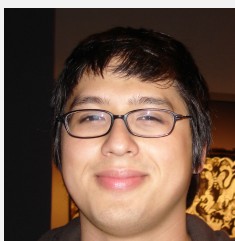 (Real Image from FFHQ)

**\<fast\>** At first glance, the image appears authentic, with natural facial features and consistent lighting. **\</fast\>**
**\<planning\>** Given the lack of immediate anomalies, a structured inspection is planned across three progressive levels: first checking for visible structural issues, then searching for subtle synthetic artifacts, and finally evaluating physical plausibility to ensure no hidden inconsistencies exist. **\</planning\>**
**\<reasoning\>** The facial structure is well-proportioned and symmetrical, with natural expressions and uniform skin tone, showing no signs of artificial manipulation. Upon closer inspection, the edges, textures, and transitions are smooth and realistic, with no signs of digital artifacts such as blurring or missing high-frequency detail. Even under deeper scrutiny, the lighting, shadows, and depth of field behave as expected in a real-world scenario, with no violations of physical laws or contextual mismatches. **\</reasoning\>**
**\<conclusion\>** All levels of analysis confirm the image is authentic, with no indications of synthetic generation or manipulation. It aligns with the characteristics of a high-quality, real-world photograph. **\</conclusion\>**
**\<answer\>** real **\</answer\>**

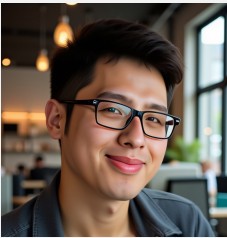 (Fake Image from Face Personalization), using InfiniteYou

**\<fast\>** At first glance, the image appears highly realistic, capturing an individual comfortably positioned in what seems to be a natural and coherent environment. **\</fast\>**
**\<planning\>** Given the absence of obvious anomalies, a thorough and systematic analysis is necessary. My approach involves examining structural consistency, searching for subtle generator-induced artifacts, and verifying the physical plausibility of elements—particularly lighting, reflections, and textures. This three-tiered inspection will aid in uncovering potential signs of manipulation that are not initially evident. **\</planning\>**
**\<reasoning\>** Upon close examination, the lighting on the face initially appears consistent, as it aligns with the ambient conditions of the surrounding setting. However, the reflection on the glasses raises suspicion. The reflection captures an image within an image—a characteristic typically achievable only through a camera lens and not through human facial features. This suggests that the reflection was artificially imposed onto the glasses, indicating potential deepfake manipulation. Additionally, the way the glasses interact with light appears slightly over-emphasized, possessing a sheen that is stronger than could be naturally expected for the material used in the lenses and frame. Furthermore, the skin texture around the cheeks and chin appears unnaturally smooth. Facial skin, particularly in high-resolution images, manifests micro-variations in texture due to natural pore structure and imperfections, which are visibly absent here. This smoothness, while subtle, hints at digital smoothing often employed to create a more aesthetically pleasing but artificially constructed appearance. Close scrutiny of the clothing texture also reveals inconsistencies—rarely is denim fabric rendered such that it is entirely uniform and seemingly airbrushed, which calls into question whether the fabric is genuinely captured or digitally imposed. **\</reasoning\>**
**\<reflection\>** But wait, there's more to consider: the coherence between the subject and the background requires attention to perspective and depth. The clarity of the person is unusually high, yet it appears slightly exaggerated. Usually, in a casual indoor setting like this, there would be multiple planes of focus to add to the three-dimensional realism, such as a slight blur to the background or slight distortion at the edges of perspective frames. In this image, the focus is unnaturally sharp despite the supposed ambient lighting, presenting an optical clarity commonly used in manipulated images where the subject is meant to stand out unnaturally from the setting. This strategic focus discrepancy likely serves to highlight the subject's face, drawing attention away from the inconsistencies. **\</reflection\>**
**\<conclusion\>** Despite an impression of authenticity at first glance, the abnormal reflections, extreme smoothing of the facial area, and subtle signs in fabric textures provide compelling evidence of AI generation. The image is convincingly realistic but subtly uncanny under detailed scrutiny. These nuanced artifacts reveal its digital fabrication, subtly illustrating how advanced AI techniques are capable of creating deepfakes that require sophisticated examination to identify. Thus, the image serves as an exemplar of medium-hard to detect forgery—a regular photograph that cleverly masks its artificial origins behind strong realism. **\</conclusion\>**
**\<answer\>** fake **\</answer\>**

Figure 14: Reasoning output of our **VERITAS** model.

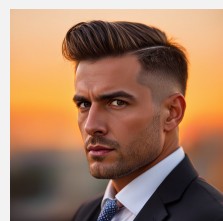 (Fake Image generated by **VAR**, using Infinity)

**<fast>** At first glance, the image looks convincingly realistic, displaying no apparent signs of forgery based purely on presentation. **</fast>**

**<planning>** Given this impression, a systematic examination is essential to identify potential evidence of AI-generation. My approach divides the analysis into three key areas: first, verifying structural realism including proportions and alignment; second, inspecting for subtle visual anomalies like edge definition and skin texture inconsistencies; and lastly, evaluating adherence to realistic lighting and environmental effects, both foreground and background. This comprehensive strategy helps pinpoint discrepancies that might be missed with a cursory inspection. **</planning>**

**<reasoning>** Observing the structural elements, **I notice the lighting conditions provide a viewpoint challenge—especially the striking intensity from the sunset**, which is visually arresting but physically implausible. This lighting dramatically shadows most of the subject's face, excluding any subtle gradations across the facial planes. Realistically, such lighting would create a variance in shadows and highlights that reflect the angle and placement of the light source accurately, yet here, there's a homogeneity within the shadow that feels artificially imposed, suggesting digital manipulation.
Moving to closer details, **the angle of the hair strands near the forehead is inconsistent**. The way light weaves through and between the strands is unnaturally precise, causing them to appear levitatively placed rather than naturally integrated with the head. The hair parting, while elegantly designed, lacks the gentle uncertainty typically seen when light falls over hair strands in a photograph—instead it appears cut rather than flowing. Such precision in grain and texture often suggests digital rendering rather than an unmediated capture of light and shadow.
Additionally, **the blurriness around the background** elements, **especially behind the suit collar**, **is unusually smooth**, which seems exaggerated to reduce focus disparity and possibly distract from deficiencies in detail. This suggests technical anomalies common in generated imagery where background elements do not align accurately with realistic depth of field. **</reasoning>**

**<reflection>** Upon re-evaluating the image again, my attention is drawn to the suit fabric, which is excessively defined with sharpness, resonating a sense of triumph, while the way it drapes over the shoulders and the creases is unnaturally perfect. In a actual photograph, it might possess some imperfections or subtle distortions due to actual body movement, which are absent here, suggesting an artificial resolution map. This combined with the harmonized lighting—although visually pleasing—can be indicative of **tooling typically managed by advanced AI algorithms**, **trying to mimic realism beyond human capability**. **</reflection>**

**<conclusion>** Despite a strong initial depiction of realism in texture and clothing detail, the lighting's implausibility, hair angle irregularities, and background blur together betray the image as AI-generated. These subtle discrepancies, infrequent in everyday photographic capture, point to the image being medium-hard to detect counterfeit, requiring deeper scrutiny to reveal its likely synthetic origins. Analysis like this underlines the importance of a meticulous inspection method for uncovering hidden deepfake artifacts effectively. **</conclusion>**

**<answer>** **fake** **</answer>**

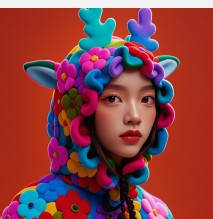 (Fake Image **in the wild**, using Dreamina)

**<fast>** Initially, the image appears quite realistic, with a composition that adheres to natural proportions and fitting colors, necessitating a deeper inspection to uncover its true nature. **</fast>**

**<planning>** Since the initial examination does not suggest any obvious abnormalities, a structured forensic analysis is required. This approach involves moving through the elements step-by-step, starting from identifying structural coherence, checking for subtle visual artifacts, and examining the image for adherence to real-world physics, starting with textures, lighting, and color consistency. This layered analysis will help detect the subtle indications of a deepfake. **</planning>**

**<reasoning>** Upon inspecting the image further, the initial mention plays out well in terms of facial alignment and proportions, which aligns with typical human alignments, yet offers no immediate conflict. Switching focus to the textures and details of the costume reveals a conspicuous issue: the portions of the costume extending over the right shoulder present an unnatural, undefined texture. **The costume is described as colorful and fabric-like, yet these portions lack typical textile characteristics**, appearing instead as amorphous blobs of color with no evident border or stitching. This is a clear deviation from genuine textile patterns, indicating AI-generated artifacts often resultant from surface texture synthesis errors. Furthermore, **the costume displays an overemphasized level of saturation and sharpness**, with **exaggerated color contrasts that defy realistic fabric photography**, common in AI-generated images aiming for exuberant detail. The lighting on the costume is also particularly uneven; the vibrant colors of the costume clash with the flat tone of the intensely red background, disrupting logical shadow casting, an indication of artificial overlay. Additionally, the surreal aesthetic of the clothing hints at a stylized rendering typical in AI-generated outputs rather than a real-world photo capture. **</reasoning>**

**<reflection>** On reflecting further, it becomes evident that the costume does not blend into plausible three-dimensional space, displaying exaggerated, bright regions that lack realistic depth and shading. The entire ensemble exudes an **almost "too perfect" or gaudy aesthetic, almost like a child's drawing**, which raises suspicion about its authenticity. Such exaggerated color saturation and surface texture inconsistencies are critical indicators that the image might be AI-generated. Despite the image's initial realistic facade, the elements betray any semblance of a natural photograph, pointing to AI involvement in rendering these costume details. Such discrepancies are clarifying indicators of deepfake manipulation, requiring close scrutiny to discern. **</reflection>**

**<conclusion>** Despite the meticulous attention to realistic proportions and initial color harmony, the costume's unnatural texture, disproportionate saturation, and unrealistic lighting clearly signpost it as AI-generated. These issues, especially prominent within the costume details, defiantly announce its artificial manipulation. Thus, the image, inherently deceptive at first glance, reveals upon comprehensive analysis its technological origins, confirming its deepfake nature and characterizing it as 'medium hard to detect' due to the sophisticated presentation initially masks its forged elements. **</conclusion>**

**<answer>** **fake** **</answer>**

Figure 15: Reasoning output of our **VERITAS** model.

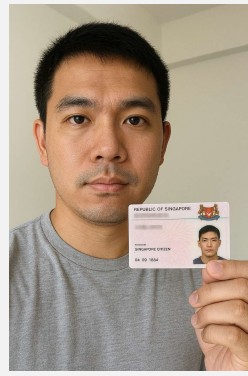

(Fake Image **in the wild**, generated by GPT-4o)

**\<fast\>** At first glance, the image appears convincingly real, with no immediate signs of forgery. **\</fast\>**
**\<planning\>** Given the initial impression of authenticity, a systematic and detailed inspection is necessary. This involves evaluating structural coherence, searching for subtle artifacts that may indicate AI generation, and verifying the physical plausibility of elements such as lighting, texture, and proportions. This multi-layered approach is essential to uncover any hidden inconsistencies that are not immediately apparent. **\</planning\>**
**\<reasoning\>** Upon closer inspection, the facial features and proportions seem consistent with a real human face, showing no obvious structural distortions. However, when analyzing the details on the Singapore Citizen card, there are some inconsistencies. The **date of birth on the card reads "04 09 1884,"** which is not only invalid due to the day being too large but also historically implausible, as **Singapore did not exist as a sovereign state in 1884**. This suggests a manipulation or error in the generation process. Additionally, the skin tone of the face differs slightly from the skin tone on the card. The lighting on the face appears slightly off, with shadows that don't align perfectly with the apparent light source, suggesting possible digital alteration. The texture of the skin on the card is also unusually smooth compared to the natural texture of the face, which could be a result of AI smoothing techniques used in generation. **\</reasoning\>**
**\<reflection\>** Wait a moment—there's another point to consider. The background is unusually plain and lacks the depth of field typically seen in real photographs, which might be an oversight in the image generation process. **\</reflection\>**
**\<conclusion\>** Despite the image's initial convincing realism, the inconsistencies in the date of birth, skin tone mismatch, lighting discrepancies, and unnatural hand positioning collectively suggest that the image is AI-generated. These elements, while not immediately obvious, reveal the image's artificial nature upon detailed scrutiny, confirming it as a medium-hard to detect forgery. **\</conclusion\>**
**\<answer\> fake \</answer\>**

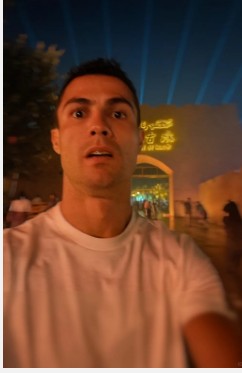

(Fake Image **in the wild**, generated by GPT-4o)

**\<fast\>** At first glance, the image appears unusually distorted and does not conform to realistic facial feature expectations, suggesting it could be AI-generated. **\</fast\>**
**\<planning\>** **Given the extreme blurriness** and odd shape, **a structured analysis is necessary**. I'll begin with an assessment of overall structural integrity, move to subtle synthesis of feature consistency, and ensure all elements align within expected physical and geometrical constraints. Due to the image's poor resolution, even minor details may be misleading, demanding careful scrutiny to distinguish genuine artifacts from image corruption. **\</planning\>**
**\<reasoning\>** The face shape is severely distorted and lacks typical human facial characteristics—no discernible structure of the eyes, nose, or mouth is evident, indicating likely artificial generation. The basic outline resembling a face seems off, with proportions that appear artificial. This misalignment strongly suggests manipulation. Furthermore, **the background elements, like the structure with text and plants, lack clear definition in comparison to the face**, raising suspicion that the face was digitally altered to seem integral to the scene, though it does not visually reconcile with the expected geometry of human features. The lighting and focus discrepancies across the image underline inconsistencies typical of AI errors, **as the face is both blurred and maintains unclear outlines simultaneously**, which is physically implausible.
Upon close inspection, there are no textures or details that should be present in a high-quality photograph, an apparent shortcoming often linked with deepfake generation. Additionally, the **entire image shares an overall blurriness, potentially hinting at an attempt to mask imperfections**, thus further suggesting the face is not naturally part of this setting.
Further reflections on lighting suggest the shadows do not align with any feasible light source, deepening the suspicion of digital alteration. This lighting inconsistency, while subtle, is vital in identifying the strangeness of deepfake nuances, reinforcing other signs of forgery. **\</reasoning\>**
**\<conclusion\>** Given the severe distortion, lack of recognizable features, and lighting inconsistencies, this image exhibits clear signs of being AI-generated. These issues, coupled with background mismatches and the odd coherence between elements, confirm it as a case of digital manipulation, despite the attempt to blend naturally. Thus, it is confidently assignable as fake, showcasing typical hallmarks of medium-hard deception challenges that become evident through detailed analysis. **\</conclusion\>**
**\<answer\> fake \</answer\>**

Figure 16: Reasoning output of our **VERITAS** model.

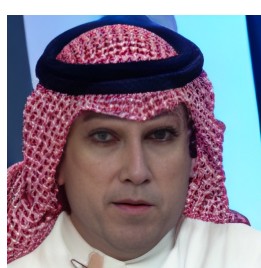

**Fake** Image from **CodeFormer**,
Face restoration
(originally face-swapped image)

**M2F2-Det (7B):** ✓

\<s\> The image is **manipulated** and does not appear naturally. **The person has mismatched beard.**\</s\>

**FakeVLM (7B):** ✓

This is a **fake** image. **The person's mouth looks a bit fake** because the mouth looks too rigid to convey expressions. The person has blurry hairlines and mismatched beard. **The person's eyebrows look a bit fake** because the eyebrows do not match the face's structure. **The person's eyes look very fake** because the eyes are placed at the wrong place on the face. The eyes look too rigid and lifeless to convey expression. The person's skin looks very fake because the skin is overly smooth and lacks details, such as skin texture around the chin. The lighting in the image is inconsistent, with unrealistic shadows. **The person's nostrils look a bit fake** because they are blurry and pixelated. ...

**SIDA-13B-description:** ✓ **(partially correct)**
**Type: part tampered**
**Areas:\<tampered position\>** On the beard and mustache of the man, specifically around the mouth and chin area, located in the central part of the image.
**Tampered Content:\<Types of objects or parts\>** facial hair, particularly the beard and mustache
**Visual Inconsistencies:\<lighting\>** The lighting on the tampered area does not match the natural lighting on the rest of the face\<**edges**\> The edges around the tampered area are unnaturally sharp and distinct\<**resolution**\> The resolution of the tampered area is slightly different from the surrounding areas\<**shadows**\> Shadows and reflections in the tampered area do not align with the light source affecting other parts of the face\<**boundary**\> The boundary of the tampered area shows signs of blending or feathering, indicating digital manipulation.
**Summary:** The given image is {part tampered} based on the above analysis.

**FakeShield (22B):** ✓
**1. Whether the picture has been tampered with / Description of the tampered area:**
The picture **has been tampered with**, specifically in the **central region covering the subject's face** and extending slightly into the upper garment area.
**2. Judgment basis:**
The tampered area shows a lack of consistent texture and pattern that is present in the surrounding garment area.
The edges of the tampered area are unnaturally smooth and do not align with the natural folds and creases expected in the fabric of the garment.
Additionally, there is a discrepancy in the lighting and shadows, as the tampered area lacks the depth and shadow consistency seen in the rest of the image.

**FFAA (7B):** ✓
**Image description:** The image shows a human face with a head covering, likely a traditional Middle Eastern outfit. The lighting is even, and the resolution is clear. The face appears to be male, middle-aged, with light skin. The expression is neutral, and there are no visible facial decorations. The face is oriented directly towards the camera.
**Forgery reasoning:** Upon close inspection, there are several abnormalities. The skin texture appears unnaturally smooth, especially around the eyes and mouth. **The eyes seem slightly misaligned** and exhibit an unusual sharpness compared to the rest of the face. The blending of the face with the head covering shows some inconsistencies, particularly around the edges where the skin meets the fabric. The overall integration of the facial features with the surrounding environment appears slightly off, suggesting manipulation.
**Analysis result: fake, Forgery type: Facial attribute manipulation**
**Match score:** 0.9953; **Difficulty:** hard

**Veritas (ours, 8B):**
**\<fast\>** Initially, the image does not appear distinctly fake, requiring closer inspection for confirmation of forgery. **\</fast\>**
**\<planning\>** Given the initial semblance of realism, a meticulous examination strategy is essential. This involves analyzing structural consistency, detecting any subtle artifacts indicative of AI-generated images, and checking whether the physical attributes, such as lighting and shadows or facial alignment, conform to realistic expectations. Such a detailed approach allows for revealing discrepancies often overlooked in less rigorous inspections. **\</planning\>**
**\<reasoning\>** At first glance, the image maintains a coherent overall structure without glaring inconsistencies. However, upon closer inspection, there is an **asymmetry detectable between the eyes**. The **left eye notably displays an indistinct appearance, lacking clarity and definition**, which is a common challenge in deepfake creation where aligning eye features can be difficult. **The pupils appear as faint lines rather than distinct round shapes**, suggesting potential issues. This muddled feature could be a result of AI struggle in generating fine details, especially in high-resolution images where realistic eye rendering is necessary. Furthermore, the nasal area presents some blurring, which prompts suspicion as genuine images typically maintain sharper textures and defined outlines. This blurriness might indicate areas where the generator struggled to maintain clarity or accurately reflect natural focus. Another point to note is the texture of the fabric near the face, which seems unnaturally smooth. Fabric in high-resolution photography usually shows more intricate and varied textures. The uniformity here could point towards AI over-smoothing artifacts during generation. **\</reasoning\>**
**\<conclusion\>** Although initially presenting a semblance of authenticity, the detailed scrutiny uncovers subtle inconsistencies such as the eye's asymmetry and blurriness around the nose. These artifacts are not glaring at first sight but confirm the image as AI-generated upon detailed inspection. The uniform lighting and lack of detailed clarity further support this conclusion, aligning with known pitfalls of deepfake technology. This image, thus, classifies as a medium hard to detect forgery, demonstrating the necessity for careful examination to identify its artificial nature. **\</conclusion\>**
**\<answer\> fake \</answer\>**

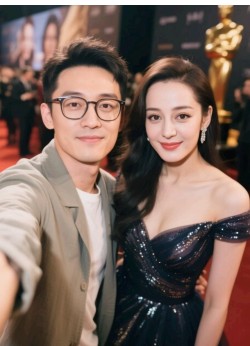

**Fake** Image from **GPT-4o**,
Entire face generation

**M2F2-Det (7B):** ✗

\<s\> The image is **authentic**.\</s\>

**FakeVLM (7B):** ✓

This is a **fake** image. The image **exhibits underlying characteristic inconsistencies in its features that suggest it is artificially created**, despite the absence of obvious artifacts or direct signs of composition.

**SIDA-13B-description:** ✗
**Type: real**
**Consistency:\<lighting\>The lighting in the image is consistent** with a single overhead light source illuminating the subjects uniformly.
\<**edges**\>The edges around the subjects and objects are smooth and continuous with no signs of tampering.\<**resolution**\>The resolution is high with clear details visible in the subjects' faces and clothing.\<**shadows**\>The shadows and reflections in the image are consistent with the light source and the physical presence of the subjects.\<**material properties**\>The materials such as the fabric of the woman's dress and the man's shirt interact with light in a realistic manner.\<**perspective**\>The perspective and scale of the subjects are realistic with appropriate depth of field and focus.
**Authenticity:\<natural imperfections\>**The image includes natural imperfections like slight wrinkles in the woman's dress and the man's casual attire.\<**environmental interaction**\>The subjects interact naturally with their environment with appropriate positioning and engagement.\<**temporal consistency**\>The attire and hairstyles are contemporary and appropriate for a formal event.\<**cultural**\>The attire and setting are culturally coherent with a formal event in a modern setting.\<**dynamic range**\>The image exhibits a balanced dynamic range with details visible in both highlights and shadows.
**Final Assessment:** The given image is {real} based on the above analysis.

**FakeShield (22B):** ✗
**1. Whether the picture has been tampered with / Description of the tampered area:**
The picture **has not been tampered with**. It appears to be a genuine photograph taken directly by a camera.
**2. Judgment basis:**
The **lighting in the image is consistent**, with the direction of light affecting both individuals in a coherent manner, casting shadows that align with the light source. The reflections on the surfaces, such as the shine on the woman's dress and the man's glasses, correspond to the environment and lighting.
The edges of the subjects and objects within the image are smooth and continuous, with no signs of unnatural pixel distribution or abrupt edges that would suggest compositing.
The resolution across the image is uniform, without any areas of blurring or pixelation that would indicate manipulation.
The perspective and scale relationships are realistic; the size of the subjects corresponds to their position in the frame, and the depth of field is consistent with the focus on the subjects while slightly blurring the background.
The shadows cast by the subjects are appropriate for the lighting conditions, and there are no unnatural shadows or reflections. There is no visible text or logos to assess for tampering.
Lastly, nothing in the image violates physical laws; all elements are in plausible positions and orientations.

**FFAA (7B):** ✓
**Image description:** The image shows a young woman with fair skin, dark hair, and brown eyes. She is smiling slightly, and her face is well-lit with even lighting. She is wearing earrings and has a smooth complexion. The resolution of the image is high, and the face is oriented directly towards the camera.
**Forgery reasoning:** Upon close examination, there are several indicators of facial attribute manipulation. **The skin appears overly smooth and lacks natural texture**, which is often a sign of digital alteration. The eyes and lips seem slightly unnatural in their sharpness and clarity compared to the rest of the face. **There are also subtle inconsistencies in the skin tone around the eyes and mouth, suggesting that these areas have been digitally modified**. The integration of the face with the hair and earrings appears mostly natural, but the overall smoothness and slight unnaturalness in facial features indicate manipulation.
**Analysis result: fake, Forgery type: Facial attribute manipulation**
**Match score:** 0.8276; **Difficulty:** hard

**Veritas (ours, 8B):**
**\<fast\>** At first glance, the image appears authentic, with no immediate signs of manipulation. **\</fast\>**
**\<planning\>** Given the absence of obvious anomalies, a structured forensic analysis is necessary. The approach involves examining structural integrity, identifying subtle generator artifacts, and verifying physical consistency, especially focusing on lighting and texture realism. This multi-tiered strategy ensures that even the most subtle signs of AI generation are uncovered. **\</planning\>**
**\<reasoning\>** The image initially presents a coherent and realistic portrayal, with facial proportions and expressions aligning with natural human features. However, upon closer inspection, the lighting on the faces is inconsistent. **The man's face appears brighter, suggesting a different light source or angle than the woman's**, which is more subdued. This discrepancy could be due to separate lighting setups during image generation or post-processing errors. Additionally, the texture of the woman's skin, particularly around the neck, is unnaturally smooth and lacks the natural variation seen in real skin, which might indicate AI smoothing effects. **The sequins on her dress also appear overly uniform** and lack the depth and randomness expected in genuine fabric, hinting at digital generation. The background, while appropriately blurred for a red carpet setting, exhibits slight pixelation and lacks the sharpness expected in high-resolution photography, potentially pointing to a composite origin. **\</reasoning\>**
**\<reflection\>** Wait, let's consider the broader context. The lighting inconsistency between the two subjects, while subtle, is a red flag in a scene where the lighting should be uniform. The **AI might have struggled to maintain consistent illumination across different facial planes**. Moreover, the subtle smoothing and uniformity in the dress's texture are common AI errors, where the generator tries to create an idealized version of reality but fails to replicate the complexity of real-world materials. The background blur, while realistic, is slightly off in terms of focus gradient, which could be an artifact of AI blending techniques. These elements, when considered together, suggest a deeper layer of artificiality not immediately apparent. **\</reflection\>**
**\<conclusion\>** Despite its convincing facade, the image contains subtle yet telling inconsistencies in lighting, texture, and background realism. These artifacts, while not glaring, collectively suggest AI generation. The nuanced nature of these errors places the image in the category of a medium-hard to detect forgery, requiring careful observation and analysis to identify its artificial origins. **\</conclusion\>**
**\<answer\> fake \</answer\>**

Figure 17: Explanation comparisons between **VERITAS** and existing MLLM-based detectors.

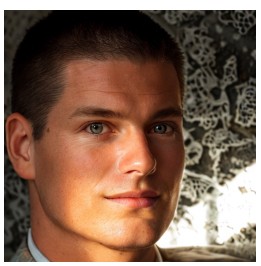

**Fake** Image from **IC-Light,**
**Face re-lighting**
**(fully synthesized)**

**M2F2-Det (7B):** ✅
 The image appears to be a **real photograph** of a person. The person has smooth skin, green eyes, and a haircut that is very neat and tidy. The person is wearing what looks like a military or formal uniform, which could suggest a profession or event that requires a certain dress code. The background has a textured pattern that could be a wall or a backdrop for a photo shoot.

**FakeVLM (7B):** ✅
This is a **fake** image. The image **exhibits underlying characteristic inconsistencies in its features that suggest it is artificially created**, despite the absence of obvious artifacts or direct signs of composition.

**FakeShield (22B):** ❌
**1. Whether the image has been tampered with/description of the tampered area:**
The image appears **authentic** and there are **no visible signs of tampering**. The overall quality and details suggest that it is a direct capture from a camera.

**2. Judgment basis:**
The visible portion of the image shows natural lighting and shadow effects on the ear and neck, which are consistent with the direction and quality of light that one would expect in an unaltered photograph.
The skin texture on the neck and the visible part of the ear looks natural, with appropriate shading and highlights that correspond to the contours of the skin.
The hair also appears to have a natural flow and texture, without any signs of digital alteration.
If this image were placed among deepfake tampered images, it could be recognized as authentic by analyzing the **consistency of the lighting, the natural skin texture, and the absence of any distorted or blurred edges** that are often telltale signs of deepfake manipulation.

**SIDA-13B-description:** ✅ **(partially correct)**
**Type:** part tampered
**Areas:<tampered position>** The tampered area is located on the upper left quadrant of the image, specifically on the upper left part of the man's face and hair, extending slightly towards the right.
**Tampered Content:<Types of objects or parts>** The man's hair and upper left part of his face
**Visual Inconsistencies:<lighting>** The lighting on the tampered area does not match the natural lighting of the rest of the image, showing a different hue and intensity**<edges>** The edges of the tampered area are unnaturally sharp and distinct, contrasting with the softer edges of the rest of the image**<resolution>** The resolution of the tampered area is slightly different, showing a lesser level of detail compared to the rest of the image**<shadows>** Shadows and reflections in the tampered area do not align with the light source and other shadows in the image**<boundary>** The boundary of the tampered area is blurred and unnaturally smooth, lacking the typical texture and detail seen in the rest of the image
**Summary:** The given image is part tampered based on the above analysis.

**FFAA (7B):** ❌
**Image description:** The image shows a close-up of a male face with fair skin, short brown hair, and blue eyes. The lighting is bright and appears to come from the left side of the image, casting shadows on the right side of the face. The resolution is high, and the face is oriented slightly to the right. The expression is neutral, and there are no visible face decorations.
**Forgery reasoning:** The facial features appear natural and well-integrated with the surrounding environment. The skin texture is consistent and shows natural imperfections, such as pores and slight wrinkles. The lighting and shadows are coherent with the light source on the left. **There are no visible artifacts, blurring, or pixelation around the facial boundaries or features.** The eyes, nose, and mouth are proportionate and aligned correctly. The hairline and hair texture also appear natural and blend seamlessly with the face. There are no signs of tampering or manipulation in terms of facial attributes or expressions.
**Analysis result:** real, **Forgery type:** None
**Match score:** 0.6767; **Difficulty:** hard

**Veritas (ours, 8B):**
**<fast>** At first glance, the image appears realistic due to its high resolution and clear depiction of fine details, making it difficult to identify as fake immediately. **</fast>**
**<planning>** Since no clear visual flaws are present initially, a structured examination is necessary, focusing on the structural integrity, subtle artifact analysis, and adherence to physical and optical realism. This methodical approach helps in uncovering less obvious discrepancies indicative of AI generation. **</planning>**
**<reasoning>** Structurally, the face exhibits normal proportions with no visible asymmetry or mismatched features, which typically delineates a real human face. However, the lighting on the face is highly suggestive of a natural light source, yet the background lacks a corresponding light source that should logically cast similar shadows and highlights. This lighting inconsistency is unusual and suggests potential AI manipulation. Upon closer inspection, the backdrop presents a peculiar and ornate pattern that strongly resembles hand-drawn or semi-artistic rendering rather than the usually expected photographic background. This could imply that the background was artificially rendered separate from the subject, a frequent error in AI-generated images known as 'cut and paste' artifacts. The lighting on the subject does not align with the lighting seen in the patterned background, indicating potential manipulation. Additionally, the shadows and highlights on the face, while seemingly real in isolation, collectively create an overly dramatic and stylized appearance. The image's striking quality, though impressive, treads on the edge of theatrical emphasis, which is a hallmark of AI generation aiming to simulate depth and dimension but inadvertently overdoing it. This stylized realism can betray an otherwise authentic-looking composition. **</reasoning>**
**<conclusion>** Upon detailed evaluation, the lighting inconsistencies with the background, the semi-artistic rendering of the backdrop, and the exaggerated shadowing on the subject suggest an AI-generated image. These subtle yet telling errors cumulatively indicate medium-hard forgery detection, highlighting these as the central deceptive elements within this seemingly authentic image portrayal. **</conclusion>**
**<answer> fake </answer>**

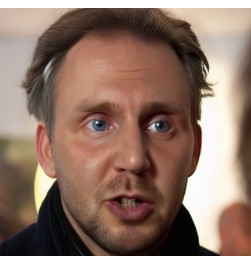

**Fake** Image from **FaceAdapter,**
**Generative face swapping**

**M2F2-Det (7B):** ✅
 The image appears to be a **manipulated or edited photograph. The person has mismatched beard.** The person's face doesn't match the overall scene.

**FakeVLM (7B):** ✅
This is a **fake** image. The image **exhibits underlying characteristic inconsistencies in its features that suggest it is artificially created**, despite the absence of obvious artifacts or direct signs of composition.

**FakeShield (22B):** ✅
**1. Whether the picture has been tampered with / Description of the tampered area:**
The picture **has been tampered with**, specifically in the **central region of the image**, covering the **area where the subject's face would be**. The tampered area extends from the top of the subject's head to the chin, and from ear to ear.

**2. Judgment basis:**
The tampering is evident due to the presence of **a solid color block that obscures the facial features**, which is not a natural occurrence in photography.
The edges of the tampered area are unnaturally straight and do not conform to the natural contours of a face. Additionally, **the tampered area lacks any texture or detail that would be consistent with the rest of the image**, such as skin texture, hair, or clothing fabric.
The lighting on the tampered area does not match the lighting on the rest of the subject, as there are no shadows or highlights that would suggest depth or contour.

**SIDA-13B-description:** ✅ **(partially correct)**
**Type:** part tampered
**Areas:<tampered position>** On the face of the individual, specifically affecting the eyes, located in the center of the image, slightly towards the left side.
**Tampered Content:<Types of objects or parts>** The eyes of the individual
**Visual Inconsistencies:<lighting>** The eyes appear unnaturally bright and overly illuminated, contrasting with the surrounding facial features**<edges>** The edges around the eyes are unusually sharp and distinct, lacking the natural blending seen in untouched images**<resolution>** The eyes exhibit a higher resolution and clarity compared to the rest of the face, indicating potential digital manipulation**<shadows>** The shadows under the eyes do not align with the light source, suggesting artificial manipulation**<boundary>** The boundaries around the eyes are overly defined, lacking the natural blending seen in untouched images
**Summary:** The given image is {part tampered} based on the above analysis.

**FFAA (7B):** ❌
**Image description:** The image shows a close-up of a man's face with light skin, blue eyes, and short, light-colored hair. The lighting is bright and even, highlighting the facial features clearly. The face is oriented towards the camera, with a neutral expression and slightly parted lips. The resolution is high, and the details of the skin texture, facial hair, and other features are clearly visible.
**Forgery reasoning:** Upon close examination, **there are no visible abnormalities in the facial organs.** The eyes, nose, and mouth appear natural and well-integrated with the rest of the face. The skin texture is consistent and detailed, with no signs of blurring, pixelation, or unnatural smoothness. There are no fusion artifacts or inconsistent skin colors at the boundaries of the face. The facial hair and other features appear natural and consistent with the lighting and surrounding environment. There are no signs of tampering or manipulation in terms of facial attributes or expressions.
**Analysis result:** real, **Forgery type:** None
**Match score:** 0.8756; **Difficulty:** hard

**Veritas (ours, 8B):**
**<fast>** At first glance, the image seems to have distinct visual inconsistencies that suggest it may be AI-generated. **</fast>**
**<planning>** Given the initial suggestion of forgery, a systematic investigation is essential. This includes examining structural elements like facial alignment and symmetry, identifying subtle textural anomalies, and evaluating overall realism through a logical progression of analysis from obvious to more nuanced details. By doing so, I can uncover the depth of manipulation hidden within the image. **</planning>**
**<reasoning>** Starting with the structural composition, the asymmetry is starkly evident, particularly in the eyes. The right eye appears more defined than the left eye, which is notably blurry and misaligned with the other facial features. This stark difference in eye detailing disrupts natural facial balance and coherence. Typically, facial features mirror each other in alignment and clarity unless intentionally distorted, which isn't the case here, suggesting a composite or synthetic assembly. Furthermore, the structure and proportion of the mouth are inconsistent with typical human anatomy; it appears to be distorted, lacking symmetry and showing a jagged alignment. This anomaly suggests that the facial area may have undergone digital manipulation or deepfake techniques that failed to maintain anatomical accuracy. **</reasoning>**
**<reflection>** Reflecting upon further, I notice the subtle 'jagged' quality around the hairline doesn't align perfectly with natural strands, which typically have smoother transitions, indicative of AI manipulation. This inconsideration in strand details hints at a lack of artificial realism, where meticulous attention to detail falters during execution. It's these finer points that betray the digital artifice. Additionally, the slight misalignment in shadow placement seems almost exaggerated, likely to attempted realism in shadows but failing to embody the natural gradation typically seen. **</reflection>**
**<conclusion>** Despite the image attempting to convey realism, the glaring structural inconsistencies, particularly in the eyes and mouth, combined with unnatural proportions and lighting errors, prominently indicate artificial generation. These issues are easily detectable, classifying the image as 'easy' to identify as a fake. The synthesis of these clear indicators confirms the image as decidedly AI-generated, highlighting typical markers associated with deepfake techniques. This detailed analysis reveals how the image's structural and textural inaccuracies depict its artificial nature conspicuously. **</conclusion>**
**<answer> fake </answer>**

Figure 18: Explanation comparisons between **VERITAS** and existing MLLM-based detectors.

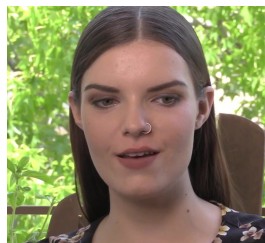

**Real** Image from **VFHQ**

**M2F2-Det (7B):** ✓

 The image appears to be an **authentic** photograph of a person. The person has distinctive features such as the nose piercing and the hairstyle, which make it likely that this is a real person in a real setting. The person is outdoors, as indicated by the natural lighting and the green foliage in the background.

**FakeVLM (7B):** ✓

This is a **real** image. The person exhibits natural skin textures, including gradual tone transitions and microscopic skin imperfections; the nostril geometry is natural with asymmetric shape and pore-level detail; the eye reflection shows physically accurate light interactions in the cornea and conjunctiva; the lip texture has visible striations with a natural moisture gradient; …

**SIDA-13B-description:** ✗

**Type:** part tampered
**Areas:<tampered position>** On the woman's face, specifically around her nose and upper cheeks, located in the central part of the image.
**Tampered Content:<Types of objects or parts>** The woman's nose and upper cheeks
**Visual Inconsistencies:<lighting>** The lighting on the tampered areas does not match the natural lighting on the rest of the face**<edges>** The edges around the tampered areas are unnaturally sharp and distinct**<resolution>** The resolution in the tampered areas is slightly different from the rest of the image**<shadows>** Shadows and reflections in the tampered areas do not align with the light source affecting other parts of the face**<boundary>** The boundaries of the tampered areas are overly defined and do not blend seamlessly with the surrounding skin
**Summary:** The given image is {part tampered} based on the above analysis.

**FakeShield (22B):** ✓

**1. Whether the picture has been tampered with / Description of the tampered area:**
The picture **has not been tampered with**. It appears to be a genuine photograph taken directly by a camera.

**2. Judgment basis:**
The lighting in the image is consistent, with the natural light source seemingly coming from the left side, as indicated by the soft shadows on the right side of the person's hair and the chair.
The colors in the image are coherent, with no signs of unnatural coloration that might suggest tampering.
The edges of the person and the chair are smooth and continuous, with no signs of pixel distortion or unnatural sharpness that could indicate manipulation.
The resolution across the image is uniform, without any areas of differing quality that might suggest a composite image.
The perspective is realistic, with the size and scale of the person and the chair appearing proportional and consistent with the scene.
**The depth of field is reasonable, with the background slightly blurred,** which is typical for a photograph taken with a camera focused on the subject.
There are no reflections or shadows that contradict the apparent light source.
There is no text or logos present to assess for tampering. Lastly, nothing in the image violates physical laws, such as gravity or the properties of light and shadow.
All these details support the conclusion that this image is real and has not been tampered with.

**FFAA (7B):** ✓

**Image description:** The image shows a close-up of a young woman with fair skin, straight brown hair, and light makeup. She has a nose ring, light-colored eyes, and is wearing light lipstick. The lighting appears natural, and the background is slightly blurred, suggesting an outdoor setting. Her facial expression is neutral, with a slight smile.
**Forgery reasoning:** Upon close examination, there are no visible abnormalities in the facial organs. The skin texture appears natural, with no signs of excessive smoothness or blurring. The facial features, including the eyes, nose, and mouth, are well-integrated and consistent with the rest of the face. There are no noticeable fusion artifacts or inconsistent skin tones at the boundaries of the face. The nose ring and makeup appear naturally placed and consistent with the lighting and shadows in the image. The hair and background also blend seamlessly with the face, indicating no signs of manipulation or forgery.
**Analysis result:** **real**, **Forgery type:** **None**
**Match score:** 0.9144; **Difficulty:** hard

**Veritas (ours, 8B):**

**<fast>** At first glance, the image appears authentic with no obvious signs of being AI-generated. **</fast>**
**<planning>** To confirm this initial impression, a structured forensic analysis is conducted across three progressive layers: visual structural integrity, subtle generation artifacts, and physical plausibility to ensure there are no hidden inconsistencies. **</planning>**
**<reasoning>** The facial structure is natural and symmetrical with consistent proportions and expression, showing no visual abnormalities. The skin tone and texture are uniform, with no artificial edges, blurring, or missing fine details like pores. Lighting and shadows align with a real-world setup, and the background context matches the subject's illumination, with no signs of stylization or optical inconsistencies. **</reasoning>**
<conclusion> Given the absence of any suspicious features across all examined aspects, the image is determined to **be authentic and likely a high-quality, real photograph.** </conclusion>
**<answer>** **real** **</answer>**

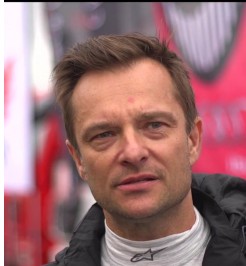

**Real** Image from **VFHQ**

**M2F2-Det (7B):** ✓

 The image **appears to be a photograph of a person.** The person has short hair, is wearing a black jacket, and has a noticeable red spot on their forehead.

**FakeVLM (7B):** ✓

This is a **real** image. The person exhibits natural skin textures, complete face features, and realistic facial expressions. The nostrils have a natural asymmetric shape with pore-level detail, and the skin texture shows gradual tone transitions with microscopic imperfections. …

**SIDA-13B-description:** ✗

**Type:** part tampered
**Areas:<tampered position>** The tampered area is located on the upper left quadrant of the image, specifically affecting the left side of the man's face and hair.
**Tampered Content:<Types of objects or parts>** The man's hair and the left side of his face
**Visual Inconsistencies:<lighting>** The lighting on the tampered area does not match the natural lighting of the rest of the image**<edges>** The edges around the tampered area are unnaturally sharp and distinct**<resolution>** The resolution of the tampered area is slightly different from the rest of the image**<shadows>** Shadows and reflections in the tampered area do not align with the light source affecting other parts of the image**<boundary>** The boundary of the tampered area shows signs of blending or feathering, indicating digital manipulation
**Summary:** The given image is part tampered based on the above analysis.

**FakeShield (22B):** ✓

**1. Whether the picture has been tampered with / Description of the tampered area:**
The picture **has not been tampered with**. It appears to be a genuine photograph taken directly by a camera.

**2. Judgment basis:**
**The lighting on the subject's hair and clothing is consistent,** with the direction of light coming from the upper left, casting natural-looking shadows to the right side.
The colors are coherent throughout the image, with no signs of unnatural color patches that could indicate tampering.
**The edges around the subject's hair and clothing are smooth and continuous,** with no signs of pixel distortion or unnatural sharpness that could suggest manipulation.
The resolution across the image is uniform, without any areas of differing quality that might indicate a composite.
The perspective is realistic, with the size of the subject's features diminishing naturally with distance, and the depth of field is consistent with what is expected from a camera's lens.
The shadows on the neck and clothing align with the light source, and there are no abnormal reflections or shadows that would suggest editing.
There is no visible text or logos to assess for tampering.
Lastly, nothing in the image defies physical laws; the subject's posture and the drape of the clothing all follow the expected behavior of real-world physics.

**FFAA (7B):** ✓

**Image description:** The image shows a middle-aged man with light skin, short brown hair, and blue eyes. The lighting appears natural, and the resolution is clear. The man's face is oriented towards the camera, and he has a neutral expression. There are no visible face decorations.
**Forgery reasoning:** Upon close examination, there are no noticeable abnormalities in the facial organs. **The skin texture appears natural, with visible pores and slight imperfections.** There are no signs of blurring, pixelation, or uncoordinated features. The skin color is consistent throughout the face, and there are no fusion artifacts or inconsistent skin tones at the boundaries. The facial features, including the eyes, nose, and mouth, appear natural and well-integrated with the surrounding hair and environment. There are no signs of tampering or manipulation in terms of facial attributes.
**Analysis result:** **real**, **Forgery type:** **None**
**Match score:** 0.8923; **Difficulty:** hard

**Veritas (ours, 8B):**

**<fast>** At first glance, the image appears authentic, but a deeper inspection is required to confirm its legitimacy. **</fast>**
**<planning>** To determine the authenticity, I will examine the image through three progressive layers: first checking for obvious structural anomalies, then analyzing for subtle generator artifacts, and finally evaluating for violations of physical plausibility. This structured approach ensures a comprehensive assessment. **</planning>**
**<reasoning>** Visually, the image shows no color mismatches, structural distortions, or unnatural expressions, and the proportions align with natural human features. Upon closer inspection, the edges, texture, and blending appear natural without signs of artificial generation. There is no missing high-frequency detail or abnormal blurring. **Even under deeper scrutiny, the lighting, shadows, and optical focus remain consistent with real-world physics,** and the makeup and context do not raise any red flags. The image does not exhibit any misleading artifacts that would suggest manipulation or synthetic generation. **</reasoning>**
**<conclusion>** Based on the absence of structural flaws, generator artifacts, and physical inconsistencies, the image is determined to be authentic. **</conclusion>**
**<answer>** **real** **</answer>**

Figure 19: Explanation comparisons between **VERITAS** and existing MLLM-based detectors.

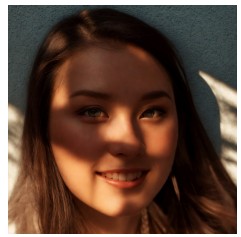 (Fake Image from **Face Relighting**, using IC-Light)

**Veritas ($s_l^\phi + s_l^\psi$):**

**‹fast›** On first inspection, the image appears realistic, but certain aspects seem slightly off, raising a subtle sense of unease about its authenticity. **‹/fast›**

**‹planning›** Given the lack of immediate and obvious flaws, a thorough examination is necessary. My analysis will focus on assessing the structural integrity of the facial features, checking for inconsistencies in texture and sharpness, and analyzing the physical plausibility of lighting and shadows. This multi-layered approach is vital for identifying any subtle characteristics of AI generation that are not apparent at first glance. **‹/planning›**

**‹reasoning›** Initially, the facial structure and proportions appear largely consistent with realistic human features, with symmetrical features and a generally natural appearance. However, upon closer inspection, some anomalies become apparent. The nose appears slightly off-center, which disrupts the natural balance expected in a truly symmetrical face, hinting at possible digital manipulation. The eyes, while seemingly natural, exhibit watery reflections that lack the depth seen in real human eyes, suggesting an artificial rendering. **Moving beyond the structural issues, let's examine the brighter parts of the face**, particularly **around the right cheek and jawline**. The **skin appears unnaturally smooth and slightly blurred**, creating a contrast with the sharper details of the hair around the face. This inconsistency in sharpness, where the hair is clearly defined while the skin lacks clarity, is common in AI-generated images and indicates possible blending errors during generation. Furthermore, **the shadows cast by the hair on the subject's forehead and neck seem too soft** and **less defined than the hair's clarity relative to the face**. These shadows should naturally match the strength and angle of the light source effecting the subject, but their inconsistency suggests poor modeling or rendering by the AI. Observing the overall lighting, there is a noticeable **lack of realistic shadow definition and depth when the lighting should naturally create more pronounced contrasts given the intensity of the shadow play**. The lighting gives the face a somewhat painted or airbrushed appearance, as if the AI tried to replicate the softness of impressionist painting rather than capture the nuanced, natural shadows and highlights of real photography.

Moreover, the setting and background, while seemingly consistent in tone and ambient lighting, also carry a slightly artificial aura that matches the subtle anomalies in facial features. This uniformity might be too perfect, hinting at a digitally integrated setting rather than a natural capture. The overall image feels a bit too harmonious, reinforcing the possibility of digital synthesis. **‹/reasoning›**

**‹conclusion›** Despite an initially convincing facade, the mismatched symmetry of facial features, the unnatural clarity and blur discrepancies, and inconsistent lighting and shadow modeling reveal the image's AI-generated origins. While these signs are not immediately obvious, the accumulation of these subtle anomalies strongly indicates that the image is a sophisticated but nonetheless artificially generated forgery. These findings confirm the image as medium hard to detect due to the subtlety of its artifacts, requiring careful observation and analysis to uncover its true nature. **‹/conclusion›**

**‹answer›** fake **‹/answer›**

**Veritas (w/o $s_l^\phi$):**

**Superficial & Less detailed**

**‹fast›** At first glance, the image appears realistic and does not exhibit any glaring inconsistencies that would immediately suggest manipulation. **‹/fast›**

**‹planning›** Given the lack of obvious structural issues, a thorough and systematic analysis is necessary. This involves evaluating the image for subtle visual anomalies, checking for logical inconsistencies in lighting and texture, and assessing the overall realism in terms of physical plausibility. This structured approach ensures that no detail is overlooked, especially since the image's authenticity is not immediately clear. **‹/planning›**

**‹reasoning›** Initially, the facial structure seems consistent with natural human proportions and symmetry, with no visible distortions or mismatches in color or expression. However, upon closer inspection, certain aspects become more apparent. **The lighting on the face appears unusually uniform and slightly too perfect**, lacking the minor variations typically seen in natural lighting conditions. This suggests the possibility of artificial enhancement. Additionally, the shadows cast by the hair and other elements on the face are slightly too defined and precise, which can be a sign of digital manipulation. The skin texture, while smooth, lacks the natural imperfections and fine details such as pores that are usually present in real human skin, hinting at a potential synthetic origin. Furthermore, the background and the way the light interacts with the subject seem harmonious, but this harmony can sometimes be a sign of advanced AI generation designed to mimic realistic settings. **‹/reasoning›**

**‹reflection›** Wait a moment, let's take a step back and consider the overall aesthetic. The image, while realistic, carries a polished, almost stylized look that can be characteristic of AI-generated images. **The uniformity in lighting** and the overly smooth skin are subtle indicators that may not be noticeable at first, but become more apparent upon deeper inspection. These elements, while not overtly artificial, deviate from the natural randomness and minor imperfections found in real-world photography. This subtle stylization, combined with the lack of high-frequency details like pores, raises further suspicion. **‹/reflection›**

**‹conclusion›** Although the image displays no obvious structural flaws and appears realistisc at first glance, the overly smooth skin texture, uniform lighting, and slightly too defined shadows suggest digital manipulation. These subtle anomalies, when considered collectively, indicate that the image is likely AI-generated. The need for detailed inspection and the presence of nuanced artifacts align with the characteristics of a hard-to-detect forgery, making this a challenging case that requires careful and comprehensive analysis to identify as fake. **‹/conclusion›**

**‹answer›** fake **‹/answer›**

Figure 20: Qualitative comparisons of the effects of non-preference $s_l^\phi$ in MiPO.

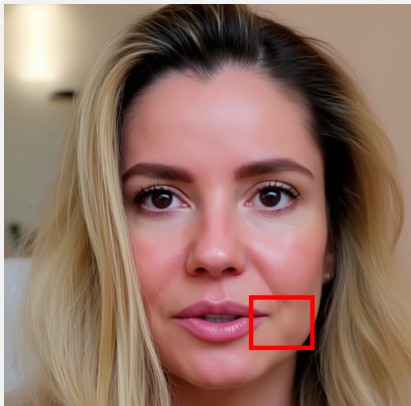

(**Fake** Image from **InfiniteYou**, face personalization)

**<fast>** At first glance, the image appears highly realistic, making it difficult to determine its authenticity without deeper inspection. **</fast>**
**<planning>** Given the initial impression of realism, a structured forensic analysis is necessary. The evaluation should progress from visual structure to subtle artifact detection and finally to an assessment of physical plausibility, ensuring that no potential anomaly is overlooked. This tiered approach allows for a comprehensive understanding of the image's authenticity. **</planning>**
**<reasoning>** Upon close inspection, the facial structure and proportions appear natural, with no obvious asymmetry or distortion. The lighting and shadows are consistent, and the background integrates seamlessly with the subject. However, subtle inconsistencies begin to emerge: the skin texture on the cheeks and forehead is unusually smooth, lacking the natural variation and fine detail typically seen in high-resolution photography. This smoothness suggests digital manipulation. Additionally, **there is a slight blurriness around the mouth** that doesn't match the sharpness of other facial features. The hairline also shows minor discrepancies in focus, further hinting at digital alteration. These artifacts, while subtle, are commonly associated with AI-generated images. **</reasoning>**
**<conclusion>** While the image passes as realistic at first glance, the accumulation of subtle artifacts——smooth skin texture, inconsistent focus, and slight stylization——strongly suggests it is AI-generated. These findings, though not immediately obvious, align with known characteristics of medium-hard to detect deepfakes, confirming the image as a sophisticated yet flawed forgery. **</conclusion>**
**<answer>** fake **</answer>**

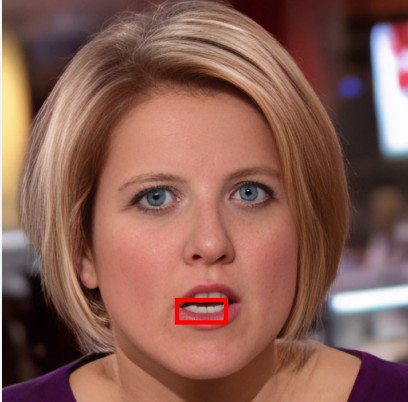

(**Fake** Image from **CodeFormer**, face restoration)

**<fast>** At first glance, the image appears highly realistic, with no immediate signs of artificial generation or manipulation. The facial structure, expression, and lighting all seem natural and consistent with a real human. **</fast>**
**<planning>** To assess whether the image is AI-generated, I will analyze it in three progressive stages: first identifying any visual structural abnormalities, then examining for subtle artifacts commonly introduced by AI generators, and finally checking for violations of physical realism that may not be immediately obvious but could indicate synthetic origin. This multi-layered approach ensures a thorough and cautious evaluation. **</planning>**
**<reasoning>** Upon closer inspection, the facial structure appears proportionate and symmetrical, with no distinct mismatch in alignment or proportions, which might initially suggest authenticity. However, subtle irregularities begin to emerge. There is a slight mismatch between the sharpness and clarity of parts of the face. The jawline appears slightly blurred when compared to the rest of the facial structure, suggesting a blending artifact. Furthermore, **the teeth near the mouth display an unusual blurriness**—this could indicate that there was a layering or compositing technique used in generating the image, making those parts less distinct than they naturally should be. Additionally, the transition between the hair and skin lacks typical complexity, appearing rather smooth, which might be due to texture smoothing techniques used in AI synthesis. The background and facial sharpness also diverge subtly from what would be expected in a single plane of focus, suggesting potential layering to simulate depth. **</reasoning>**
**<reflection>** Wait, upon further analysis, these inconsistencies could point towards a manipulation not immediately obvious, particularly in the difference in blurriness between the teeth and surrounding facial features. This slight discrepancy in clarity hints at potential digital adjustments commonly seen in AI-generated content where blending or feature reenactment is employed. What's more, the realism of the image and absence of typical AI artifacts in simpler images suggests that the generation process was of high fidelity but masked these telltale signs that, when examined closely, betray its synthetic roots. **</reflection>**
**<conclusion>** Considering the detailed examination, there are nuanced inconsistencies in focus and texture that, while subtle, align with known AI generation artifacts. These details—particularly the mismatch in clarity across facial features and minor blending issues—suggest that the image has been artificially generated. Despite its realistic appearance at first glance, the cumulative evidence reveals its digital nature, placing it firmly in the category of a hard-to-detect forgery requiring deep analysis to confirm. **</conclusion>**
**<answer>** fake **</answer>**

Figure 21: Illustration of model's capacity to perceive those barely noticeable artifacts. This shows certain advantages to human system.

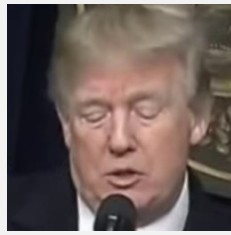

(**Real** Image from **DeepFaceLab**)

<fast> At first glance, the image appears authentic, with no immediate signs of manipulation or artificial generation. </fast>
<reasoning> Upon closer inspection, the facial structure and proportions appear consistent with those of a real human, with no obvious distortions or mismatches in color, expression, or alignment. The lighting and shadows are also physically plausible, and the background context matches the subject's appearance. However, subtle inconsistencies begin to emerge. The skin texture lacks the natural variation and fine detail typically seen in real faces. This uniformity suggests a potential artifact of AI generation. Additionally, there is a severe blurriness around the hair and eyes, which are critical for facial recognition and often challenging for generative models to render accurately. This blurriness could be an attempt by the generator to mask imperfections or blend different image components. </reasoning>
<conclusion> While the image initially appears authentic and passes basic checks for structural and physical realism, deeper analysis reveals subtle but telling signs of AI generation. The lack of natural skin texture variation, blurring around key facial features, and a stylized aesthetic collectively suggest that the image is likely AI-generated. These findings confirm the image as a hard-to-detect forgery, requiring careful and comprehensive evaluation to identify its artificial nature. </conclusion>
<answer> **fake** </answer>

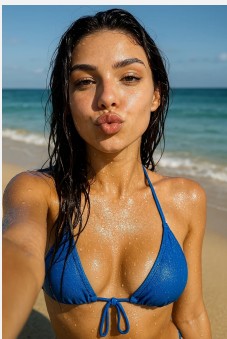

(**Fake** Image from **GPT-4o**)

<fast> At first glance, the image appears authentic, with natural facial features, realistic water droplets, and a coherent beach setting. </fast>
<planning> To determine the authenticity, we must proceed methodically: first evaluating obvious structural elements, then inspecting for subtle generator artifacts, and finally analyzing for violations of physical plausibility. Since no clear anomalies are visible, a deeper forensic approach is necessary. </planning>
<reasoning> The facial structure is symmetrical and proportionate, with consistent skin tone and natural expression, ruling out obvious manipulation. Upon closer inspection, the edges and textures appear smooth and realistic, with no signs of artificial blurring or missing high-frequency details like water droplets. The lighting and shadows are consistent with a natural beach environment, and the background context matches the subject's appearance. However, a subtle clue emerges: the water droplets on the skin and hair are slightly too perfect, exhibiting a uniformity that may not occur naturally due to environmental factors like wind or humidity. This could indicate digital enhancement or generation. </reasoning>
<**reflection**> While the image passes most visual and physical plausibility checks, the overly uniform water droplets raise a subtle red flag. In real-world conditions, water droplets are often irregular in shape and distribution, especially in a windy beach environment. This artificial perfection, though minor, may point to post-processing or AI generation techniques designed to enhance realism but inadvertently introduce minor inconsistencies. </**reflection**>
<conclusion> Considering all evidence, the image is highly realistic and likely authentic. However, the slight over-regularity of the water droplets introduces doubt, suggesting possible digital manipulation. While not definitive proof of fakeness, this subtle artifact places the image in the medium difficulty category, requiring careful analysis to discern its true origin. </conclusion>
<answer> **real** </answer>

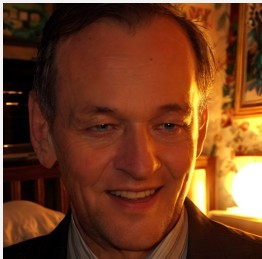

(**Fake** Image from **IC-Light, face relighting**)

<fast> At first glance, the image appears authentic with natural lighting and realistic facial features. </fast>
<planning> To confirm authenticity, we should examine the image through three progressive layers: first checking for obvious structural issues, then inspecting for subtle generator artifacts, and finally evaluating physical plausibility. Since no clear anomalies are visible, a detailed forensic analysis is necessary. </planning>
<reasoning> The facial structure is well-proportioned and symmetrical, with consistent skin tone and natural expression, ruling out any immediate signs of manipulation. Upon closer inspection, the edges and textures appear smooth and realistic, with no signs of artificial blurring or missing high-frequency details like hair strands. The lighting and shadows are consistent across the face and background, and there are no implausible makeup effects or focus discrepancies. The only potentially misleading aspect is the overly warm lighting, which could be a stylistic choice rather than a natural occurrence, but it does not contradict physical laws. </reasoning>
<reflection> However, the warm lighting, while not definitive evidence of fakeness, introduces a level of uncertainty. In real-world photography, such consistent lighting is uncommon without artificial setup, and AI-generated images sometimes overemphasize warmth to enhance realism. This subtle clue, combined with the image's otherwise perfect quality, could mislead a less thorough analysis. </reflection>
<conclusion> Considering all evidence, the image is most likely authentic, with the warm lighting being the only potentially misleading artifact. The absence of structural, textural, or physical inconsistencies strongly supports its realism, making this a high-difficulty case that requires nuanced judgment. </conclusion>
<answer> **real** </answer>

Figure 22: Failure cases of **VERITAS**.

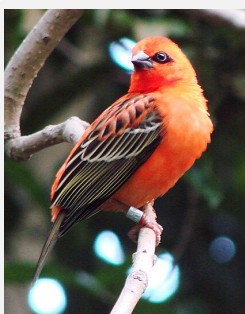

(**Real** Image from **LOKI**, animals)

**<fast>** At first glance, the image appears authentic, with vivid colors and natural bird features that align with real-world expectations. **</fast>**

**<planning>** To determine the authenticity, I will examine the image through three progressive layers: first checking for obvious structural inconsistencies, then inspecting for subtle AI artifacts, and finally evaluating adherence to physical realism. Since the initial judgment is uncertain, a detailed but focused analysis is necessary. **</planning>**

**<reasoning>** The bird's coloration is consistent and natural-looking, with no mismatched hues or unnatural patterns. The feathers exhibit realistic texture and detail, and the edges of the bird and branches appear smooth without digital artifacts. There is no blurring or blockiness that would suggest manipulation. Upon deeper inspection, the lighting and shadows behave as expected under real-world physics, and the background blur matches typical photographic effects. There is no loss of high-frequency detail or abnormal focus transitions that would point to synthetic generation. The overall composition and style remain consistent with a genuine photograph. **</reasoning>**

**<conclusion>** Based on the coherent structure, absence of AI artifacts, and adherence to physical realism, the image is most likely authentic. The lack of misleading elements and the presence of natural visual and physical cues strongly support this conclusion. **</conclusion>**

**<answer>** real **</answer>**

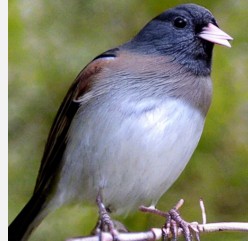

(**Fake** Image from **LOKI**, animals)

**<fast>** At first glance, the image seems realistic, with a well-maintained balance between lighting and shadow, consistent feathering, and natural angular placement of the bird. **</fast>**

**<planning>** Given this initial realism, I need to take a multi-step approach to determine authenticity. This involves first assessing structural coherence, then inspecting for subtler signs of generative artifacts, and finally evaluating adherence to physical constraints like depth of field and lighting consistency. This progressive analysis helps ensure that I don't miss underlying artifacts that might betray its machine-generated origin. **</planning>**

**<reasoning>** Upon closer inspection, the structure appears intact, with no immediately obvious deformations or mismatches in shape. However, critically evaluating the image reveals texture inconsistencies——specifically, an unusual rash-like texture appears on the bird's head near the eyes, which is more prominent than natural dappled shadow effects. This abrupt transition is not typical of natural lighting and suggests potential suboptimal rendering or blending artifacts from the image generation process. Furthermore, the bird's legs, when closely examined, showcase an unusual knot of claw shapes at the end, inconsistent with expected wildlife anatomy. These attributes typically result from inadequate attention to realism when combining elements during deepfake generation. Additionally, the unrealistic blurriness in the surrounding foliage compared to the sharpness of the bird raises hypothesis of digital manipulation, as normally, a consistent focus plane should exist within the same focal depth. **</reasoning>**

**<reflection>** Another aspect introducing suspicion is the interaction between the bird and the branch it's perched on. The lack of detailed identification of bark and natural contour symbiosis and overly smooth junction might be an artifact of replacing or manipulating the natural elements, revealing the digital deepfake nature of the image. Despite the initial convincing nature of the composition, these elements collectively suggest the image is digitally altered, as they defy the usual complexity and unique natural texture of living subjects. **</reflection>**

**<conclusion>** Although initially strongly convincing and convincingly mimicking real-life positioning, the rash-like texture anomalies, irregular claw shapes, and inconsistencies in focus depth, together signify the image as a medium-hard to detect deepfake. Lightweight AI detection techniques likely struggle with these subtleties, but careful inspection affirms the artificial origins that are otherwise concealed via typical digital generation methods. **</conclusion>**

**<answer>** fake **</answer>**

Figure 23: Reasoning output of **VERITAS** on AIGC images.

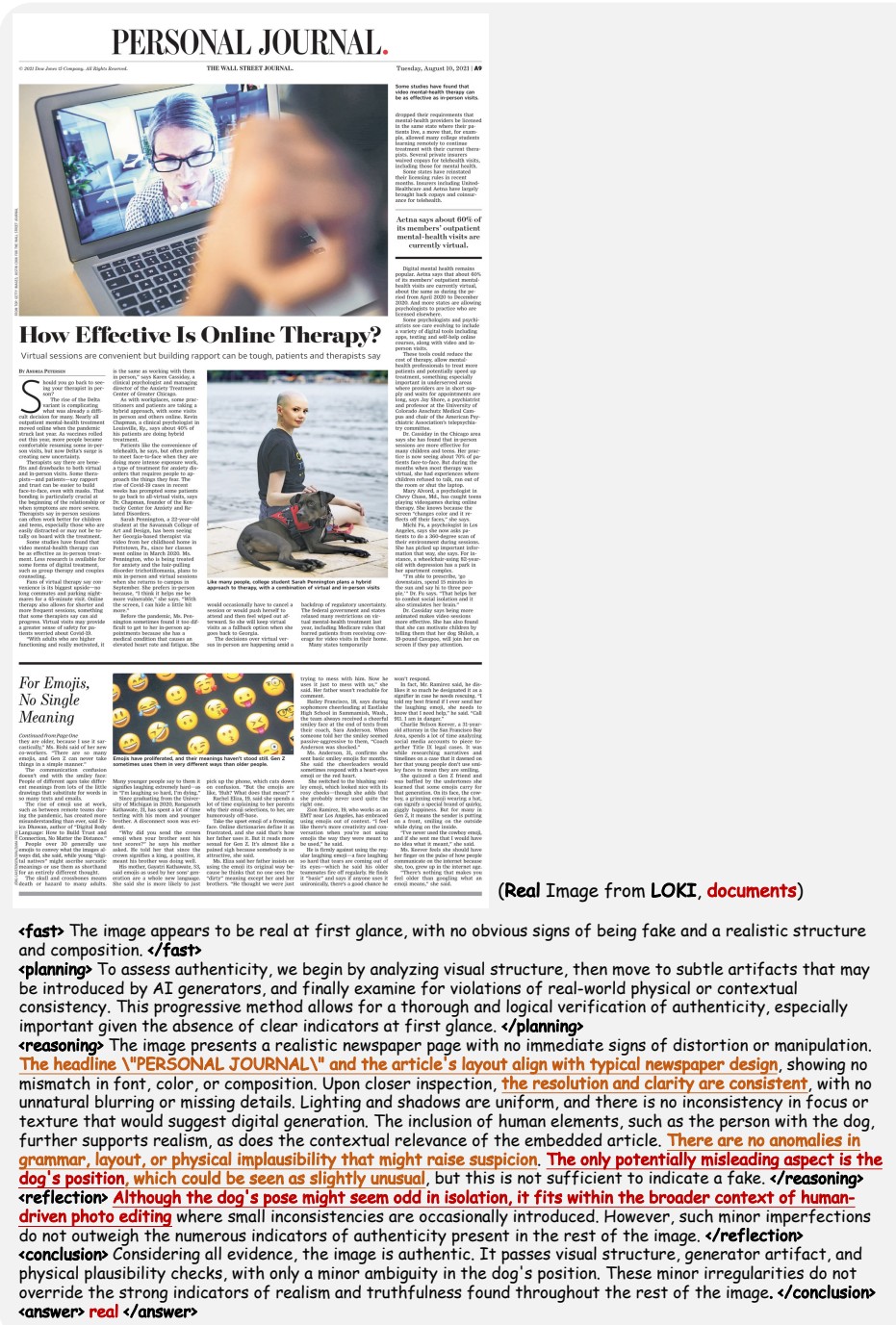

**(Real** Image from **LOKI**, **documents)**

**\<fast>** The image appears to be real at first glance, with no obvious signs of being fake and a realistic structure and composition. **\</fast>**

**\<planning>** To assess authenticity, we begin by analyzing visual structure, then move to subtle artifacts that may be introduced by AI generators, and finally examine for violations of real-world physical or contextual consistency. This progressive method allows for a thorough and logical verification of authenticity, especially important given the absence of clear indicators at first glance. **\</planning>**

**\<reasoning>** The image presents a realistic newspaper page with no immediate signs of distortion or manipulation. The headline \"PERSONAL JOURNAL\" and the article's layout align with typical newspaper design, showing no mismatch in font, color, or composition. Upon closer inspection, the resolution and clarity are consistent, with no unnatural blurring or missing details. Lighting and shadows are uniform, and there is no inconsistency in focus or texture that would suggest digital generation. The inclusion of human elements, such as the person with the dog, further supports realism, as does the contextual relevance of the embedded article. There are no anomalies in grammar, layout, or physical implausibility that might raise suspicion. The only potentially misleading aspect is the dog's position, which could be seen as slightly unusual, but this is not sufficient to indicate a fake. **\</reasoning>**

**\<reflection>** Although the dog's pose might seem odd in isolation, it fits within the broader context of human-driven photo editing where small inconsistencies are occasionally introduced. However, such minor imperfections do not outweigh the numerous indicators of authenticity present in the rest of the image. **\</reflection>**

**\<conclusion>** Considering all evidence, the image is authentic. It passes visual structure, generator artifact, and physical plausibility checks, with only a minor ambiguity in the dog's position. These minor irregularities do not override the strong indicators of realism and truthfulness found throughout the rest of the image. **\</conclusion>**

**\<answer>** real **\</answer>**

Figure 24: Reasoning output of **VERITAS** on AIGC images.

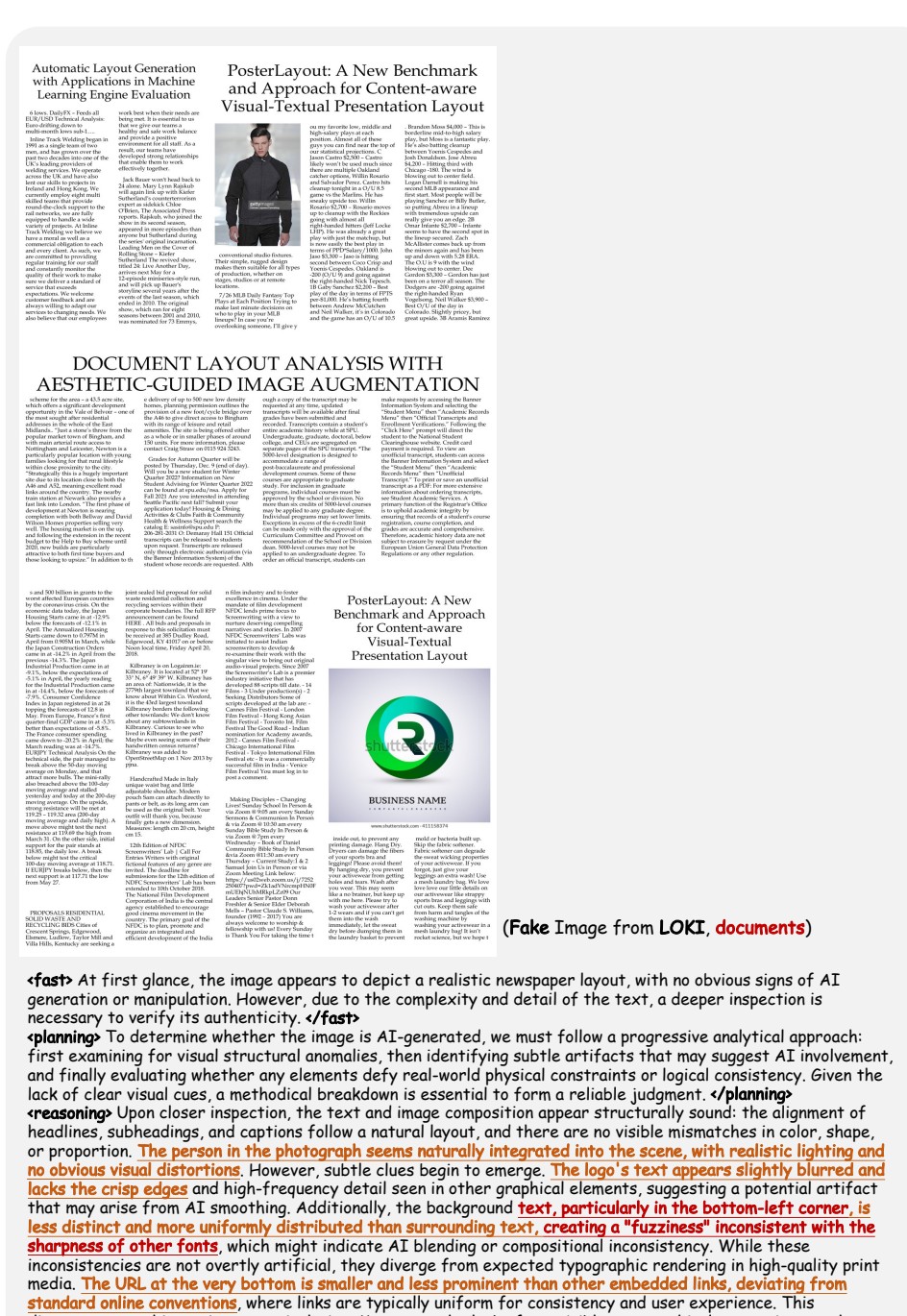

(**Fake** Image from **LOKI**, **documents**)

**<fast>** At first glance, the image appears to depict a realistic newspaper layout, with no obvious signs of AI generation or manipulation. However, due to the complexity and detail of the text, a deeper inspection is necessary to verify its authenticity. **</fast>**

**<planning>** To determine whether the image is AI-generated, we must follow a progressive analytical approach: first examining for visual structural anomalies, then identifying subtle artifacts that may suggest AI involvement, and finally evaluating whether any elements defy real-world physical constraints or logical consistency. Given the lack of clear visual cues, a methodical breakdown is essential to form a reliable judgment. **</planning>**

**<reasoning>** Upon closer inspection, the text and image composition appear structurally sound: the alignment of headlines, subheadings, and captions follow a natural layout, and there are no visible mismatches in color, shape, or proportion. The person in the photograph seems naturally integrated into the scene, with realistic lighting and no obvious visual distortions. However, subtle clues begin to emerge. The logo's text appears slightly blurred and lacks the crisp edges and high-frequency detail seen in other graphical elements, suggesting a potential artifact that may arise from AI smoothing. Additionally, the background text, particularly in the bottom-left corner, is less distinct and more uniformly distributed than surrounding text, creating a "fuzziness" inconsistent with the sharpness of other fonts, which might indicate AI blending or compositional inconsistency. While these inconsistencies are not overtly artificial, they diverge from expected typographic rendering in high-quality print media. The URL at the very bottom is smaller and less prominent than other embedded links, deviating from standard online conventions, where links are typically uniform for consistency and user experience. This discrepancy can hint at image manipulation. Moreover, the lack of any visible typographical annotations, such as page breaks or section dividers, further weakens the appearance of a structured, physical layout. **</reasoning>**

**<conclusion>** Considering all observed factors, though the image maintains an initially convincing appearance, subtle inconsistencies in detail, clarity, and textual uniformity suggest it is likely AI-generated. The varied quality, particularly in the logo and background text, adds layers of doubt that, although sophisticated, are indicative of digital manipulation, classifying this image as a medium-hard to detect forgery. These nuances invite closer scrutiny, revealing its potential artificial nature beyond its initial realism. **</conclusion>**

**<answer>** fake **</answer>**

Figure 25: Reasoning output of **VERITAS** on AIGC images.

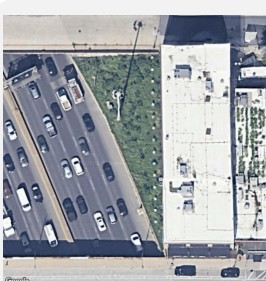

(**Real** Image from **LOKI**, satellite)

**\<fast\>** At first glance, the image appears authentic with no obvious signs of AI manipulation. The aerial view shows typical urban features such as roads, cars, and buildings that are consistent with real-world imagery. **\</fast\>**

**\<planning\>** To determine the authenticity more thoroughly, we need to examine the image through three progressive layers: first checking for visual structural abnormalities, then analyzing for subtle generator artifacts, and finally evaluating for violations of physical plausibility. Each level of inspection builds upon the previous one, ensuring a comprehensive forensic review. **\</planning\>**

**\<reasoning\>** The image shows no structural inconsistencies such as mismatched color tones, distorted shapes, or unnatural proportions. The proportions of the building and the alignment of cars are realistic, and the shadows are logically placed, suggesting a natural scene. Upon closer inspection, the edges and textures appear smooth and coherent, with no signs of artificial blending or unnatural blurring. Details like individual cars and text on the building are preserved, and the overall sharpness is uniform. There are no missing high-frequency textures or blocky artifacts typically introduced by generative models. Deeper analysis reveals that the lighting and shadows follow natural patterns, and the surrounding context--such as the road layout and vegetation--is contextually consistent. There is no stylization or focus discrepancy that would suggest AI generation or post-processing artifacts. **\</reasoning\>**

**\<conclusion\>** Considering the absence of visual, subtle, or physical-level anomalies, the image maintains a high level of realism and coherence characteristic of an authentic aerial photograph. There is no compelling evidence to suggest it is fake, leading to the conclusion that it is most likely a real image. **\</conclusion\>**

**\<answer\>** real **\</answer\>**

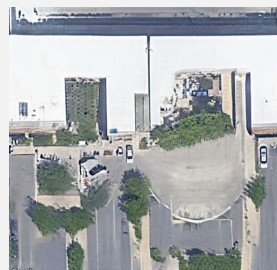

(**Fake** Image from **LOKI**, satellite)

**\<fast\>** At first glance, the image presents an unusually distorted aerial view, raising initial suspicion about its authenticity due to apparent unnaturalness. **\</fast\>**

**\<planning\>** Given the initial impression of distortion, a detailed analysis is necessary to ascertain the image's authenticity. This involves examining structural coherence, assessing for subtle artifacts of artificial generation, and evaluating structural integrity, especially in areas where the distortion reveals irregularities that defy real-world geometry. A comprehensive inspection is essential to uncover hidden anomalies indicative of a deepfake creation. **\</planning\>**

**\<reasoning\>** Starting with an evaluation of the structures, the building near the center-left of the image appears severely warped and misplaced. Its angles and proportions are inconsistent with typical architectural forms, with the left side seemingly blending into the street boundary below, suggesting a mishandled synthesis or massaging error during deepfake creation. This kind of misalignment is a classic giveaway in identifying artificially altered imagery. Examining the greenery across the scene, there is a lack of coherent integration, where the shapes and textures do not harmonize with realistic environmental expectations. Trees appear muddled and blurred, as if the algorithm struggled to form distinct foliage from pixel data. These distortions might be attributed to inadequate training data or issues in segmentation tasks typical of AI-generated images. While assessing the image's overall geometric consistency, the problem of perspective becomes apparent. The angle and depth transitions between the foreground and background hold no adherence to a singular, logical viewpoint. This is likely due to AI's capacity to struggle with seamless integration across varying viewpoints, resulting in an image where depth appears more subjective than spatially accurate. This inconsistency in perspective further cements the likelihood of digital tampering or synthetic origins. Additionally, the traffic near the bottom of the image seems compressed and lacks realistic flow, reinforcing the idea that elements were not carefully aligned or realistically integrated. **\</reasoning\>**

**\<reflection\>** Upon deeper reflection, it becomes evident that the aforementioned warping and distortions in both structure and perspective contribute to a visual 'cognitive dissonance,' where the human mind begins to question the image's 'reality.' The inability to recognize typical structural features, coupled with unnatural blending lines between elements, contributes to this effect. It's as though the AI attempted to portray an organized space but failed to execute complex realistic spatial comprehension, leading to an extrusion of visual cues detaching from natural harmony. These errors, while subtle in implication, collectively betray its synthetic origins, showcasing the challenges AI faces in emulating complex real-world environments with fidelity. **\</reflection\>**

**\<conclusion\>** Despite its initial suggestion of an aerial perspective, the image's distortions——particularly in structure integrity, coherence of natural elements, and perspective alignment——strongly signal its artificial nature. These errors, especially when viewed collectively, unveil the image as an AI-generated forgery. The chaotic blend of unnatural architecture and blurred green textures, coupled with problematic perspective transitions, confirms the image as an easy-to-detect deepfake, illustrating common failings in AI's capability to render realistic, coherent scenes. Such anomalies generally arise due to its segmentation and construction challenges, affirming its identification as fake without needing advanced scrutiny. **\</conclusion\>**

**\<answer\>** fake **\</answer\>**

Figure 26: Reasoning output of **VERITAS** on AIGC images.

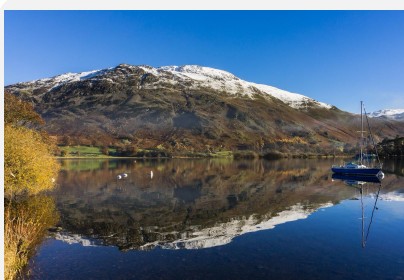

(**Real** Image from **LOKI**, **scene**)

**\<fast\>** At first glance, the image appears authentic with natural mountain and forest scenery under a clear sky. **\</fast\>**

**\<planning\>** To assess authenticity, we proceed in three stages: first evaluating obvious structural features, then checking for subtle AI artifacts, and finally analyzing for violations of physical realism. Since no clear anomalies are visible, a detailed inspection is necessary to ensure no hidden clues suggest artificial generation. **\</planning\>**

**\<reasoning\>** The mountain's snow cover and surrounding terrain show no signs of distortion, color inconsistency, or unnatural blending. The lighting and shadows align with a single light source, and there is no mismatch between the sky, clouds, and landscape. Upon closer inspection, the edges and textures are smooth and realistic, with no artificial boundaries or blur. The image retains high-frequency details like snow texture and forest variations, which are often missing in AI-generated images. There are no signs of stylization or focus discrepancies that would imply generation or post-processing manipulation. **\</reasoning\>**

**\<conclusion\>** Synthesizing all observations, the image exhibits no signs of AI generation or manipulation. It maintains visual, textural, and physical consistency with a real-world photograph. The initial judgment and final classification as real align with the detailed forensic evidence present. **\</conclusion\>**

**\<answer\>** real **\</answer\>**

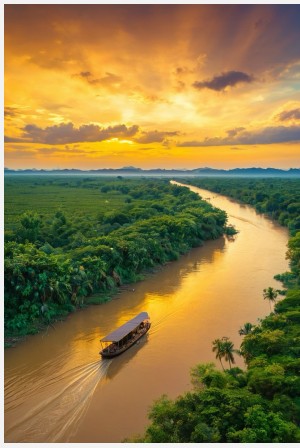

(**Fake** Image from **LOKI**, **scene**)

**\<fast\>** At first glance, the image presents a realistic and captivating scene, but an immediate determination of authenticity is not possible without deeper inspection. **\</fast\>**

**\<planning\>** Given the initial impression of realism, a systematic deep dive into the image is necessary. We will start by examining visual structural coherence and lighting, then proceed to detect subtle artifacts often introduced by AI, and finally assess whether any elements violate physical or contextual plausibility. This layered approach ensures no detail is overlooked, especially since the image does not reveal its nature superficially. **\</planning\>**

**\<reasoning\>** At first, the landscape elements, such as the river flow and the surrounding greenery, appear natural without obvious mismatches in color or structure. The light source on the horizon creates realistically long shadows, and the integration between the boat and the water suggests dynamic interaction. However, diving deeper into the finer details reveals potential anomalies——such as the smooth and uniform green coloration of the vegetation and the sharpness of the riverbank edges, which lack some expected natural variation and texture seen in real environments. The water surface seems extremely polished and reflective while the reflection itself is overly perfect——suggesting a synthetic material appearance that could be a marker of AI rendering. Additionally, the skies exhibit a dramatic dynamic range in both hue and texture that may be too exaggerated for a naturally occurring scene, potentially indicating digital enhancements typical of deepfake generation techniques. A notable indication is the absence of natural imperfections such as weather effects on the water or variations in foliage texture, further pointing towards the possibility of the image being AI-generated. Another layer of scrutiny reveals an unusual interaction of light around the clouds and the horizon, where the shadows and highlights don't transition smoothly. This can signify an artificial attempt at replicating lighting effects that real-world physics might not produce consistently. **\</reasoning\>**

**\<reflection\>** Let's consider the environmental context——real-world settings usually include nuanced elements like floating particles, subtle aberrations in reflections or minor shadows mismatched due to atmospheric conditions. Here, the scene feels slightly stylized to an almost painted aesthetic, which, while masterfully done, diverges from the chaotic unpredictability of actual nature. This kind of stylization can be a telltale sign of AI involvement, disguising imperfections under a veneer of hyper-realism. Moreover, the image exhibits an amusing clarity in certain aspects——like the boat and river——are overly precise in detail compared to other vague elements in the scene, creating an incoherent focus balance that does not align with real camera optics. Something feels subtly 'just right' that feels hyper-real yet artificial. **\</reflection\>**

**\<conclusion\>** Even though the image initially radiates authenticity through its visual coherence, the overly smooth texture of the water, consistent reflections, and stylistic blending of natural elements suggest an AI-generated origin. The subtle discrepancies in environmental interactions and optical focus, although nuanced, point to technologies rather than an authentic snapshot. Thus, despite its convincing realism, the meticulous examination highlights its deepfake nature, confirming it as a sophisticated AI creation. **\</conclusion\>**

**\<answer\>** fake **\</answer\>**

Figure 27: Reasoning output of **VERITAS** on AIGC images.

---

**Pattern-aware SFT Data Annotation (Stage-1 Prompt for Fake Images)**

```
{real image} {fake image}
```
You are given two images:
1. A real face image captured by real-world camera.
2. A fake face image generated by AI, using `{Forgery Type and Explanation}`.

Please compare them visually and select **two most noticeable abnormalities** that best indicate the second image is fake.

Choose from the following list:
**1. Color Difference**
Inconsistent skin tone between facial regions (e.g., skin tone mismatches in face-swapped regions).
**2. Structure Abnormal**
The structures of facial features (e.g., eyes, nose, eyebrows, mouth) are distorted or asymmetric, which do not conform to the biometric patterns found in real human faces.
**3. Edge Abnormal**
Artificially sharp or jagged facial contours/edges, which are inconsistent with natural soft transitions.
**4. Expression Abnormal**
Unnatural or distorted facial expression which are not common in real human faces (e.g., mismatched smile-eye movements).
**5. Facial Proportion Abnormal**
Unusual proportion of facial components (e.g., abnormally wide eye distance or overly large forehead).
**6. Texture Abnormal**
Overly smooth/rough or discontinuous skin textures.
**7. Blending Boundary**
In blending techniques, the facial skin exhibits hard transitions or obvious block boundary.
**8. Unnatural Blur**
Abnormal local/block blurriness on facial regions or facial contour.
**9. High-Frequency Detail Missing**
Although the image is in high-resolution and looks realistic, some human biological details (e.g., pores and fine hair) are missing.
**10. Non-physically Plausible Lighting and shadow**
Inconsistent light/shadow directions across facial regions or light/shadow that violate physical illumination laws.
**11. Contextual Element Mismatch**
Incongruous background or mismatched style/lighting between face and background.
**12. Makeup Implausibility**
Conflicting specular highlights and makeup distribution (e.g., highlight on non-glossy skin areas).
**13. Holistic Stylization**
In advanced entire face generation, the image looks realistic, but is stylized overall (e.g., hyper-realistic digital art).
**14. Abnormal Optical Focus Discrepancy**
Entirely generated faces sometimes fail to simulate natural camera optics, exhibiting simultaneous sharpness in geometrically incompatible depth planes or abrupt defocus transitions violating optical distance gradients.

Return only your selected clues, exactly as listed, separated by " , ".
Please strictly follow the format.

---

Figure 28: Prompt for fake images SFT data annotation (Stage 1: anomalies identification).

---

**Pattern-aware SFT Data Annotation (Stage-2 Prompt for Fake Images)**

`{image}`
### Background Information
A fake face image generated by AI has multiple possible artifacts. Generally, these can be divided into three categories.
1. **Visual Structural Abnormalities**: Obvious abnormalities that can be clearly perceived visually, typically including the following types:

    1.1 Color Difference

    Inconsistent skin tone between facial regions (e.g., skin tone mismatches in face-swapped regions).

    ...
2. **Subtle Artifacts from Generator**: Subtle artifacts introduced by generators that require more careful observation, typically including the following types:

    2.1 Edge Abnormal

    Artificially sharp or jagged facial contours/edges, which are inconsistent with natural soft transitions.

    ...
3. **Violation of Physical Laws**: Implicit artifacts that require deeper observation and thinking, connecting visual clues with common knowledge. This typically includes the following types:

    3.1 Non-physically Plausible Lighting and shadow

    Inconsistent light/shadow directions across facial regions or light/shadow that violate physical illumination laws.

    ...

### Preliminary Observation
The given image is a fake image generated by `{Forgery Type and Explanation}`. After careful inspection, we conclude two most noticeable abnormalities that indicate the image is fake:
`{Stage-1 Results}`

### Task Definition
Your task is to examine the given image, then:
1. Give your initial judgement of the authenticity of the image:

- Do not use any priori information provided above

- If you think it is hard to make a judgment, you can point this out faithfully

2. Extract **meticulous visual facts** that mainly conform to above two abnormalities. Specifically:

- Take a careful examination of the given image.

- Perform step-by-step forensics analysis according to the above three progressive categories.

3. Draw a comprehensive conclusion based on your findings.

Keep your answer detailed and factual.

---

Figure 29: Prompt for fake images SFT data annotation (Stage 2: visual facts forensics). The omitted parts in "Background Information" are consistent with the artifacts list in Figure 28.

---

**Pattern-aware SFT Data Annotation (Stage-3 Prompt for Fake Images)**

Your task is to convert the given information into **logical chain-of-thought (CoT)**. The length and complexity should be conditioned on the given information.

### Extracted Evidence
The following information is the explanation to the artifacts of a fake image generated by AI. Specifically, we partition the explanation into three parts:

1. **Initial Judgement**
We have required previous model truthfully give the judgment at a first glance. If the image has obvious artifacts, it will generate certain judgment. Otherwise, it will need further inspection.
The initial judgement of current sample is as follows:
`{Initial Judgement from Stage-2}`

2. **Detailed Evidence**
We cluster the possible artifacts into three progressive groups and then require previous model to conduct point-by-point forensic analysis. The extracted evidence of current sample is as follows:
`{Forensics Analysis from Stage-2}`

3. **Conclusion**
Generally, for some ambiguous samples, we need to make a comprehensive judgement based on different aspects. Therefore, we require previous model to draw a comprehensive conclusion based on the extracted evidence.
The conclusion of current sample is as follows:
`{Conclusion from Stage-2}`

### Task Definition
Your task is to convert the given information into logical Chain-of-Thought (CoT).
You are not given the image, and you should keep faithful to the above information.
You can **flexibly** decide whether to perform Long-CoT or Short-CoT, based on the sample's difficulty.

- Long-CoT: Hard samples need to generate comprehensive and **logical** reasoning content. Often follow a **structured pattern**: Fast Judgement (`<fast>`) - Problem Planning (`<planning>`) - Evidence Collection (`<reasoning>`) - Conclusion (`<conclusion>`)

- Short-CoT: Medium and easy samples need to generate brief yet **critical** reasoning content. Often follow a **structured pattern**: Fast Judgement (`<fast>`) - Evidence Collection (`<reasoning>`) - Conclusion (`<conclusion>`)

The following guidance is only useful when you need it:

- For "Problem Planning", you should analyze the current state and draw a progressive and reasonable plan

- For "Evidence Collection", you should convert the given evidence into logical and coherent content. Do not mechanically perform step-by-step analysis using conjunctions like "first" and "next". Instead, make your reasoning smooth and natural

- If you suppose the current sample is extremely hard, you can insert "Self-Reflection" pattern before "Conclusion": You can smartly move some hard-to-detect artifacts from "Evidence Collection" into this part. Use natural conjunctions like "However", "But wait", etc. Enclose in `<reflection>` tags. The reflective content is not a "restatement" but discovering something new that you have not considered before, which should be coherent with your previous reasoning content

- For "Conclusion", you should draw a comprehensive conclusion finally

Figure 30: Prompt for fake images SFT data annotation (Stage 3: thinking patterns injection).

---

**Pattern-aware SFT Data Annotation (Stage-1 Prompt for Real Images)**

`{image}`
### Background Information
The authentic images can be manipulated by AI for improper use.
A fake face image generated by AI may suffer from some of the following artifacts. Generally, these can be divided into three categories.
1. **Visual Structural Abnormalities**: Obvious abnormalities that can be clearly perceived visually, typically including the following types:
    1.1 Color Difference
    Inconsistent skin tone between facial regions (e.g., skin tone mismatches in face-swapped regions).
    ...
2. **Subtle Artifacts from Generator**: Subtle artifacts introduced by generators that require more careful observation, typically including the following types:
    2.1 Edge Abnormal
    Artificially sharp or jagged facial contours/edges, which are inconsistent with natural soft transitions.
    ...
3. **Violation of Physical Laws**: Implicit artifacts that require deeper observation and thinking, connecting visual clues with common knowledge. This typically includes the following types:
    3.1 Non-physically Plausible Lighting and shadow
    Inconsistent light/shadow directions across facial regions or light/shadow that violate physical illumination laws.
    ...

### Preliminary Observation
The given image is an authentic image captured by real-world camera.
Besides, we divide the authentic samples into three difficulty levels: easy, medium and difficult. In general, easy samples have high visual clarity and are extremely realistic.
Samples of medium difficulty are in lower quality, but there are still strong evidence indicating its authenticity.
High-difficulty samples are in low quality and may contain some misleading artifacts, requiring careful thinking and comprehensive judgment.
The currently given image is considered as `{difficulty}`.

### Task Definition
Your task is to examine the given image, then:
1. Give your initial judgement of the authenticity of the image:

- Do not use any priori information provided above

- If you think it is hard to make a judgement, you can point this out faithfully

2. Provide an explanation that can distinguish the given real image from fakeness. Specifically:

- Take a careful examination of the given image

- Perform step-by-step reasoning according to the above three progressive categories

- If there exits some **misleading artifacts** in the given image, you can point them out truthfully

3. Draw a comprehensive conclusion based on your reasoning
Keep your answer detailed and factual.

---

Figure 31: Prompt for real images SFT data annotation (Stage 1: visual facts forensics). We divide the difficulty of real images upon datasets. FFHQ and CelebAHQ are considered as simple for their clear visual details. FaceForensics++ and CelebA are classified as medium for their miss of visual details. LFW is considered as hard for its low resolutions and unexpected noises. The omitted parts in "Background Information" are consistent with the artifacts list in Figure 28.

---

**Pattern-aware SFT Data Annotation (Stage-2 Prompt for Real Images)**

```
{image}
```
Your task is to convert the given information into **logical chain-of-thought (CoT)**. The length and complexity of the reasoning chain should be conditioned on the given information.

### Extracted Evidence
The following information is a detailed explanation that distinguishes a given real image from fakeness.
Specifically, we partition the explanation into three parts:
1. **Initial Judgement**
We have required previous model truthfully give the judgement at a first glance. If the image has obvious artifacts, it will generate certain judgement. Otherwise, it will need further inspection. The initial judgement of current sample is as follows:
```
{Initial Judgement from Stage-1}
```
2. **Detailed Evidence**
We cluster the possible artifacts into three **progressive** groups and then require previous model to conduct point-by-point forensic analysis. The extracted evidence of current sample is as follows:
```
{Forensics Analysis from Stage-1}
```
3. **Conclusion** Generally, for some ambiguous samples, we need to make a comprehensive judgement based on different aspects. Therefore, we require previous model to draw a comprehensive conclusion based on the extracted evidence. The conclusion of current sample is as follows: `{Conclusion from Stage-1}`

### Difficulty Information
To enable better control of the length of the reasoning chain, we divide the authentic samples into three difficulty levels: easy, medium and hard.
In general, easy samples have high visual clarity and are extremely realistic. Medium samples are in lower quality, but there are still strong evidence indicating its authenticity. Hard samples are in extremely low quality and may contain some misleading artifacts, requiring careful thinking and comprehensive judgment. The currently given image is roughly classified as `{difficulty}`.

### Task Definition
You are not given the image, and you should keep faithful to the above information. You can **flexibly** decide whether to perform Long-CoT or Short-CoT, based on the sample's difficulty.

- Long-CoT: Hard samples need to generate comprehensive and **logical** reasoning content. Often follow a **structured pattern**: Fast Judgement (`<fast>`) - Problem Planning (`<planning>`) - Evidence Collection (`<reasoning>`) - Conclusion (`<conclusion>`)

- Short-CoT: Medium and easy samples need to generate brief yet **critical** reasoning content. Often follow a **structured pattern**: Fast Judgement (`<fast>`) - Evidence Collection (`<reasoning>`) - Conclusion (`<conclusion>`)

The following guidance is only useful when you need it:

- For "Problem Planning", you should analyze the current state and draw a progressive and reasonable plan

- For "Evidence Collection", you should convert the given evidence into logical and coherent content. Do not mechanically perform step-by-step analysis using conjunctions like "first" and "next". Instead, make your reasoning smooth and natural

- If there are any misleading artifacts in the provided information, should put them into "Reflection" pattern before "Conclusion", using natural conjunctions like "However", "Although", etc. Enclose in `<reflection>` tags. ONLY insert "Reflection" pattern when there are **known misleading artifacts**.

- For "Conclusion", you should draw a comprehensive conclusion finally

Figure 32: Prompt for real images SFT data annotation (Stage 2: thinking patterns injection).

---

**Reasoning quality (Score evaluation)**

You are a helpful assistant proficient in analyzing vision reasoning problems.

## Instruction: Please examine the provided image attentively and serve as an unbiased judge in assessing the quality of the response from an AI assistants regarding the instruction. You will receive a single response from the assistant to user's instruction.

## Noticement: Your assessment should identify whether the assistant effectively adheres to the user's instructions and addresses the user's inquiry.
In your evaluation, weigh factors such as preciseness, comprehensiveness, clarity, creativity, and the granularity of the responses.
Do not allow the length of the responses to influence your evaluation.
Do not favor certain names or positions of the assistants. Be as objective as possible.

## Criteria:
Use scores to show the quality of the response. Here is the detailed scoring rubric for evaluating the quality of responses from AI assistants:
**Incorrect (1)**: The answer is incorrect (e.g., predicting real for the fake image), the score should be 1.
**Poor (2)**: The response fails to address the query effectively. Although the answer is correct, the reasoning process is vague and repetitive. Creativity and granularity are absent or poorly executed.
**Average (3)**: The response shows superficial analysis of the given image. It lacks depth in creativity and granularity. The reasoning process is not detailed.
**Good (4)**: The response is well-aligned with the given image, demonstrating a high degree of preciseness and comprehensiveness. It shows creativity and a nuanced understanding of the topic, with a detailed granularity that enhances the response quality.
**Excellent (5)**: The response shows great preciseness, comprehensiveness, creativity, and granularity. It provides an insightful, detailed, and thorough answer, indicating a deep and nuanced understanding of the user's inquiry.

## Desired Output Format:
Present your verdict in a JSON format, with the key 'analysis' for a short reason of your judgement and the key 'judgment' to indicate your decision: use "[[1]]", "[[2]]", "[[3]]", "[[4]]", "[[5]]" to indicate your evaluate score.

## Question:
{image} This is a {Ground Truth} image. The following are the user's query and the model's output.
[The Start of User Instruction]
Please determine the authenticity of this image.
[The End of User Instruction]
[The Start of Assistant's Answer]
{Model's Reasoning Output}
[The End of Assistant's Answer]

Figure 33: Prompt for reasoning quality evaluation (Score evaluation).

---

**Reasoning quality (Pairwise evaluation)**

You are a helpful assistant proficient in analyzing vision reasoning problems.

## Instruction:
Please examine the provided image attentively and serve as an unbiased judge in assessing the quality of responses from two AI assistants regarding the user's question shown beneath the image.

## Noticement:
Your assessment should identify the assistant that more effectively adheres to the user's instruction and provides more detailed, more precise and high-quality reasoning.
In your evaluation, weigh factors such as preciseness, comprehensiveness, clarity, creativity, and the granularity of the responses.
Avoid any position biases and ensure that the order in which the responses were presented does not influence your decision.
Do not allow the length of the responses to influence your evaluation.
Do not favor certain names of the assistants. Be as objective as possible.

## Desired Output Format:
Present your verdict in a JSON format, with the key 'analysis' for a short reason of your judgement and the key 'judgment' to indicate your decision: use "[[A]]" if assistant A prevails, "[[B]]" if assistant B does, and "[[C]]" for a tie.

## Question:
{image} This is a {Ground Truth} image. The following are the user's query and the model's output.
[The Start of User Instruction]
Please determine the authenticity of this image.
[The End of User Instruction]
[The Start of Assistant A's Answer]
{Model A's Reasoning Output}
[The End of Assistant A's Answer]
[The Start of Assistant B's Answer]
{Model B's Reasoning Output}
[The End of Assistant B's Answer]

Figure 34: Prompt for reasoning quality evaluation (Pairwise evaluation).

---

**Generation of Personalization Prompts**

{image}
Your task is to create customized prompts for the input image to fool deepfake detectors.

### Requirements
1. Tailor the prompt based on the specific input image. Change the context. For example, "Old man with beard", "a chef in a bustling kitchen, exuding expertise and dedication", "beautiful bride, traditional, attire, floral braid, sequin headdress, orchid backdrop, pastels", etc.
2. Keep the prompt concise and effective.
3. Avoid using any obscure words.

Figure 35: Input for generating customized personalization prompts.

---

**Prompt for Reflection Quality Reward Model**

`{image}`
You are provided with an image and a question for this image. The provided response is the reasoning process of determining its authenticity.
You should re-examine the image carefully, and then review the self-reflection content enclosed in the `<reflection> </reflection>` tags:
1. Is the reflection content redundant with the reasoning content? Is the reflection content just a restatement or conclusion of previous reasoning? The reflection should introduce new insights rather than restatement.
2. The reflection should not be vague statements such as "too perfect" or "lack of imperfections". Instead, it should be specific and detailed.

From 0 to 100, how do you rate for the reflection quality?
Be strict, give low score if it is not aligned with the above principles.
Provide a few lines for explanation and the rate number at last after "Final Score:".

Your task is provided as follows:

Question: [`{Question}`]
Response: [`{Reasoning Output}`]

---

Figure 36: Prompt for Reflection Quality Reward model (UnifiedReward-Qwen-3B).

---

**Input Prompt for Veritas (All stages)**

### `System`:
You are an image authenticity expert. Your task is to determine the authenticity of the given facial image.

Firstly, give an overall judgement to the authenticity of the image, enclosed in `<fast> </fast>` tags.
Then, make a careful and structured thinking before reaching an answer. Based on your thinking, draw a comprehensive conclusion. Enclose the corresponding part in different tags, e.g., `<planning>` or `<reasoning>` or `<reflection>` or `<conclusion>`.
Finally, give the final answer with "real" or "fake", enclosed in `<answer> </answer>` tags.

### `User`:
`{image}` Please determine the authenticity of this image.

---

Figure 37: Input prompt for Veritas. The prompts for all training stages are consistent.

---

**Input Prompt for Qwen2.5-VL-7B InternVL3-8B and GLM-4.1V-9B-Thinking**

### `System`:
You are given an facial image. Please analyze the provided facial image and determine whether it is authentic or fake based on the following classification criteria:
Real Captured Facial image
- Images captured using a real camera or device without any alternations or manipulation.
Fake Facial Image
- Images generated or manipulated using digital technologies, such as deepfakes, face swapping, face reenactment, photo editing software, entire face synthesis, etc.
Output the thinking process in {`<think>`} {`</think>`} and final answer ("real" or "fake") in {`<answer>`} {`</answer>`} tags, i.e., the output answer format should be as follows:
{`<thinking>`} your thinking process here {`</thinking>`} {`<answer>`} your judgement here {`</answer>`} Please strictly follow the format.

### `User`:
{`image`} Please determine the authenticity of this image.

---

Figure 38: Input prompt for Qwen2.5-VL-7B, InternVL3-8B and GLM-4.1V-9B-Thinking.

---

**Input Prompt for MiMo-VL-7B**

### `System`:
You are Qwen, created by Alibaba Cloud. You are a helpful assistant.

### `User`:
{`image`} Please determine the authenticity of this image. Output your final answer ("real" or "fake") in `<answer>` `</answer>` tags

---

Figure 39: Input prompt for MiMo-VL-7B. We found that providing priori knowledge does no good for MiMo-VL-7B, hence we remove any priori in system prompt.

---

**Input Prompt for GPT-4o and Gemini-2.5-Pro**

### `System`:
You are an image authenticity expert. Your task is to determine the authenticity of the given facial image.

Firstly, give an overall judgement to the authenticity of the image, enclosed in `<fast>` `</fast>` tags.
Then, make a careful and structured thinking before reaching an answer. Based on your thinking, draw a comprehensive conclusion. Enclose the corresponding part in different tags, e.g., `<planning>` or `<reasoning>` or `<reflection>` or `<conclusion>`.
Finally, give the final answer with "real" or "fake", enclosed in `<answer>` `</answer>` tags.

### `User`:
{`image`} Please determine the authenticity of this image.

---

Figure 40: Input prompt for GPT-4o and Gemini-2.5-Pro. Similar to MiMo-VL-7B, we found that providing priori knowledge is not helpful. We keep the default system prompt and only customize user prompt by constraining the output format.

