# OpenReview forum: "Veritas: Generalizable Deepfake Detection via Pattern-Aware Reasoning"
_ICLR.cc/2026/Conference — ICLR 2026 Oral_

### Official Review · Reviewer_wJ4c · 2025-10-20

**Soundness:** 3
**Presentation:** 3
**Contribution:** 3
**Rating:** 4
**Confidence:** 4

**Summary:**

The paper introduces the HydraFake dataset and the VERITAS model.
The HydraFake dataset encompasses a diverse range of deepfake generation techniques and real-world forgery cases. It provides a rigorous training and evaluation protocol that covers unseen model architectures, emerging forgery techniques, and novel data domains.The VERITAS detector, built upon a Multimodal Large Language Model (MLLM), differs from conventional Chain-of-Thought (CoT) based reasoning frameworks. It employs a two-stage training pipeline — pattern-guided cold start followed by pattern-aware exploration — with a particular emphasis on key reasoning modes such as “planning” and “self-reflection”.By simulating the human forensic reasoning process, VERITAS integrates the cognitive capabilities of large language models into deepfake detection (DFD), thereby achieving enhanced cross-manipulation and cross-domain generalization performance.

**Strengths:**

The proposed VERITAS framework demonstrates strong innovation in the field of Deepfake detection. Its core contribution lies in transforming the traditional feature-memorization-based detection paradigm into a reasoning-based framework grounded in pattern-aware reasoning.Unlike previous models that rely on specific forgery artifacts, VERITAS explicitly models forgery patterns as transferable reasoning units, enabling the understanding of common structural characteristics shared across different forgery types. In addition, the authors introduce a Chain-of-Thought (CoT) reasoning mechanism, allowing the model to progressively identify manipulated regions through multi-step logical inference, thereby enhancing interpretability and robustness.
The paper further proposes a Mixed Preference Optimization (MiPO) strategy, which balances the learning preferences among various forgery patterns, effectively improving the model’s cross-manipulation and cross-domain generalization capabilities.Overall, this study presents substantial innovation and academic value across three dimensions — detection paradigm, reasoning mechanism, and optimization strategy.

**Weaknesses:**

1. Lack of fair comparison with recent large-scale multimodal frameworks.

Although VERITAS demonstrates impressive performance, the experimental comparison is incomplete and lacks fairness.
The paper does not include evaluations against recent large-scale multimodal or reasoning-based deepfake detection frameworks, such as FakeShield (ICLR 2025), M2F2-Det (CVPR 2025), and SIDA (CVPR 2025).

These methods share similar reasoning or multimodal fusion paradigms and therefore represent the most relevant baselines for comparison.

In contrast, most of the reported baselines are lightweight detectors (with only tens or hundreds of millions of parameters), while VERITAS is built upon an 8B-scale MLLM.


2. Limited domain coverage.

The current work focuses exclusively on face-oriented deepfake detection, whereas a growing number of studies have begun exploring cross-domain and generalized forgery detection, such as AIGC-generated content, IMDL (image manipulation localization), and general multimedia forensics frameworks (e.g., Effort, FakeShield, ForensicHub(NeurIPS 2025) ).

Expanding the scope of VERITAS to handle non-facial or multi-domain forgery types would not only enhance its practical value but also align it with the emerging trend toward universal content authenticity verification.

**Questions:**

Q1: Could the authors provide additional comparisons or discussions with recent large multimodal reasoning-based detectors (e.g., FakeShield, M2F2-Det, SIDA) to better contextualize VERITAS?

If this question is properly addressed, it would increase my score, as a fair comparison with large models is the most critical issue.

Q2: Does the proposed VERITAS framework have the potential to generalize beyond face forgery detection — for example, to AIGC-generated or other cross-domain manipulation tasks?

If this aspect is explored or discussed, it would further raise my score, though it is secondary to the first point.

---

> ### Author Response · Authors · 2025-11-19
> **Author Responses**
>
> > **Q1. Comparisons with recent large multimodal reasoning-based detectors.**
>
> Thanks for the valuable suggestion. We make both **Quantitative (below)** and **Qualitative (in our revised manuscript, Figure 6, Figure 17~19)** comparisons to FakeShield (ICLR'25) [1], M2F2-Det (CVPR'25) [2], SIDA (CVPR'25) [3], FakeVLM (NeurIPS'25) [4] and FFAA [5]. To ensure a fair comparison, we further restrict the training scope of our model to FF++, StyleGAN, StableDiffusion XL and FFHQ (similar to FFAA [5]), yielding "Veritas-mini". The results have been added to our revised manuscript (Table 1).
>
> + M2F2-Det [2] and FFAA [5], which are specialized for Deepfake Detection, achieve promising results on several subsets (e.g., CodeFormer and FaceAdapter), but limited generalization on most CM sets.
>   - M2F2-Det excels at performing faithful analyses within facial region. However, it lacks consideration of deeper dimensions, resulting in suboptimal performance on those fully synthesized data, which require considering the overall context or physical plausibility.
>   - FFAA provides more detailed analyses. However, the logical coherence between "description" and "reasoning" part is weak, and the "reasoning" part lacks in-depth understanding. For example, in **Figure 6**, Veritas is able to perceive that "_texture is excessively detailed, almost unnaturally exaggerated_" and correctly identifies the image as AI-generated, **whereas FFAA is restricted to observations like** "_the wrinkles on the forehead and the stubble on the face are detailed_". Such superficial analysis causes failures on similar high-fidelity samples.
> + FakeShield [1] is not proficient at detecting facial forgeries, achieving 60.8% average performance.
>   - FakeShield falls short on analyzing fully synthesized facial data, but it demonstrates certain advantages in local artifact analysis, since it is trained for IMDL tasks.
> + SIDA [3] achieves 76.3% average results with 7B parameters, while 69.8% with 13B model, suggesting that parameter scale is not determinative to the generalization ability. Moreover, SIDA-7B achieves strong performance on CM sets  (e.g., 97.3% on ADF and 95.0% on Infinity), while still exhibiting certain shortages on CF (e.g., 60.6% on IC-Light) and CD (e.g., 68.9% on GPT-4o) sets.
>   - Following official guidelines, SIDA-13B-description is adopted for generating explanations. It provides generally high-quality explanations. However, we notice that it has a tendency to classify real facial images as fake.
> + FakeVLM [4] achieves the best average performance, achieving over 75% accuracy on most subsets.
>   - However, FakeVLM provides low-quality explanations regarding **facial forgeries**, e.g., **most cases are explained as** "The image exhibits underlying characteristic inconsistencies in its features that suggest it is artificially created". Such **vague** and template-like explanantions are likely due to its large-scale SFT training nature.
> + In contrast, Veritas achieves superior performance with more holistic and in-depth reasoning. Even with limited training scope, **Veritas-mini** **still outperforms existing MLLM-based detectors**, indicating the effectiveness of our pattern-aware reasoning framework and overall training strategy.

---

> ### Author Response · Authors · 2025-11-19
> **Author Responses (continued)**
>
> | Model            | #Param. | ADF      | FLUX     | StarryAI | MAGI     | HART     | Infinity | StarGAN2 | ICLight  | CodeF.   | InfiniteY. | PuLID    | FaceAda. | DeepFace. | InfY.    | Dreamina | Hailuo   | 4o       | FFIW     | Avg.     |
> | ---------------- | ------- | -------- | -------- | -------- | -------- | -------- | -------- | -------- | -------- | -------- | ---------- | -------- | -------- | --------- | -------- | -------- | -------- | -------- | -------- | -------- |
> | M2F2-Det         | 7B      | 56.0     | 57.7     | 59.8     | 61.8     | 61.3     | 55.4     | 78.9     | 65.5     | 80.0     | 57.4       | 57.5     | 76.3     | 73.0      | 56.3     | 67.2     | 50.6     | 53.0     | 70.6     | 63.2     |
> | FakeShield       | 22B     | 64.3     | 64.0     | 61.5     | 63.1     | 61.8     | 63.3     | 64.0     | 57.3     | 60.9     | 58.1       | 63.6     | 63.7     | 50.2      | 83.8     | 53.8     | 51.3     | 53.9     | 55.6     | 60.8     |
> | SIDA             | 7B      | **97.3** | 97.7     | 79.5     | 59.3     | 98.5     | 95.0     | 59.8     | 60.6     | 62.3     | 89.7       | 94.4     | 63.3     | 50.4      | 81.9     | 80.0     | 78.0     | 68.9     | 57.3     | 76.3     |
> | SIDA             | 13B     | 80.7     | 78.5     | 54.8     | 52.5     | 91.3     | 82.4     | 63.7     | 61.2     | 68.2     | 56.7       | 67.1     | 84.3     | 60.8      | 58.2     | 88.3     | 74.0     | 74.1     | 59.9     | 69.8     |
> | FFAA             | 7B      | 55.1     | 50.9     | 72.9     | 63.5     | 60.8     | 57.6     | 82.7     | 70.9     | 71.8     | 58.4       | 62.4     | 86.0     | 67.7      | 58.4     | 55.3     | 59.2     | 49.6     | 68.3     | 64.0     |
> | FakeVLM          | 7B      | 78.2     | 78.5     | 77.0     | 74.5     | 76.5     | 76.8     | 70.8     | 76.2     | 76.2     | 76.9       | 76.5     | 77.7     | **75.7**  | 83.6     | 81.5     | 80.8     | 78.7     | 74.5     | 77.3     |
> | **Veritas-mini** | 8B      | 95.5     | 99.1     | **97.3** | 72.8     | 97.0     | 96.1     | 82.5     | **76.3** | 90.0     | 83.7       | 82.9     | 79.3     | 72.5      | 78.7     | 92.0     | **93.0** | 85.5     | 70.6     | 85.8     |
> | **Veritas**      | 8B      | 94.8     | **99.8** | 97.0     | **99.9** | **99.9** | **99.9** | **90.3** | 75.7     | **97.0** | **91.8**   | **95.1** | **91.7** | 58.6      | **84.1** | **92.3** | 90.2     | **89.2** | **78.5** | **90.3** |

---

> ### Author Response · Authors · 2025-11-19
> **Author Responses (continued)**
>
> > **Q2. Does Veritas have the potential to generalize beyond face forgery detection?**
>
> We sincerely thank the reviewer for the insightful question.
>
> First of all, we found that Veritas already demonstrates *promising AIGC analysis capacity*, although being trained exclusively on facial data. As shown in our revised manuscript, Veritas can effectively analyze forgeries in animals (**Figure 23**), documents (**Figure 24~25**), satellite (**Figure 26**) and scene (**Figure 27**) images. This suggests that Veritas is *learning how to determine authenticity* rather than memorizing specific samples.
>
> Building on this foundation, we take the cheaper way (as discussed in **L1555-1557**) to extend Veritas. We fine-tune Veritas on a mixed dataset of 6K facial data (from original training set) + 3K AIGC data, using the proposed P-GRPO. *Without the need for dedicated CoT curation*, this fine-tuning results in promising improvements on generic AIGC detection task (**+5.4%** on LOKI [6] and **+6.5%** on FakeClue [4]).
>
> Overall, such efficient scalability is attributed to the generalization ability of Veritas, which guarantees high-quality rollouts in P-GRPO. We will explore more extensions based on Veritas (e.g., IMDL) in our future work.
>
> $^\dagger$ means fine-tuned on a mixed dataset of original training images and 3K AIGC images.
>
> | Model                 | LOKI     | FakeClue | Forensics-Bench | HydraFake |
> | --------------------- | -------- | -------- | --------------- | :-------- |
> | UniFD$^\dagger$       | 64.7     | 75.5     | 57.8            | 76.9      |
> | ProDet$^\dagger$      | 62.9     | 72.1     | 63.4            | 80.8      |
> | Co-SPY$^\dagger$      | 61.7     | 68.1     | 70.9            | 84.2      |
> | D3$^\dagger$          | 67.0     | 79.3     | 58.0            | 79.4      |
> | Effort$^\dagger$      | 73.7     | 85.2     | 62.7            | 85.1      |
> | InternVL3-8B          | 53.0     | 59.1     | 60.5            | 58.5      |
> | MiMo-VL-7B            | 63.1     | 65.2     | 63.8            | 73.4      |
> | M2F2-Det              | 46.4     | 57.3     | 60.0            | 63.2      |
> | FakeShield            | 62.0     | 67.5     | 73.1            | 60.8      |
> | SIDA-7B               | 61.2     | 56.3     | 58.2            | 76.3      |
> | Veritas               | 72.1     | 85.9     | 70.8            | 90.3      |
> | **Veritas$^\dagger$** | **77.5** | **92.4** | **73.8**        | **91.1**  |

---

> ### Author Response · Authors · 2025-11-19
> **Author Responses (continued)**
>
> > **References**
>
> [1] Fakeshield: Explainable image forgery detection and localization via multi-modal large language models. In ICLR, 2025.
>
> [2] Rethinking Vision-Language Model in Face Forensics: Multi-Modal Interpretable Forged Face Detector. In CVPR, 2025.
>
> [3] SIDA: Social media image deepfake detection, localization and explanation with large multimodal model. In CVPR, 2025.
>
> [4] Spot the fake: Large multimodal model-based synthetic image detection with artifact explanation. In NeurIPS, 2025.
>
> [5] FFAA: Multimodal large language model based explainable open-world face forgery analysis assistant. arXiv preprint arXiv:2408.10072 (2024).
>
> [6] LOKI: A comprehensive synthetic data detection benchmark using large multimodal models. In ICLR, 2025.
>
> [7] Forensics-Bench: A Comprehensive Forgery Detection Benchmark Suite for Large Vision Language Models. In CVPR, 2025.

---

> ### Author Response · Authors · 2025-11-21
> **Thanks for the positive feedback!**
>
> We sincerely thank the reviewer for their positive and timely feedback.
>
> We have rechecked the paper and corrected the bolding issue in Table 1. The manuscript has been re-uploaded.
>
> Besides, we are already in the process of preparing the code and associated data. All the codes, datasets and model checkpoints will be open-sourced to the community upon the acceptance of the paper.

---

### Official Review · Reviewer_dw7z · 2025-10-31

**Soundness:** 4
**Presentation:** 4
**Contribution:** 4
**Rating:** 8
**Confidence:** 4

**Summary:**

This paper makes two primary contributions to the field of deepfake detection. First, it introduces HydraFake dataset, which extends the evaluation into a rigorous and hierarchical setting, providing a more realistic and challenging benchmark to the field. Second, the paper proposes Veritas, a novel MLLM-based detector. Experiments show both qualitatively and quantitatively the effectiveness of the model.

**Strengths:**

- The proposed dataset is well-motivated. The division of four evaluation levels (i.e., in-domain, cross-model, cross-forgery and cross-domain) is reasonable. A fine-grained evaluation protocol is critical and reasonable at the moment, and the constructed dataset is of high quality, providing a challenging evaluation suite for the community.
- The proposed pattern-aware reasoning is effective and insightful compared to previous explainable methods. Experiments clearly show its superiority compared to previous pipelines.
- The proposed MiPO and P-GRPO is coherent with the proposed reasoning framework. Thorough ablations are conducted to validate their effectiveness.

**Weaknesses:**

- The authors should provide some failure cases to understand the model’s limitations.
- More fine-grained ablations on the reasoning patterns could be done, e.g., what if removing the “reflection”/“planning” pattern?
- How "reflection" improves model's generalization capability to unseen forgeries? The author should provide more explanations to it.
- The human has very good reasoning capabilities. Why even human cannot accurately detect some (realistic) deepfakes? Is semantic-level reasoning capability truly crucial for deepfake detection?

**Questions:**

1.	In Figure 1, the proposed Veritas can perceive the textual anomalies. Is such type of analysis contained in the training set? If not, can the base model conduct similar analysis?

2.	In the MiPO stage, how is the image selected? What if removing the non-preference $s_l^{\phi}$ in the training data? It would be great to provide some case studies.

---

> ### Author Response · Authors · 2025-11-19
> **Author Responses**
>
> > **Q1. The authors should provide some failure cases.**
>
> Thanks for the suggestion. We have incorporated it in the revised manuscript.
>
> **Real images:** Most failures occur on the low-resolution data. These data are generally of low quality, and the unexpected artifacts such as localized blurriness would affect the model's judgement.
>
> **Fake image:** Failures mainly occur on totally unseen forgery types such as face relighting. However, although the final answer is incorrect, our model’s reasoning process still figures out suspicious clues, providing valuable insights that could be used for further scrutiny or future improvements.
>
> > **Q2. More fine-grained ablations & How "reflection" improves generalization.**
>
> >> **Q2.1 More fine-grained ablations could be done.**
>
> Thanks for the suggestion. To evaluate the effects of different patterns, we conduct a fine-grained ablations:
>
> |                  | In-domain   | Cross-model | Cross-forgery | Cross-domain | Avg.        |
> | ---------------- | ----------- | ----------- | ------------- | ------------ | ----------- |
> | `<think> <answer>` | 96.2        | 94.3        | 81.2          | 76.8         | 87.1        |
> | Ours             | 96.9        | 98.4 | **87.4**      | **80.1**     | **90.7**    |
> | w/o `<fast>`       | **97.3**    | **98.8**    | 86.9   | 79.1  | 90.5 |
> | w/o `<planning>`   | 96.7        | 96.9        | 85.0          | **80.1**     | 89.7        |
> | w/o `<reflection>` | 97.0        | 97.2        | 82.5          | 77.3         | 88.5        |
> | w/o `<conclusion>` | 97.2 | 98.2        | 86.2          | 79.0         | 90.1        |
>
> - "fast judgement" is helpful for CF and CD, but is not critical overall.
> - "planning" is more effective on CM, which is due to that high-resolution and fully-synthesized images require a more holistic and structured analysis.
> - "self-reflection" is critical in most settings, especially on CF and CD, as reflection incentivizes the model to discover those unseen artifacts.
> - "conclusion" provides certain gains, suggesting that synthesizing the separate evidence into a coherent verdict is also important.
>
> >> **Q2.2 How "reflection" improves generalization on unseen forgeries?**
>
> Thanks for the insightful question. In `<reasoning>`, the model conducts _basic_ analysis to facial evidence, while in `<reflection>` the model performs _deeper_ consideration. This involves extensive associations to common sense and world knowledge, **enabling the model to move beyond learned artifacts** and **discover those unseen artifacts**. For instance, badge anomalies in **Figure 1**, focus discrepancy in **Figure 14**, overall coherence with background in **Figure 15**. Such capacity distinguishes our model from post-hoc explanation methods, yielding greater generalization on unseen forgeries.
>
> > **Q3. Is reasoning capability truly crucial for deepfake detection?**
>
> We sincerely thank the reviewer for this insightful question. We agree that humans often fail to detect highly realistic deepfakes, while this stems from the **physiological limit of human perception.** The human excels at high-level semantic reasoning, such as judging contextual plausibility, but is less equipped to detect subtle, low-level digital artifacts like subtle blurriness or unnatural texture patterns. In contrast, the **machine can be trained to perceive these subtle artifacts** with superhuman accuracy. The primary challenge, which traditional detectors face, is not perception but generalization, i.e., they tend to overfit to specific artifact patterns.
>
> This is where the pattern-aware reasoning in our Veritas model becomes crucial. Our goal is not to mimic a human's intuitive guess, but to emulate a forensic expert's systematic investigation. Our approach uniquely **combines** _the machine's superhuman perception_ **with** _a structured and human-like reasoning framework_ (e.g., planning, reasoning, self-reflection, conclusion). To clarify this, we provide **case studies in our revised manuscript (Figure 21)** , where the model can perceive subtle artifacts (e.g., barely noticeable blurriness and faint texture anomalies) that are nearly imperceptible to the human.
>
> Therefore, reasoning is crucial not to replicate the fallible human eye, but to provide a logical structure that effectively leverages the model's perceptual abilities for robust generalization.

---

> ### Author Response · Authors · 2025-11-19
> **Author Responses (continued)**
>
> > **Q4. Is textual anomalies analysis contained in the training set? Can the base model conduct similar analysis?**
>
> Thanks for pointing this out. The textual anomalies analysis is not contained in our training set. Our training data consists exclusively of images where the face is the primary subject, and the corresponding reasoning data is mainly focused on facial artifacts.
>
> When the base model (InternVL3-8B) is prompted with generic instruction (e.g., to perform `<think> ... </think> <answer> ... </answer>` for deepfake detection), it **fails** to identify the textual anomaly in Figure 1. However, this can be triggered if we provide _highly specific, hand-crafted instructions._ For example, by explicitly prompting the model to "Pay attention to any text, clothing, ...", the base model can perform a similar analysis. Note that such prompt is not effective for all samples. In contrast, Veritas conducts thorough analysis without explicit customization, which highlights its effectiveness.
>
> > **Q5. How is the image selected in MiPO stage?**
>
> Sorry for the confusion. For MiPO, we select samples that the SFT model fails to reach all correct answers under 8 rollouts. The selected images are then balanced across forgery types, finally yielding 800 images. Then we 1) manually select 4 non-preference trajectories (from SFT model's outputs) for each image, and 2) annotate a preference trajectory, resulting in 3K samples for MiPO. We have added the illustration in our revised version.
>
> > **Q6. What if removing the non-preference $\mathbf{s}_l^{\phi}$ in MiPO?**
>
> - **Quantitative:** $\mathbf{s}_l^{\phi}$ helps improve the performance on CF (+1.3%) and CD (+0.8%) scenarios.
> - **Qualitative (in our revised manuscript, Figure 20):** Without $\mathbf{s}_l^{\phi}$, although the final answer is still correct, the reasoning process is less detailed compared to Veritas, e.g., in **Figure 20**, the model trained without $\mathbf{s}_l^{\phi}$ _successfully perceives lighting anamoly_ and _reach correct answer_, but **the analysis is superficial** ("The lighting on the face appears unusually uniform and slightly too perfect").
>
> |                                  | In-domain | Cross-model | Cross-forgery | Cross-domain | Avg. |
> | -------------------------------- | --------- | ----------- | ------------- | ------------ | ---- |
> | Veritas                          | 97.3      | 98.6        | 90.3          | 82.2         | 92.1 |
> | w/o MiPO                         | 96.9      | 98.4        | 87.4          | 80.1         | 90.7 |
> | MiPO (w/o $\mathbf{s}_l^{\phi}$) | 96.9      | 98.6        | 89.2          | 81.4         | 91.5 |
> | MiPO (w/o $\mathbf{s}_l^{\psi}$) | 65.3      | 64.8        | 58.6          | 54.3         | 60.8 |

---

### Official Review · Reviewer_MgHh · 2025-11-01

**Soundness:** 3
**Presentation:** 3
**Contribution:** 3
**Rating:** 8
**Confidence:** 3

**Summary:**

This paper introduces HydraFake, a deepfake detection dataset with hierarchical OOD evaluation (in-domain, cross-model, cross-forgery, cross-domain), and VERITAS, an MLLM-based detector using pattern-aware reasoning. VERITAS employs five thinking patterns (fast judgment, planning, reasoning, self-reflection, conclusion) trained via a two-stage pipeline: pattern-guided cold-start with MiPO and pattern-aware exploration with P-GRPO. The approach shows significant improvements in cross-forgery and cross-domain scenarios, which are important in real-life scenarios.

**Strengths:**

- The proposed HydraFake dataset addresses a crucial gap between academic benchmarks and industrial deployment scenarios, making it valuable for real-world applications.
- The authors have conducted comprehensive experiments, including comparisons with SOTA methods and detailed ablation studies, verifying the effectiveness of the proposed VERITAS model.
- The pattern-aware reasoning approach is reasonable, drawing inspiration from human cognitive processes to create more interpretable and robust detection systems.

**Weaknesses:**

- The two-stage training pipeline with MiPO and P-GRPO, though modified for forgery detection tasks, still seems to be a direct application of vanilla DPO and GRPO methods, which may undercut its novelty.
- The paper evaluates VERITAS exclusively on the proposed HydraFake dataset, raising concerns about overfitting to their specific evaluation protocol. It would strengthen claims about VERITAS's superiority and provide more convincing evidence of its effectiveness if authors could perform evaluations on other benchmarks such as LOKI [cite 1] and Forensics-bench [cite 2].

[cite 1] LOKI: A COMPREHENSIVE SYNTHETIC DATA DETECTION BENCHMARK USING LARGE MULTIMODAL MODELS. In ICLR 2025.

[cite 2] Forensics-Bench: A Comprehensive Forgery Detection Benchmark Suite for Large Vision Language Models. In CVPR 2025.

**Questions:**

None

---

> ### Author Response · Authors · 2025-11-19
> **Author Responses**
>
> > **Q1. The two-stage training pipeline still seems to be a direct application of vanilla DPO and GRPO methods.**
>
> We thank the reviewer for their valuable feedback. We acknowledge that our method is based on well-established algorithms like DPO and GRPO. However, we respectfully argue that our training strategy **is not naive applications** of these methods, but rather a well-motivated adaptation to the deepfake detection task.
>
> **Firstly**, DPO is not well-suited for our task. A key challenge in deepfake detection is that a model can arrive at the correct answer ("fake") but with a vague or incorrect reasoning process, as shown in **Figure 1**. Based on such empirical observation, we propose mixed non-preference optimization, which involves trajectories with correct answer but imprecise reasoning ($\mathbf{s}_l^{\phi}$), outperforming standard DPO in both score evaluation (4.4214 vs. 4.2863) and pairwise Elo rating (1359.0 vs. 1210.0) as validated in **Table 7**.
>
> **Secondly**, our P-GRPO incentivizes meaningful in-depth analysis through pattern-aware reward, providing finer signals compared to pure outcome-level reward while inheriting the merits of GRPO. Results in **Table 3** empirically validated the superiority of P-GRPO over naive accuracy reward.
>
> **Overall**, the training pipeline is an initial and effective attempt to transform these general-purpose reinforcement algorithms into deepfake detection task. Existing methods mostly utilize SFT with large-scale training data, while several methods try reinforcement learning by either directly utilizing GRPO [1] or applying standard DPO [2], lacking dedicated solutions to the task. Our design is motivated by empirical observations in deepfake detection and finally shapes an effective training pipeline.
>
> > **Q2. Evaluations on other benchmarks such as LOKI [3] and Forensics-bench [4].**
>
> We thank the reviewer for this constructive suggestion. Our work focuses on Deepfake Detection (i.e., face forgery), which is why benchmarks for generic images (i.e., AIGC Detection) were not included in main text. To validate that Veritas is not overfitted to the HydraFake protocol, we conduct additional evaluations on LOKI (ICLR'25) [3], Forensics-Bench (CVPR'25) [4], FakeClue (NeurIPS'25) [5], AIGIBench (NeurIPS'25) [6] and Nano-banana-150K [7].
>
> - **Quantitative:** Notably, Veritas shows promising performance on those AIGC benchmarks (e.g., **72.1%** on LOKI and **85.9%** on FakeClue) **with only facial training data**.
> - **Qualitative (in our revised manuscript, Figure 23~27):** We provide reasoning cases of Veritas on LOKI. The analysis suggests that Veritas has learned *how to determine the authenticity* (the meta principle shared across forensics tasks), allowing it to identify artifacts beyond facial domain.
>
> We have added these results and qualitative cases to the manuscript to provide more convincing evidence of Veritas's effectiveness and generalizability.
>
> | acc/F1              | LOKI              | LOKI (facial)     | FakeClue          | Forensics-bench   | AIGIBench         | Nano-banana-150K  |
> | ------------------- | ----------------- | ----------------- | ----------------- | ----------------- | ----------------- | ----------------- |
> | UniFD (CVPR'23)     | 54.5/58.7         | 74.8/69.7         | 61.6/64.2         | 53.6/54.8         | 78.0/80.2         | 49.1/36.0         |
> | ProDet (NeurIPS'24) | 53.8/56.6         | 63.2/66.4         | 62.9/69.8         | 65.1/72.0         | 74.8/73.9         | 64.6/63.8         |
> | Co-SPY (CVPR'25)    | 61.7/65.8         | 79.1/75.6         | 68.1/72.4         | **70.8**/**76.0** | 81.6/84.3         | 52.0/40.9         |
> | D3 (CVPR'25)        | 47.3/41.2         | 79.5/80.1         | 60.7/59.2         | 56.6/59.8         | 77.8/75.3         | 70.7/73.0         |
> | Effort (ICML'25)    | 53.8/50.0         | 84.3/84.6         | 65.0/63.2         | 57.0/59.4         | 84.7/87.6         | 62.7/52.1         |
> | InternVL3-8B        | 53.0/51.6         | 52.6/15.3         | 59.1/62.2         | 60.5/65.4         | 55.6/56.5         | 51.3/38.9         |
> | MiMo-VL-7B          | 65.1/64.3         | 69.7/65.0         | 67.2/71.6         | 63.8/70.5         | 62.8/64.3         | 60.7/55.6         |
> | **Veritas**         | **72.1**/**77.8** | **89.0**/**88.2** | **85.9**/**88.4** | **70.8**/74.9     | **88.9**/**90.4** | **86.3**/**89.0** |

---

> ### Author Response · Authors · 2025-11-19
> **Author Responses (continued)**
>
> > **References**
>
> [1] BusterX: MLLM-Powered AI-Generated Video Forgery Detection and Explanation. arXiv preprint arXiv:2505.12620 (2025).
>
> [2] AIGI-Holmes: Towards Explainable and Generalizable AI-Generated Image Detection via Multimodal Large Language Models. In ICCV, 2025.
>
> [3] LOKI: A comprehensive synthetic data detection benchmark using large multimodal models. In ICLR, 2025.
>
> [4] Forensics-Bench: A Comprehensive Forgery Detection Benchmark Suite for Large Vision Language Models. In CVPR, 2025.
>
> [5] Spot the fake: Large multimodal model-based synthetic image detection with artifact explanation. In NeurIPS, 2025.
>
> [6] Is Artificial Intelligence Generated Image Detection a Solved Problem? In NeurIPS, 2025.
>
> [7] Echo-4o: Harnessing the Power of GPT-4o Synthetic Images for Improved Image Generation. arXiv preprint arXiv:2508.09987 (2025).

---

### Official Review · Reviewer_bRzS · 2025-11-04

**Soundness:** 3
**Presentation:** 3
**Contribution:** 3
**Rating:** 6
**Confidence:** 4

**Summary:**

This paper first introduces the HydraFake dataset, which aggregates real and fake images from existing sources and by reimplementing and crawling 10K deepfake samples produced by 10 advanced generators, resulting in 50K real and 50K fake images. It also proposes a two-stage training pipeline for MLLM-based deepfake detection. In the first (SFT) stage, MiPO is introduced to internalize reasoning patterns; in the second stage, the pattern-aware GPRO promotes comprehensive reasoning and enables potential self-reflection. Extensive experiments on in-domain, cross-model, cross-forgery, and cross-domain evaluation sets demonstrate the effectiveness of the proposed method.

**Strengths:**

a.The proposed dataset spans diverse domains and sources of real and manipulated images, including generative face‑swapping, visual autoregressive models, and deepfakes collected from social media.
b.The two‑stage training pipeline substantially outperforms existing deepfake detection methods, as demonstrated in Table 1.

**Weaknesses:**

a.The proposed method employs SFT and GPRO within an MLLM‑based deepfake detection framework—an established post‑training strategy.
b.The difference between the proposed pattern "<fast><planning><reasoning><conclusion>" and the commonly used "<think>... </think>" paradigm has not been analyzed.

**Questions:**

see weaknesses

---

> ### Author Response · Authors · 2025-11-19
> **Author Responses**
>
> > **Q1. The proposed method employs SFT and GPRO——an established post-training strategy.**
>
> We thank the reviewer for their valuable feedback. We would respectfully clarify that our training strategy is different from the well-adopted "SFT+GRPO" pipeline.
>
> Specifically, we recognize the challenge of low-quality reinforcement exploration in deepfake detection: As discussed in recent studies, SFT tends to memorizing the seen corpus [1]. In the field of deepfake deteciton, especially, where the base model shows poor reasoning capacity [2], the SFT model might produce correct answers based on superficial cues or memorized biases. Therefore, naively cascading GRPO after SFT would suffer from low-quality explorations (rollouts), leading to inefficient learning. Building on this observation (**L76-77, L92-93**), we propose to establish a high-quality policy model for deepfake deteciton, and thus propose mixed non-preference optimization (MiPO). Such design greatly enhances the exploration efficiency, as discussed in **L458-459**, resulting in **2.9%** and **2.1%** improvements on cross-forgery and cross-domain settings (**Figure 4**) and significant reasoning quality improvements (Elo rating in **Table 5: 1359.0 vs. 984.0**), compared to the well-established "SFT+GRPO" pipeline.
>
> > **Q2. The difference between the proposed pattern and the commonly used paradigm has not been analyzed.**
>
> Thanks for the suggestion. We would like to politely point out that there seems to be a formatting issue in the comment, as "[object Object]" is displayed. We would be grateful if the reviewer could clarify the intended meaning. In **Table 2**, we have compared the proposed "`<fast> <reasoning> <conclusion> <answer>`" (i.e., pattern-aware reasoning) with "`<think> <answer>`" (i.e., flexible reasoning). Specifically, the "flexible reasoning" in Table 2 is implemented using "`<think>...</think> <answer>...</answer>`" following recent approaches. The results show that our proposed pattern achieves significant improvements, especially on OOD scenarios (i.e., **+6.2%** on cross-foggery and **+3.3%** on cross-domain).
>
> > **References**
>
> [1] SFT memorizes, RL generalizes: A comparative study of foundation model post-training. In ICML, 2025.
>
> [2] LLMs Are Not Yet Ready for Deepfake Image Detection. arXiv preprint arXiv:2506.10474 (2025).

---

### Author Response · Authors · 2025-11-29
**Rebuttal Summary**

Dear AC,

We truly appreciate your time and great effort.
We would like to very **briefly** clarify that:

During rebuttal, we have addressed all raised concerns point‑by‑point.
Reviewer wJ4c raised two insightful questions and actively engaged in the discussion (in their official review column, removed now). After we provided detailed responses about **1)** comparisons with recent MLLM-based detectors, **2)** extensibility of our model, and **3)** open-source plan, they strongly acknowledged our work and raised their score to 8 (bringing the scores to 8886).

We politely mention this only in case it is helpful for your assessment.
If you have any questions to our paper, we are glad to provide further clarification.
Thanks again for your time and consideration. We sincerely appreciate your effort!

---

### Meta-Review · Area_Chair_eLWt · 2026-01-07

**Summary:**

This paper receives 4 high-quality reviews, with 3 positive initial ratings (8, 8, 6) and 1 negative initial rating (4).

The 1 negative reviewer has previously participated in the rebuttal discussions actively. According to the authors' responses, this reviewer's concerns have been addressed.

The rest of reviewers initially have positive ratings, and the authors have provided good responses regarding to the reviewers' concerns.

**Reviewer Concerns:**

All reviewers' concerns have been addressed well.

**Reviewer Scores:**

Three reviewers give positive initial ratings, and the gegative reviewer's concerns have been addressed.

---

### Decision · Program_Chairs · 2026-01-26

Accept (Oral)